# Plant carbonic anhydrase-like enzymes in neuroactive alkaloid biosynthesis

Ryan S. Nett[1,2,3] ✉, Yaereen Dho[4], Chun Tsai[2], Daria Passow[5], Jaime Martinez Grundman[3], Yun-Yee Low[6] & Elizabeth S. Sattely[1,2] ✉

Plants synthesize numerous alkaloids that mimic animal neurotransmitters[1]. The diversity of alkaloid structures is achieved through the generation and tailoring of unique carbon scaffolds[2,3], yet many neuroactive alkaloids belong to a scaffold class for which no biosynthetic route or enzyme catalyst is known. By studying highly coordinated, tissue-specific gene expression in plants that produce neuroactive Lycopodium alkaloids[4], we identified an unexpected enzyme class for alkaloid biosynthesis: neofunctionalized α-carbonic anhydrases (CAHs). We show that three CAH-like (CAL) proteins are required in the biosynthetic route to a key precursor of the Lycopodium alkaloids by catalysing a stereospecific Mannich-like condensation and subsequent bicyclic scaffold generation. Also, we describe a series of scaffold tailoring steps that generate the optimized acetylcholinesterase inhibition activity of huperzine A[5]. Our findings suggest a broader involvement of CAH-like enzymes in specialized metabolism and demonstrate how successive scaffold tailoring can drive potency against a neurological protein target.

The plant kingdom produces many compounds that affect cognition in animals[6]. These molecules probably act to protect against herbivory and also make plants a rich source of therapeutics for treating neurological diseases[1,7]. Many of these neuroactive compounds are alkaloids—nitrogen-containing compounds derived predominantly from amino acids—which act like neurotransmitter mimics to affect animal nervous systems[1]. Neuroactive alkaloids modulate the function of many different proteins involved in neuronal signalling, thereby causing alterations in behaviour and cognition. These bioactivities have long been recognized, as alkaloid-rich plants have served as important botanical medicines for thousands of years and many neuroactive alkaloids, such as the US Food and Drug Administration (FDA)-approved drugs morphine (analgesic), galantamine (dementia treatment) and atropine (muscarinic acetylcholine receptor antagonist), are still used in the clinic[8].

The diversity in neuroactive alkaloid structures in plants is generated through complex biosynthetic mechanisms that convert primary building blocks (for example, amino acids) into a variety of scaffolds which can be tailored to produce specific, bioactive end-products. Alkaloid scaffolds are typically generated by an enzymatic transformation that condenses two substrates to yield a polycyclic structure[2]. However, unlike other major classes of plant natural products (for example, terpenoids and polyketides), there is no single chemical theme or enzyme class that is implicated in alkaloid scaffold generation. For example, although several alkaloid families are generated through Pictet–Spengler condensations, the enzymes which catalyse these reactions belong to unrelated protein families that have convergently evolved this activity[9]. Furthermore, many classes of plant alkaloids are derived through chemical transformations for which there is no known biosynthetic precedent. This is exemplified in the lysine-derived quinolizidine and Lycopodium alkaloids, which consist of hundreds of bioactive compounds[10] and whose scaffolds are thought to be constructed through reactions for which no enzyme catalyst has yet been observed in nature[4,11]. This challenge to readily predict enzymes that build alkaloid scaffolds confounds the rapid elucidation of biosynthetic pathways and suggests that there are enzyme classes in plant metabolism yet to be identified.

Our interest in alkaloid scaffold biogenesis led us to focus on the Lycopodium alkaloids. These molecules are produced by plants in the Lycopodiaceae family (clubmosses)[4] and consist of more than 400 structurally diverse, polycyclic alkaloids that have been studied as toxins and potential medicines[12,13]. Perhaps the most well-known member of this alkaloid class is huperzine A (HupA, **17**)[5], an acetylcholine mimic that reversibly inhibits acetylcholinesterase (AChE), an important enzyme at the neural synapse. This pharmaceutical activity has led to interest in the use of **17** as a potential treatment for the symptoms of dementia[14] and elucidating its biosynthesis offers the possibility for the engineered production of this molecule, which has historically been non-sustainably sourced from wild *Huperzia* plants[15]. More broadly, the complexity and diversity of structures in the Lycopodium alkaloids has intrigued chemists for more than a century[16] and these compounds continue to be targets for chemical synthesis strategies and isolation of unique structures[17]. However, although significant progress has been made in their total syntheses[18,19], the mechanisms that plants use to synthesize the many, diverse Lycopodium alkaloid scaffolds have remained largely unknown and suggest the involvement of previously undescribed enzyme classes.

[1]Department of Chemical Engineering, Stanford University, Stanford, CA, USA. [2]HHMI, Stanford University, Stanford, CA, USA. [3]Department of Molecular and Cellular Biology, Harvard University, Cambridge, MA, USA. [4]Department of Chemistry, Stanford University, Stanford, CA, USA. [5]Biophysics Program, Stanford University, Stanford, CA, USA. [6]Department of Chemistry, Faculty of Science, Universiti Malaya, Kuala Lumpur, Malaysia. ✉e-mail: rnett@fas.harvard.edu; sattely@stanford.edu

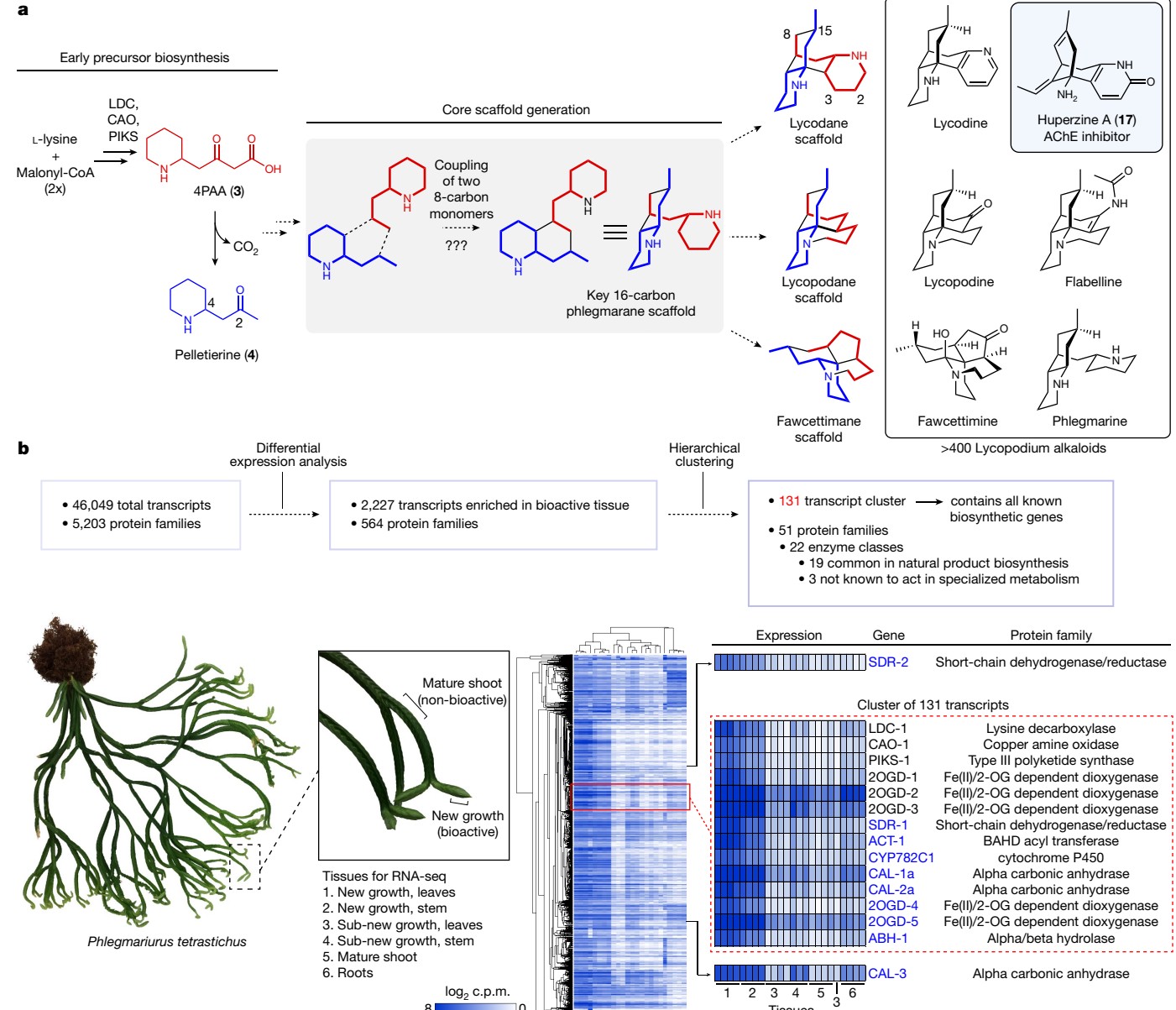

**Fig. 1 | Assessing unknown steps in Lycopodium alkaloid biosynthesis.**
**a**, An unknown series of chemical transformations are necessary to convert early precursors into the diverse Lycopodium alkaloid scaffolds found in club mosses. Note that the scaffold types shown in red/blue here are not compounds observed in nature but are visual representations of the main structural classes in the Lycopodium alkaloids. Shown in boxes are representative Lycopodium alkaloids, including the AChE inhibitor, HupA (**17**). **b**, Overview of transcriptomic-guided workflow for identifying new biosynthetic enzyme candidates. c.p.m., counts per million. The image of *Phlegmariurus tetrastichus* is adapted with permission from ref. 23 (Nett et al.), National Academy of Sciences.

## Discovery of scaffold-generating enzymes

Previous isotope tracer studies (Supplementary Fig. 1) have demonstrated that the Lycopodium alkaloid scaffolds originate from two units each of a lysine-derived heterocycle (1-piperideine, **1**) and a polyketide substrate derived from malonyl-CoA (3-oxoglutaric acid, **2**, or its thioester analogue)[4]. These experiments enabled the recent identification of a biosynthetic route to 4-(2-piperidyl)acetoacetic acid (4PAA, **3**) and pelletierine (**4**), the likely building blocks for all Lycopodium alkaloids (Fig. 1a), in several clubmoss species[20–23]. In our previous work, we demonstrated that three enzymes from the HupA-producing clubmoss *Phlegmariurus tetrastichus* (lysine decarboxylase, *Pt*LDC; copper amine oxidase, *Pt*CAO; and piperidyl ketide synthase, *Pt*PIKS) are sufficient to convert the primary metabolites L-lysine and malonyl-CoA into **3**, which can spontaneously decarboxylate to yield **4** (Fig. 1a)[23].

Although radio-isotope labelling studies with **4** have demonstrated this compound to be incorporated into downstream alkaloids, it was determined that this 8-carbon precursor is only incorporated into one half of 16-carbon Lycopodium alkaloid scaffolds[24, 25] (Fig. 1a and Supplementary Fig. 1). By contrast, L-lysine, cadaverine, **1** and **2**, which are presumed precursors to **4**, were all shown to be incorporated into both halves of this scaffold[24, 26–30] (Supplementary Fig. 1). These data suggest that a phlegmarane-type scaffold (Fig. 1a) is formed through the pseudodimerization of a **4**-like molecule and a compound from which it is irreversibly derived, which has been proposed to be **3** or an oxidized derivative[24,25].

Although a condensation between **3** and **4** is plausible (Supplementary Fig. 1), it was unclear what type of enzyme could catalyse this type of reaction. Moreover, it was not clear if **3** and/or **4** needed to be further tailored before coupling. Because of this, we chose to rely heavily on

the high level of biosynthetic gene co-expression that we had previously observed in our transcriptome of *P. tetrastichus*[23]. To leverage our previous results, we performed hierarchical clustering with these data to generate co-expressed clusters of transcripts (Fig. 1b). This analysis revealed a single co-expressed cluster of 131 transcripts (cluster131) that contained all the previously identified biosynthetic genes (*Pt*LDC-1, *Pt*LDC-2, *Pt*CAO-1, *Pt*CAO-2, *Pt*PIKS-1, *Pt*PIKS-2, *Pt*2OGD-1, *Pt*2OGD-2 and *Pt*2OGD-3)[23]. This cluster was highly enriched with transcripts encoding for metabolic enzymes from several protein families commonly involved in natural product biosynthesis—for example, cytochromes P450 (CYPs), Fe(II)/2-oxoglutarate-dependent dioxygenases (2OGDs), methyltransferases, acyltransferases and dehydrogenase/reductase enzymes—suggesting that it may contain the requisite biosynthetic machinery for Lycopodium alkaloid scaffold biosynthesis.

It had previously been proposed that a **4**-derived diene (**8**) could potentially serve as one of the cosubstrates for scaffold formation[24]. We considered that the formation of this compound would require two key events: an oxidation of **4** to form the imine and the reduction and elimination of the ketone oxygen. A related sequence of transformations had been reported in the context of morphine biosynthesis[31], suggesting its plausibility in Lycopodium alkaloid biosynthesis. To test candidate enzymes for this proposed route, we used *Agrobacterium*-mediated DNA delivery in *Nicotiana benthamiana* as a transient gene expression platform. This allowed for production of **3** and **4** as substrates and the combinatorial testing of selected gene candidates (Methods). We were unable to identify an oxidase that could act directly on **4**, so we instead gave priority to dehydrogenase/reductase family enzymes in cluster131 that could potentially catalyse the ketone reduction. Only one short-chain dehydrogenase/reductase (SDR) family gene was found in this cluster (*Pt*SDR-1) and this had a close homologue (*Pt*SDR-2; 88.6% amino acid identity) which could be found in a slightly expanded co-expression cluster (273 transcripts; cluster273). When added to the transient expression experiments in *N. benthamiana* (containing *Pt*LDC, *Pt*CAO and *Pt*PIKS), both SDR homologues led to a decrease of **4** and the detection of two mass ions through liquid chromatography–mass spectrometry (LC–MS) that correspond to reduction of the ketone to the alcohol ([M + H]$^+$ = $m/z$ 144.1383) (Fig. 2 and Extended Data Fig. 1a,b). Comparison to a standard of 1-(piperidin-2-yl)propan-2-ol (that is, reduced pelletierine) stereoisomers (**5**) confirmed these two compound peaks to be diastereomers of **5**. *Pt*SDR-1 and *Pt*SDR-2 seemed to form different ratios of **5** stereoisomers (Extended Data Fig. 1a). It had previously been noted that *Pt*PIKS produces racemic **4** (and therefore **3**, as **4** can be derived through spontaneous decarboxylation of **3**)[22], which suggested that these SDR enzymes may preferentially act on different enantiomers of **4** as substrate. Chiral LC–MS analysis confirmed the production of racemic **4** by *Pt*PIKS and further demonstrated that *Pt*SDR-1 mainly consumed (*S*)-**4** to produce (2*S*, 4*S*)-**5** (otherwise known as (+)-sedridine) but also apparently acted on (*R*)-**4** to produce a small amount of (2*S*, 4*R*)-**5** (otherwise known as (+)-allosedridine) (Extended Data Fig. 1c,d). *Pt*SDR-2 consumed both enantiomers of **4** to produce an equimolar amount of (2*S*, 4*S*)-**5** and (2*S*, 4*R*)-**5** (Extended Data Fig. 1c,d) and seemed to be more active in our system. Taken together, these results demonstrate that these SDR enzyme homologues each catalyse the ketone reduction of **4** with conserved stereoselectivity to yield an alcohol in the (*S*) orientation but also that they have different enantioselectivity, with *Pt*SDR-1 preferably reducing (*S*)-**4**, whereas *Pt*SDR-2 seems to act equally well on both enantiomers of **4**.

Given the precedent for O-acylations in generating leaving groups for elimination in natural product biosynthesis[31,32], we next screened BAHD acyltransferase family enzymes for activity, as six unique gene sequences from this family could be found in cluster131. Adding one acyltransferase (*Pt*ACT-1) to the transient co-expression system led to consumption of both (2*S*, 4*S*)-**5** and (2*S*, 4*R*)-**5** and production of a new compound ([M + H]$^+$ = $m/z$ 186.1489) consistent with the addition of an

O-acetyl group (Fig. 2 and Extended Data Fig. 1e, f). Comparison to a synthesized standard and chiral LC–MS analysis confirmed this to be a mixture of two O-acetylated diastereomers, (2*S*, 4*S*)-**6** and (2*S*, 4*R*)-**6**, which shows that *Pt*ACT-1 can catalyse O-acetylation regardless of the stereochemistry of the piperidine-alkyl (C3–C4) bond (Extended Data Fig. 1g,h).

The production of (2*S*, 4*S*)-**6** and (2*S*, 4*R*)-**6** was consistent with our hypothesis for elimination-mediated formation of the proposed diene (**8**). Because formation of **8** would require oxidation of the O-acetylated substrate(s), we next screened CYP and 2OGD family enzymes found in cluster131. One CYP enzyme (*Pt*CYP782C1) was found to consume both (2*S*, 4*S*)-**6** and (2*S*, 4*R*)-**6** in our transient expression system (Fig. 2 and Extended Data Fig. 2a). This coincided with the presence of two new compounds: one that corresponded to a single oxidation (desaturation) of the O-acetylated substrate (**7**, [M + H]$^+$ = $m/z$ 184.1332, retention time 1.86 min), as well as another which shared the same exact mass as **8** ([M + H]$^+$ = $m/z$ 124.1121, retention time 1.93 min). Both compounds were almost entirely lost if samples were incubated at room temperature for 1 h (Extended Data Fig. 2b), which is consistent with previous descriptions of the instability of **8** (ref. 33). Although this limited our ability to access authentic standards, tandem mass spectrometry (MS/MS or MS$^2$) analysis supported the proposed structures of **7** and **8** and ultraviolet (UV) analysis of **8** corroborated the presence of the predicted α,β-unsaturated imine in this molecule[34] (Extended Data Fig. 2c,d). Reactions of **6** (mixture of stereoisomers) with *Pt*CYP782C1-enriched microsomes produced in yeast confirmed the activity of this enzyme (Extended Data Fig. 2e–g) and allowed us to access **8** as an in vitro-generated substrate for any downstream enzymatic studies. These results indicate a series of transformations in which the diastereomers of **6** are oxidized to produce **7**, which then undergoes an allylic elimination to yield **8** (Extended Data Fig. 2h,i), although it is uncertain whether the elimination is spontaneous or enzyme-catalysed by *Pt*CYP782C1.

Production of **8** in our transient expression system suggested that we had potentially accessed the relevant substrates for initial phlegmarane scaffold formation (Fig. 1a). However, it was difficult to select candidate enzymes given the lack of precedence for enzymes that could promote this type of chemistry; furthermore, it was uncertain exactly what 'dimer' substrate combination should be tested. As a more untargeted approach, we opted to test candidates from cluster131 in batch combinations by enzyme family regardless of their previous association with specialized metabolism. In this process, we observed that a batch of four α-carbonic anhydrase (CAH) family proteins produced a new mass signature ([M + H]$^+$ = $m/z$ 164.1434) when transiently expressed in *N. benthamiana* leaves with the rest of the established pathway (Fig. 2 and Extended Data Fig. 3a,b). The calculated molecular formula of this feature ($C_{11}H_{18}N$) was unanticipated given the expected 8-carbon substrates and this metabolite was only found using untargeted metabolomic analysis of our data[35]. However, further analysis of in-source MS adducts and fragments[36] identified two co-eluting mass signatures that corresponded to the mass of a 16-carbon molecule ([M + H]$^+$ = $m/z$ 247.2169 and [M + 2H]$^{2+}$ = $m/z$ 124.1121), suggesting that we had potentially accessed a scaffold from the condensation of two 8-carbon, nitrogen-containing substrates (Extended Data Fig. 3c,d). With this in mind, the $m/z$ 164 ion seemed to correspond to an ionization-induced loss of 1-piperideine during MS analysis (Extended Data Fig. 3d). We note that of the three MS adducts observed for this molecule, the $m/z$ 164 ion was the most abundant and therefore this was used as a diagnostic ion for all following analyses. Subsequent MS$^2$ fragmentation of both the parent ion ($m/z$ 247) and the in-source fragment ($m/z$ 164) suggested that this compound (designated as **9**, although the structure was not immediately evident) possesses a bicyclic, phlegmarane-type scaffold (Extended Data Fig. 3e–h) and UV analysis supported the presence of an α,β-unsaturated imine[34] (Extended Data Fig. 3i). Compound **9** could be detected in extracts

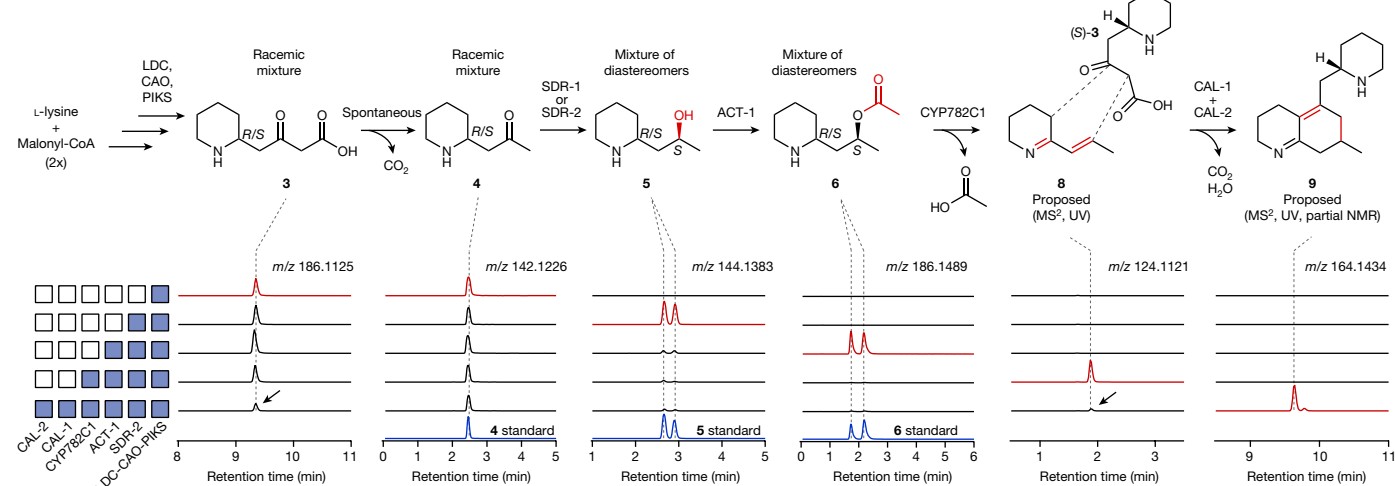

**Fig. 2 | Stepwise discovery of early biosynthetic enzymes contributing to scaffold formation.** Shown are the extracted ion chromatograms (EICs) pertaining to the relevant *m/z* value for each proposed intermediate (*m/z* values shown under each compound) upon the transient co-expression of candidate biosynthetic genes from *P. tetrastichus* (blue boxes) in *N. benthamiana*. In general, compounds were observed as the [M + H]⁺ ion. For **9**, the in-source fragment *m/z* 164.1434 ([M-C₅H₉N + H]⁺) is the principal detected ion and thus serves as a diagnostic for this compound. All compounds were detected through LC–MS using a HILIC column, with the exception of **6** diastereomers, which were observed by using a C18 column. Note that *y* axes for each set of chromatograms are on different scales but scales are constant within an EIC plot. Black arrows indicate observed depletion of substrates on addition of CAL-1 and CAL-2. The natures of compounds **4**–**6** were confirmed through comparison to synthesized or commercially available standards. The structure of **8** is proposed on the basis of MS² and UV analysis (Extended Data Fig. 2). The structure of **9** is proposed on the basis of MS², UV analysis and partial NMR of purified **9** (Extended Data Fig. 3 and Supplementary Figs. 10 and 11), as well as MS² and full NMR of an oxidized byproduct (**9′**, Extended Data Fig. 3 and Supplementary Figs. 12–18).

from the biosynthetically active tissue of *P. tetrastichus* (Extended Data Fig. 3e,f), which gave us confidence that this compound was relevant to Lycopodium alkaloid metabolism. We subsequently found that two of the batch-tested CAH-like (CAL) proteins (named as *Pt*CAL-1a and *Pt*CAL-2a) were required to be transiently expressed with the rest of the upstream pathway for this scaffold formation to occur and that no apparent activity could be detected with either of the CALs on their own (Fig. 2 and Extended Data Fig. 3a). Also, we found that cluster131 contained homologues of *Pt*CAL-1a (*Pt*CAL-1b, 89.6% amino acid identity) and *Pt*CAL-2a (*Pt*CAL-2b, 70.1% amino acid identity) which exhibited the same activity (Extended Data Fig. 3b).

We considered that formation of **9** could result from a dimerization of **8**. However, when **8** was provided as a substrate to *Pt*CAL-1a and *Pt*CAL-2a independently of the full reconstituted pathway in *N. benthamiana* (**8** was generated by co-infiltrating **6** as a substrate for transiently expressed *Pt*CYP782C1), formation of **9** was not observed (Extended Data Fig. 3j,k). Given this result, we predicted that the scaffold may be a pseudodimer that requires **8** and another upstream pathway intermediate as a cosubstrate. In support of this hypothesis, we could reconstitute production of **9** in *N. benthamiana* through the combination of *Pt*CAL-1a/*Pt*CAL-2a with a module for producing **8** (*Pt*CYP782C1 and synthetic **6** as substrate) and a module for producing **3** and **4** (*Pt*LDC, *Pt*CAO and *Pt*PIKS) (Extended Data Fig. 3j,k). Also, we observed consumption of both **8** and **3** that was concurrent with production of **9**, which supports these compounds as the 'pseudodimers' that are condensed to form the scaffold molecule (Fig. 2 and Extended Data Fig. 3l). Because *Pt*PIKS produces a racemic mixture of **3** (Extended Data Fig. 1c; note that the stereochemistry of **3** is inferred by the measurement of **4** enantiomers), it was plausible that either (*R*)-**3** or (*S*)-**3** could be incorporated into this scaffold, which would result in the formation of two **9** diastereomers. Indeed, under optimal LC–MS conditions, we could observe a second, nearly co-eluting peak with an identical MS² fragmentation pattern to **9** (Extended Data Fig. 3m). However, this compound was only present at around 10% the amount of the main **9** diastereomer, which suggests that a single enantiomer

of **3** is preferably used as the cosubstrate. Through chiral LC–MS, we determined that (*S*)-**3** was partially consumed by *Pt*CAL-1a/*Pt*CAL-2a (28% decrease, *P* = 0.09) whereas (*R*)-**3** was not (*P* = 0.66), which supports the specific condensation of (*S*)-**3** with **8** to form **9** (Extended Data Fig. 3n). Overall, these results from heterologous pathway expression in *N. benthamiana* are consistent with previously proposed mechanisms that implicate **3** as the nucleophile to initiate scaffold formation with an electrophilic cosubstrate, which we have shown to be **8** (refs. 24,25).

We scaled up production of **9** in *N. benthamiana* for purification and structural determination of this compound. This proved to be difficult, as the compound seemed to degrade during purification and we were only able to obtain a moderately pure proton NMR. However, we were able to purify a putative oxidized product of this scaffold (**9′** [M + H]⁺ = *m/z* 263.2118) that accumulated during purification (Extended Data Fig. 3o–q) and structural analysis of this molecule through NMR and MS² confirmed that it contained the predicted phlegmarane scaffold (see Supplementary Information for NMR data of **9** and **9′**). Considering the structure of **9′**, MS² fragmentation and UV analysis of **9** and the chemical logic of a condensation between (*S*)-**3** and **8**, we predict the structure of **9** as a bicyclic phlegmarane scaffold with a conjugated α/β-unsaturated imine (Figs. 2 and 3a). Notably, a similar α/β-unsaturated imine moiety has been used in the chemical synthesis of Lycopodium alkaloid scaffolds[17], for which it was noted to be oxygen sensitive and the NMR structure of our oxidized byproduct (**9′**) is consistent with an oxidation of **9**. Thus, we propose that *Pt*CAL-1 and *Pt*CAL-2 homologues act together to form **9** through the condensation of (*S*)-**3** and **8** and that this serves as the key phlegmarane scaffold-forming reaction in Lycopodium alkaloid biosynthesis (Fig. 3a and Extended Data Fig. 3r).

The ability of CAH-like proteins to act directly in specialized metabolite biosynthesis represents a striking neofunctionalization in this enzyme family, as CAH enzymes canonically catalyse the interconversion of $CO_2$ and bicarbonate as an aspect of numerous biological functions including pH control, $CO_2$ concentrating/solubilization and lipid metabolism[37]. Also, we were surprised by the apparent cofunctionality

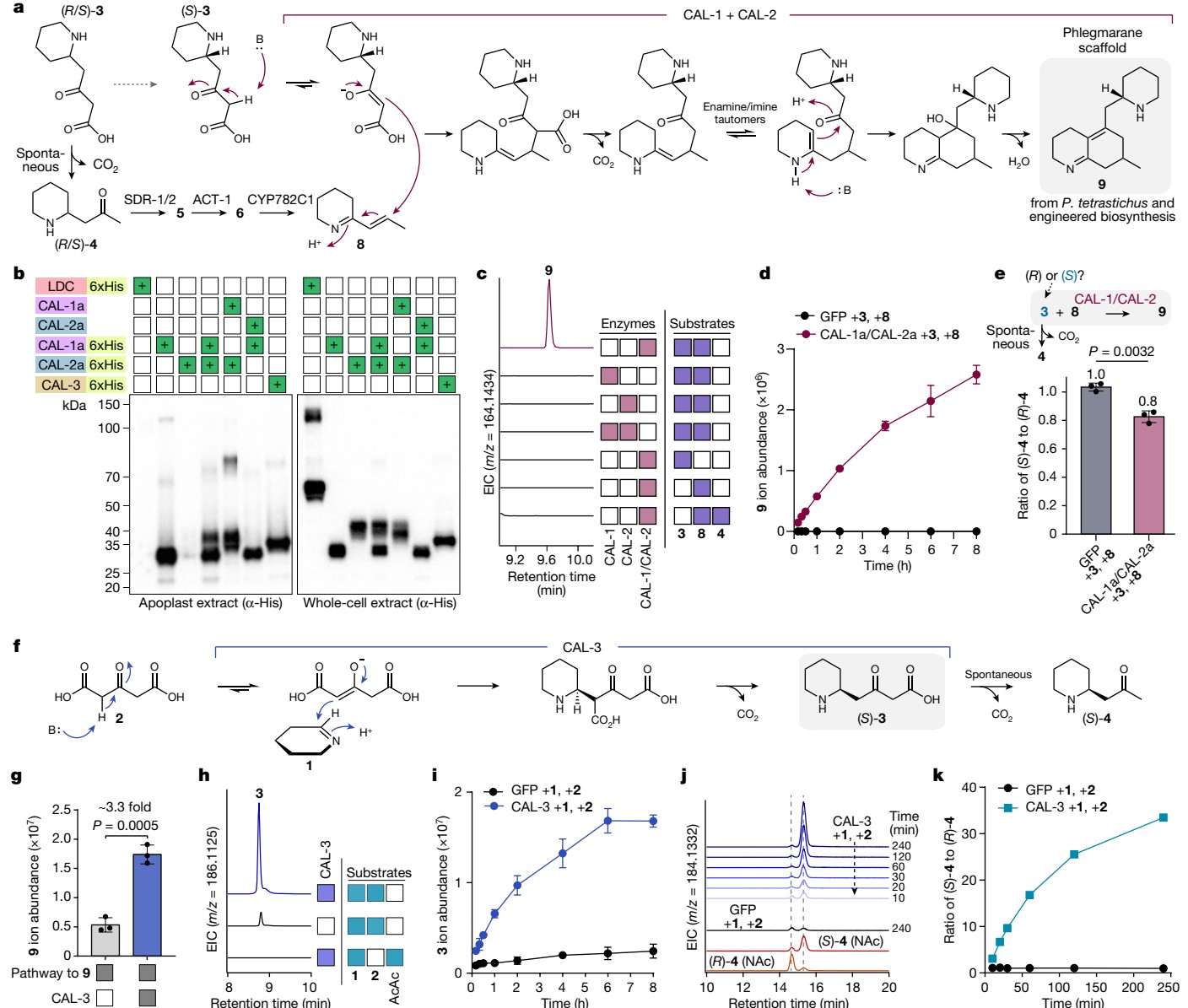

**Fig. 3 | Neofunctionalized CAL enzymes in Lycopodium alkaloid biosynthesis. a**, Proposed mechanism for biosynthesis of **9** by *Pt*CAL-1a/*Pt*CAL-2a. **b**, Representative western blot of 6xHis-tagged CALs expressed alone or co-expressed with untagged gene constructs in *N. benthamiana*. A 6xHis-tagged *Lc*LDC construct was included as an intracellular protein control. This experiment was performed more than three times with similar results routinely observed. **c**, EICs for production of **9** (*m/z* 164.1434) by apoplastic *Pt*CAL-1a and *Pt*CAL-2a (alone or co-expressed) with different substrate combinations. **d**, Time course of **9** production by apoplastic *Pt*CAL-1a/*Pt*CAL-2a compared to a GFP apoplast control. *n* = 3 reactions per condition. **e**, Assessment of which **3** enantiomer is used as a substrate in the formation of **9**. Because **3** enantiomers could not be directly observed, chirality was inferred by measuring **4** enantiomers, which form through spontaneous decarboxylation of **3**.

Enantiomers were analysed as N-acetylated derivatives. Average ratios are above each bar. *n* = 3 reactions per condition. **f**, Proposed condensation of **1** and **2** catalysed by *Pt*CAL-3 to produce (*S*)-**3**. **g**, Co-expression of *Pt*CAL-3 with the rest of the pathway required to produce **9**. *n* = 3 infiltrated leaves per condition. **h**, EICs for **3** (*m/z* 186.1125) production by *Pt*CAL-3 apoplast protein with different substrates. **i**, Formation of **3** over time by apoplastic *Pt*CAL-3 with **1** and **2** as substrates, as compared to a GFP apoplast control. *n* = 3 reactions per condition. **j**, Assessment of enantiospecific product formation by *Pt*CAL-3 through chiral LC–MS analysis of **4** enantiomers (after N-acetylation). **k**, Ratio of (*S*)-**4** and (*R*)-**4** over time in the apoplastic *Pt*CAL-3 reaction. For bar and line graphs, plotted values represent the mean, with error bars representing ±s.d. Statistical analyses were performed using a two-tailed Welch's *t*-test assuming unequal variance.

of *Pt*CAL-1a/*Pt*CAL-2a because proteins from the CAH family usually function as monomers[38,39]. To better understand the unique functionality of *Pt*CAL-1a/*Pt*CAL-2a, we next worked to establish an in vitro reaction assay. Although we could obtain solubilized versions of these proteins through both heterologous expression in *Escherichia coli* and cell-free protein production with wheat germ extract, we were unable to recapitulate the previously observed enzyme activity for *Pt*CAL-1/*Pt*CAL-2 obtained from either system, which indicated that there may be factors

in the context of living plant cells that are critical for activity (for example, post-translation modifications or subcellular localization). Both proteins possess a predicted N-terminal signalling peptide that indicates trafficking through the secretory pathway, which suggested that they may be localized to the apoplast (the extracellular compartment in plant leaves). To assess this, we produced His-tagged versions of these proteins in *N. benthamiana* and used western blotting of different protein fractions (apoplast and cellular) to evaluate *Pt*CAL-1a and *Pt*CAL-2a

localization. This demonstrated that both proteins can be found in the apoplast in this heterologous system but also that this localization is affected by their co-expression (Fig. 3b). In particular, whereas *Pt*CAL-1a exhibited apoplastic localization independently of co-expression with *Pt*CAL-2a, very little *Pt*CAL-2a protein could be detected in the apoplast when it was expressed alone and it instead seemed to be mainly in the intracellular fraction, which contains both cytosolic and organellar proteins (Fig. 3b). However, on co-expression of *Pt*CAL-1a, apoplastic *Pt*CAL-2a was readily detected and seemed to exhibit post-translational modifications of an unknown nature (Fig. 3b). These results indicate that *Pt*CAL-1a may have a critical role in the proper post-translational modification and/or trafficking of *Pt*CAL-2a, although further work will be necessary to determine the mechanism by which this occurs. Beyond providing details on localization, this information was critical for enzyme assay development because the pH of the apoplast is typically relatively low (about pH 5)[40], which suggested that these proteins may have optimal function at a lower pH. Also, apoplast extracts can be readily isolated from *N. benthamiana* leaves expressing these proteins[40], thereby providing a potential means to evaluate CAL protein outside the living plant system.

Using isolated apoplast extracts in vitro, we demonstrated activity (Extended Data Fig. 4a–d) for *Pt*CAL-1a/*Pt*CAL-2a when **3** and **8** were supplied as substrates enzymatically (through the action of purified *Pt*PIKS +**1** and malonyl-CoA and *Pt*CYP782C1-enriched microsomes +**6** and NADPH, respectively). Notably, the enzymatic activity of *Pt*PIKS yields both **3** and **4** (through spontaneous decarboxylation of **3**) and no production of **9** was observed for *Pt*CAL-1a/*Pt*CAL-2a when synthesized **4** was added together with **8** (Fig. 3c and Extended Data Fig. 4c), which further supports **3** as the cosubstrate. We confirmed that the free acid of **3** (as opposed to a thioester conjugate) acts as the cosubstrate, as **3** generated in situ (through spontaneous condensation of **1** and **2**) could replace the *Pt*PIKS enzyme reaction in this in vitro system (Fig. 3c,d and Extended Data Fig. 4b–d). As observed with in planta experiments, only the (*S*) enantiomer of **3** was consumed in the presence of *Pt*CAL-1a/*Pt*CAL-2a (Fig. 3e and Extended Data Fig. 4e, *P* = 0.0032), thereby supporting enantiospecific scaffold generation with this cosubstrate. Consistent with *N. benthamiana* experiments and western blot analysis, we only observed activity in apoplast extracts from leaves in which *Pt*CAL-1a and *Pt*CAL-2a were co-expressed; individual expression of each protein and subsequent mixing did not yield detectable product formation (Fig. 3c and Extended Data Fig. 4c), indicating that the co-occurrence of each protein in the plant is critical for their proper production and function.

Condensation to form **9** could initiate through the nucleophilic attack of (*S*)-**3**, a β-keto acid, with the electrophilic, α,β-unsaturated imine of **8**. In theory, this may occur through two routes (Extended Data Fig. 4f): (1) decarboxylation of (*S*)-**3** to generate the enolate of **4**, which would then serve as the nucleophile for condensation with **8** ('decarboxylation-first' mechanism) or (2) formation of the (*S*)-**3** enolate through tautomerization, followed by addition to **8**, then decarboxylation ('addition-first' mechanism)[41]. Although a decarboxylation-first mechanism is reminiscent of several canonical strategies for C–C bond formation (for example, in fatty acid biosynthesis), an addition-first mechanism could also leverage $CO_2$ release to drive the reaction equilibrium to completion and thus seemed to be a plausible alternative. To probe these possibilities, we designed experiments to test if the CAL proteins accelerated decarboxylation (and presumably enolate formation) in the absence of their respective electrophiles. When we supplied **3** as the substrate to *Pt*CAL-1a/*Pt*CAL-2a in the absence of **8**, we did not observe accelerated decarboxylation of **3** (beyond the rate of spontaneous decarboxylation of this β-keto acid) (Extended Data Fig. 4g,h). This result and the fact that **4**, which would be in equilibrium with an equivalent enol tautomer, does not serve as a cosubstrate with **8** (Fig. 3c and Extended Data Fig. 4c), suggests that the addition of (*S*)-**3** to **8** probably precedes decarboxylation. This mechanism implies that

the CAL proteins may enhance formation of the enolate tautomer of **3**, which could serve as the requisite nucleophile (Extended Data Fig. 4f); however, our results do not rule out the possibility that binding of **8** is required for the decarboxylation of (*S*)-**3** to occur. Although further work will be necessary to firmly establish this enzymatic mechanism, our data suggest an addition-first mechanism by which one half of the Lycopodium alkaloid scaffold (**8**) is combined with a cosubstrate (**3**) from which it is irreversibly derived (Fig. 3a). The use of **3** as the nucleophilic cosubstrate for scaffold formation is in direct agreement with past isotope labelling studies that demonstrated incorporation of **4** into only one half of Lycopodium alkaloids scaffolds[24,25]. In this mechanism, the '**4**-derived' half is represented by **8**, which we have shown is enzymatically synthesized from **4**, whereas (*S*)-**3** serves as the other half. Although we favour the role of (*S*)-**3** as the initial nucleophile attacking the **8** electrophile in this reaction, we note that an alternative sequence of bond formation is also plausible. For example, formation of the enamine tautomer of **8** could allow for this molecule to serve as the initial nucleophile, wherein the enamine would attack the carbonyl of (*S*)-**3** first, followed by decarboxylative condensation to generate the final phlegmarane scaffold.

Encouraged by the identification of these neofunctionalized CAH proteins, we considered that other transcriptionally coregulated CAL genes (Pearson's *r* > 0.9 when compared to expression of other pathway genes) might also have a role in this biosynthetic pathway. On testing another four CAL candidates with our established biosynthetic pathway through transient expression in *N. benthamiana*, we found a distinct CAL gene (*Pt*CAL-3) that caused about a threefold increase (*P* = 0.0005) in the abundance of **9** (Fig. 3g). Analysis of all pathway intermediates that accumulated in this experiment showed a shift in the abundance of **5** diastereomers to an enrichment of (2*S*, 4*S*)-**5**, suggesting that *Pt*CAL-3 could be acting to influence the stereochemistry of precursor substrates (Extended Data Fig. 5a). Including *Pt*CAL-3 with different combinations of pathway genes demonstrated that this enzyme is acting upstream of **4** formation, as we could observe a shift from racemic **4** to an enrichment of (*S*)-**4** when *Pt*CAL-3 was included (Extended Data Fig. 5b–d, *P* = 0.009). As with *Pt*CAL-1a and *Pt*CAL-2a, *Pt*CAL-3 contains an N-terminal signalling peptide and was found to be localized to the apoplast (Fig. 3b) and we were able to establish functional in vitro assays using apoplast extract from *N. benthamiana* leaves expressing this gene. In these assays, we demonstrated that including *Pt*CAL-3 apoplast with the PIKS reaction (using **1** and malonyl-CoA as substrates) led to an enrichment of (*S*)-**4** to (*R*)-**4** over time (Extended Data Fig. 5h) and we speculated that *Pt*CAL-3 protein may be accelerating the rate of **3** and/or **4** formation in a stereoselective manner. To decouple the activity of *Pt*CAL-3 from *Pt*PIKS, we generated **3** as a substrate in situ through the spontaneous condensation of **1** and **2** (Extended Data Fig. 4a). When this reaction mixture was added to *Pt*CAL-3 apoplast, we observed a drastically accelerated increase in the formation of **3** in comparison to a control apoplast extract (Fig. 3h, i and Extended Data Fig. 5i). Also, we determined that the (*S*) enantiomer of **3** (inferred through measurement of **4** enantiomers) was enriched over time (Fig. 3j,k and Extended Data Fig. 5j,k), which indicated that *Pt*CAL-3 is catalysing a stereospecific condensation of **1** and **2**. Similar to the proposed addition-first mechanism of *Pt*CAL-1a/*Pt*CAL-2a, *Pt*CAL-3 did not accelerate the rate of decarboxylation of **2** (Extended Data Fig. 5l, m). Furthermore, acetoacetate, the product of the decarboxylation of **2**, did not serve as a viable cosubstrate with **1** (Fig. 3h). Thus, our data suggest a mechanism in which *Pt*CAL-3 catalyses a Mannich-like addition of a **2** enolate to the imine of **1** in a stereospecific manner, after which decarboxylation occurs to yield (*S*)-**3** (Fig. 3f). This mechanism aligns well with the observed data from biosynthetic pathway reconstitution in *N. benthamiana* because *Pt*PIKS produces **2** as a major product[22,23] and thus the requisite substrates for *Pt*CAL-3 (**1** and **2**) are present from the activity of earlier enzymes in the pathway (*Pt*CAO and *Pt*PIKS).

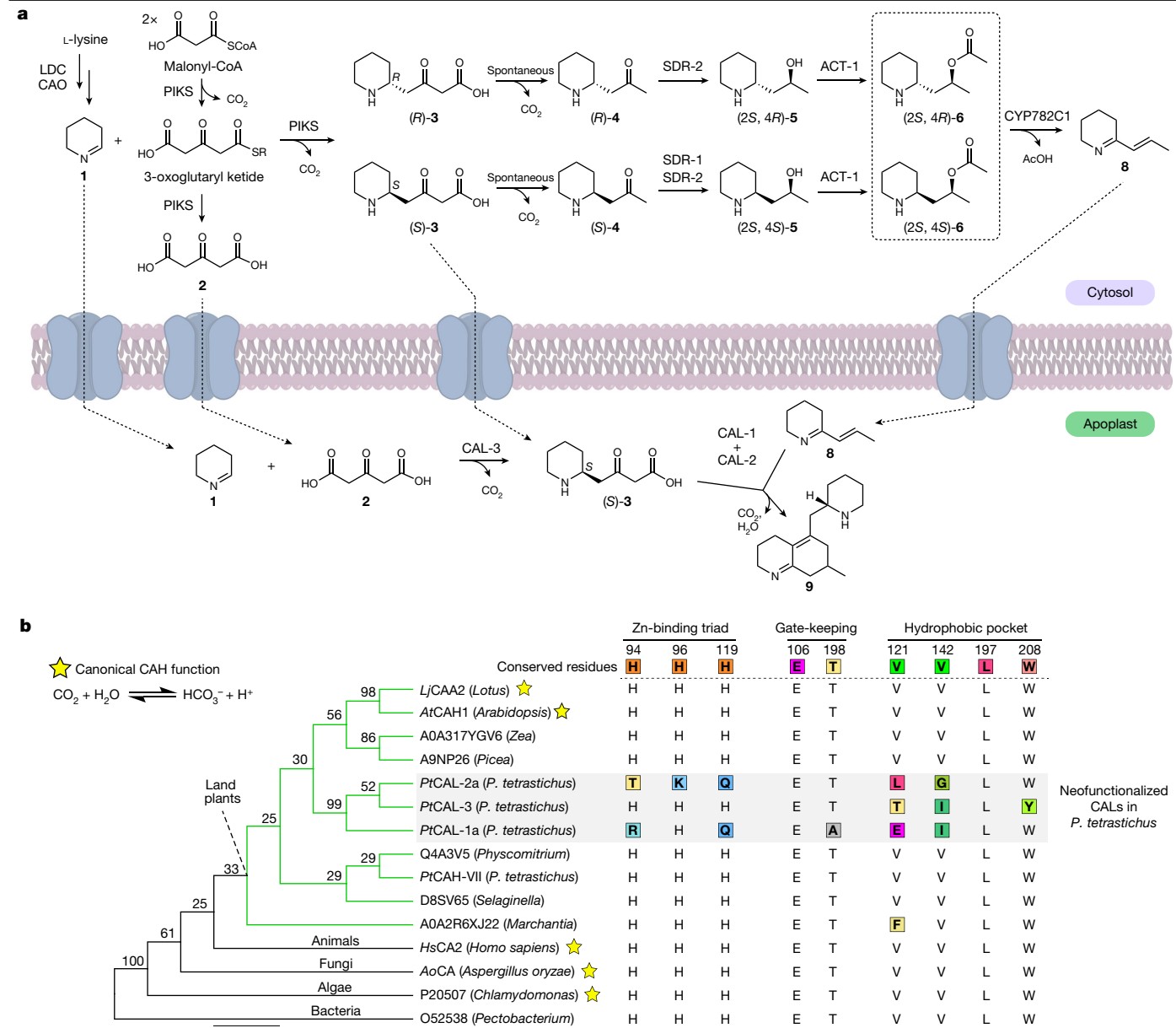

**Fig. 4 | A prominent role for CAL enzymes in early Lycopodium alkaloid biosynthesis. a**, Biosynthetic proposal for the early chemical transformations in Lycopodium alkaloid biosynthesis. Note that transport of intermediates across the membrane is speculative. **b**, Phylogenetic tree (MUSCLE alignment, neighbour-joining tree) of CAH family proteins from several kingdoms of life. Bootstrap values (100 replicates) are located at nodes. Also shown are the main active site residues for each aligned protein, with numbering corresponding to human carbonic anhydrase 2 (*Hs*CA2, UniProt ID: P00918). Changes to the canonical/conserved sequence are highlighted in coloured boxes. Stars indicate proteins that have verified canonical CAH activity. An expanded alignment/phylogenetic tree can be found in Supplementary Fig. 2.

Beyond elucidation of the pathway for initial scaffold formation, our results on early Lycopodium alkaloid biosynthesis help rationalize the observed synthesis of racemic **3** and **4** by the PIKS enzyme[22,23], which was relatively unusual given that most enzymes synthesize products in an optically pure form[42]. Specifically, we have shown that the subsequent enzymes in the biosynthesis of **8** (for example, *Pt*SDR-2, *Pt*ACT-1 and *Pt*CYP782C1) lack substrate stereoselectivity and thus both the (*R*) and (*S*) enantiomers of **4** can be converted into **8** (Fig. 4a). Although this is unusual for a metabolic pathway, we predict that the stereoselectivity of these particular enzymes may not have been strongly selected for during the evolution of **8** biosynthesis because the stereocentre of **4** (as well as **5** and **6**) is eventually lost in the formation of **8**. Ultimately, the activity of *Pt*CAL-3 provides a bypass of these events to generate a specific enantiomer, (*S*)-**3**, for scaffold generation. This scenario would necessitate

movement of **1** and **2** into the apoplast for *Pt*CAL-3 because this protein seems to be secreted extracellularly. Because *Pt*CAL-1a/*Pt*CAL-2a condense (*S*)-**3** with **8** to generate the core Lycopodium alkaloid scaffold, the specific production of (*S*)-**3** by *Pt*CAL-3 helps to explain the observed boost in the production of **9** when *Pt*CAL-3 is present in *N. benthamiana* pathway reconstruction and in vitro enzyme assays (Fig. 3g and Extended Data Fig. 4d). Indeed, the addition of *Pt*CAL-3 also leads to a significantly increased ratio (increased from 10:1 to 50:1, *P* = 0.01) of **9** over its minor diastereomer (Extended Data Fig. 5e,f). We predict that formation of the minor diastereomer is because of low-level use of (*R*)-**3** as a substrate by *Pt*CAL-1a/*Pt*CAL-2a and that the increased proportion of (*S*)-**3** from *Pt*CAL-3 activity leads to further enrichment of **9** as the main product. The probable colocalization of these CAL proteins in the apoplast provides a mechanism by which (*S*)-**3** can be directly used

in scaffold formation without being fully consumed by the enzymatic steps that synthesize **8**, which are localized to the cytosol (Fig. 4a). Together, these data reveal a pathway for how neofunctionalized CAL enzymes activate carboxylate substrates and catalyse stereoselective C−C bond formation in plant specialized metabolism.

The identification of these three CALs demonstrates that proteins from the CAH family can participate directly in specialized metabolic pathways. The unexpected functions of the CALs suggest that their fundamental mechanisms of catalysis are probably distinct from archetypical CAHs. It has been well-established that canonical CAHs use an extremely highly conserved histidine triad to coordinate a $Zn^{2+}$ cofactor, which acts as a Lewis acid for generating the reactive hydroxide ion that hydrates $CO_2$ (refs. 43,44). Thus, it is notable that in homologues of both *Pt*CAL-1 and *Pt*CAL-2, this histidine triad has been mutated (Fig. 4b, Extended Data Fig. 6 and Supplementary Fig. 2). In the case of *Pt*CAL-1, two of the three histidines are mutated, whereas all three are mutated in *Pt*CAL-2. Previous analysis of analogous mutations in CAHs have determined that perturbation of this triad leads to a loss in $Zn^{2+}$-binding and CAH activity[45,46] and thus the mutations observed in *Pt*CAL-1 and *Pt*CAL-2 would seem to indicate a different mechanism of catalysis. Although *Pt*CAL-3 retains this histidine triad, several other highly conserved active site residues involved in substrate binding have been altered (Fig. 4b and Supplementary Fig. 2), presumably to accommodate the increase in substrate size relative to $CO_2$/bicarbonate. For each CAL, the addition of a Zn-chelating reagent[44] to the apoplastic protein did not lead to any discernable loss in their biosynthetic activity, nor was there any effect from the supplementation of Zn to these reactions (Extended Data Figs. 4i and 5n). This is in contrast to the effect of Zn chelators on canonical CAH enzymes, which typically show near complete loss of activity after such treatments[47]. Although this suggests that the CALs no longer use Zn as a cofactor, more comprehensive examination of these proteins will be needed to understand their cofactor requirements as well as the fundamental mechanisms of their catalysis. Structural modelling[48] of the three CALs demonstrates that they exhibit the conserved tertiary structure found in the CAH family[37] (Extended Data Fig. 6). With that considered, we expect that the observed alterations in highly conserved active site residues in these CALs will provide a prominent starting point for future mechanistic studies.

Beyond understanding the detailed catalytic mechanisms of the CALs, further work will be necessary to establish the reason(s) for *Pt*CAL-1/*Pt*CAL-2 codependence. Although computational modelling[48] predicts *Pt*CAL-1a and *Pt*CAL-2a to interact with a moderate amount of confidence (Supplementary Fig. 3), de novo prediction of protein heterodimers remains challenging without experimental validation. Thus, it will be necessary in future work to rigorously assess potential interaction between these two proteins, as well as how this interaction may affect function. For example, although we have shown that the co-expression of *Pt*CAL-1 critically affects the localization and post-translational modification of *Pt*CAL-2, it is not yet clear how *Pt*CAL-1 may cause this change and more questions remain as to how these proteins may be cooperating to carry out phlegmarane scaffold formation. Thus, these CALs will provide an exciting model not only for investigating the catalytic mechanisms of a neofunctionalized subclass of enzymes but also for understanding the nuanced roles for transport and protein cooperativity in specialized metabolism.

## Enzymatic tailoring for the production of neuroactive HupA

Although we were not immediately successful in finding enzymes that could process **9**, we next sought to investigate further downstream reactions in Lycopodium alkaloid metabolism. In our previous study of HupA (**17**) biosynthesis, we identified three 2OGDs (*Pt*2OGD-1, *Pt*2OGD-2 and *Pt*2OGD-3) which function in the downstream tailoring reactions

required to produce **17** from proposed precursors[23]. However, we were initially unable to identify an enzyme that could act on these substrates to form the 8,15-double bond (see Fig. 1a for numbering) that is present in **17** and many other Lycopodium alkaloids, suggesting that we had not been testing the correct substrate(s). The simplest Lycopodium alkaloid with the same 'lycodane' scaffold (Fig. 1a) as **17** is flabellidine (**10**)[49], which contains an N-acetyl group on the A-ring nitrogen and could plausibly be derived from **9** (Fig. 5). Milligram quantities of this molecule had previously been purified[50], which allowed us to test this as a substrate in *N. benthamiana* leaves expressing our oxidase gene candidates from cluster131 (CYPs and 2OGDs). Through this approach, we identified a pair of 2OGD enzymes which acted sequentially to convert **10** into downstream, oxidized products (Supplementary Results give a detailed description of these enzymes). The first of these enzymes (*Pt*2OGD-4) oxidized **10** to a molecule with an exact mass that is consistent with the installation of a carbonyl (proposed structure **11**, $[M + H]^+ = m/z$ 303.2067) (Extended Data Fig. 7), whereas the second enzyme (*Pt*2OGD-5) consumed **11** and produced a desaturated compound (proposed structure **13**, $[M + H]^+ = m/z$ 301.1911) (Extended Data Fig. 8). Although authentic standards were not available for these compounds, we suspected that *Pt*2OGD-4 was catalysing formation of the A-ring carbonyl, whereas *Pt*2OGD-5 was installing the 8,15-double bond.

If our predictions for the oxidations catalysed by *Pt*2OGD-4 and *Pt*2OGD-5 were correct, then the only remaining oxidation would be A-ring desaturation, which we have shown to be catalysed by *Pt*2OGD-3 (ref. 23). However, *Pt*2OGD-3 did not consume **13** and thus we hypothesized that N-deacetylation must precede this desaturation. Accordingly, we found an α/β hydrolase family enzyme (*Pt*ABH-1) in cluster131 that consumed **13** to produce the N-deacetylated compound lycophlegmarinine D (**14**)[51], which verified the positioning of the carbonyl and double bond installed by *Pt*2OGD-4 and *Pt*2OGD-5, respectively (Extended Data Fig. 9). Addition of *Pt*2OGD-3 to the transiently co-expressed combination of *Pt*2OGD-4, *Pt*2OGD-5 and *Pt*ABH-1 led to the consumption of **14** and the formation of huperzine B (**15**) (Extended Data Fig. 10a and Supplementary Fig. 4). Subsequent addition of *Pt*2OGD-1 and *Pt*2OGD-2 allowed for the production of huperzine C (**16**) and, ultimately, **17** (Extended Data Fig. 10a and Supplementary Fig. 5), thus establishing a biosynthetic route for the complete, stepwise biosynthesis of **17** from **10**. Although **17** has generated the most interest as a potential pharmaceutical[14], hundreds of Lycopodium alkaloids have been isolated and structurally characterized[13], including many congeners of **17** pathway intermediates which differ in their degree of unsaturation. Indeed, by mixing and matching enzymes from this downstream biosynthetic module, we were able to reconstitute the biosynthesis of 15 different Lycopodium alkaloids from **10** as an initial substrate. This included nine previously isolated and characterized compounds which were verified with authentic standards, as well as six previously unreported alkaloids (Fig. 5, Extended Data Fig. 10a−d and Supplementary Fig. 5; Supplementary Results gives more details of these experiments). This demonstrates that the enzymes we identified contribute to a metabolic network of Lycopodium alkaloids in the endogenous plants, thereby explaining much of the structural diversity found among this class of alkaloids.

The biological functions for most Lycopodium alkaloids in the native plants have not been determined but the ability of many of these compounds to inhibit AChE, a critical enzyme in animal neuronal signalling, suggests that they may act to deter herbivory through this mechanism. In support of this, AChE is a common target of insecticides[7] and **17** has been shown to exhibit antifeedant activity on several insect species[52], suggesting a possible AChE inhibition mechanism for **17** in the defence of the plant against insect herbivory. It is notable that **17** exhibits the most potent AChE inhibitory activity of any Lycopodium alkaloid measured thus far and that this inhibition activity decreases with each previous intermediate in the pathway (Fig. 5). This seems to represent a metabolic structure–activity relationship among the

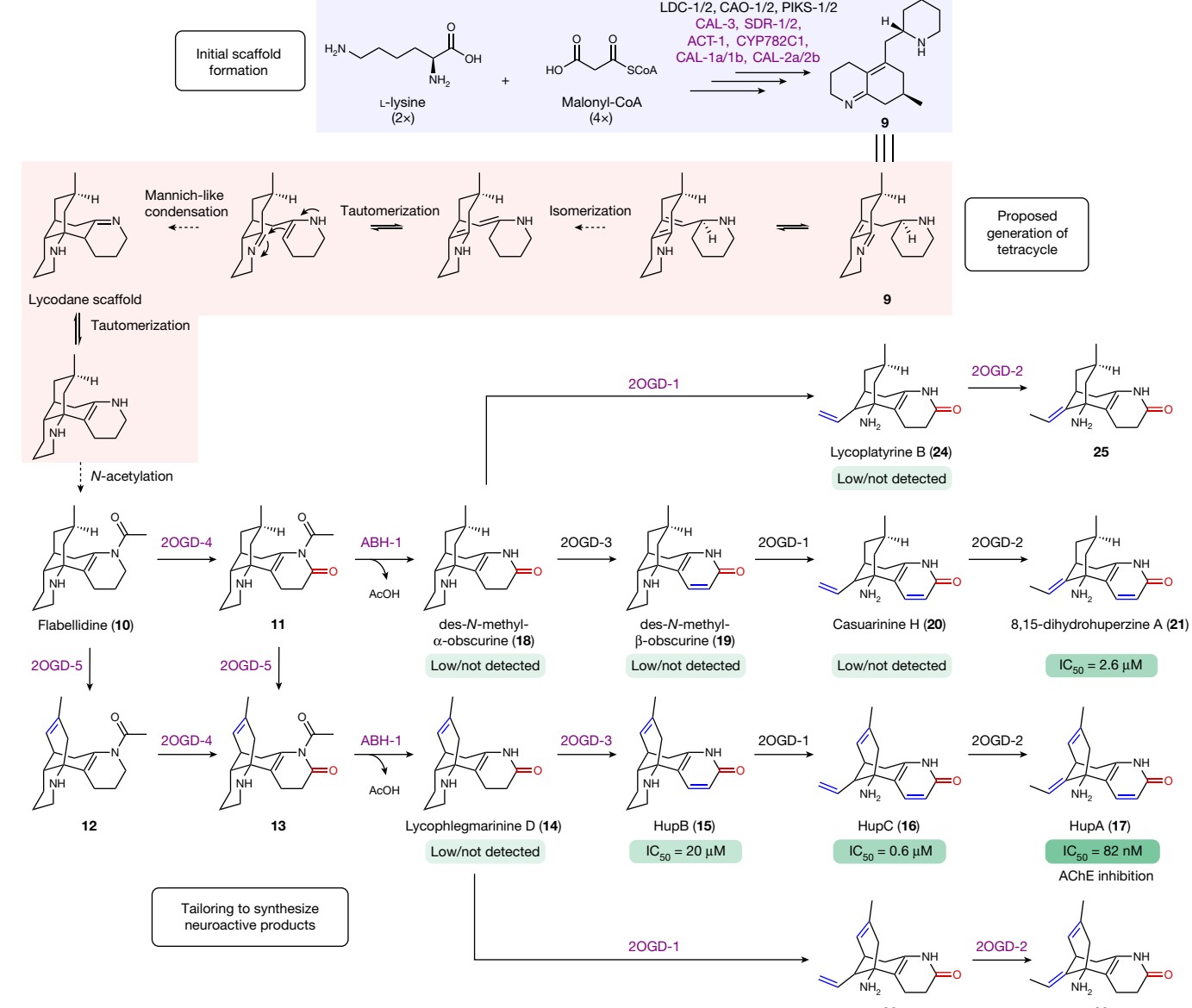

**Fig. 5 | A metabolic network for the generation of an optimized AChE inhibitor, HupA (17).** New enzymes or new reactions for previously described enzymes, are coloured purple. Any Lycopodium alkaloids with common names have been verified with authentic standards. Shown below Lycopodium alkaloids are their IC$_{50}$ values for the inhibition of AChE, if previously tested. Citations for these values can be found in the Methods. Note that the stereochemistry of the methyl group in **9** is predicted on the basis of the typical stereochemistry observed in isolated Lycopodium alkaloids[53].

Lycopodium alkaloids, wherein each of the enzymatic transformations en route to **17** enhances AChE inhibitory activity. Although we cannot be sure that the biological function of **17** is to inhibit animal AChE enzymes, the relationship between Lycopodium alkaloid biosynthesis and AChE inhibitory activity suggests that this metabolic pathway has evolved successive biosynthetic steps that increase the potency of these alkaloids step-by-step to achieve the production of an 'optimized' AChE inhibitor. However, we note that alternative explanations for the evolution of **17** biosynthesis are plausible, particularly given the complex, metabolic network of Lycopodium alkaloids that exists in extant plants. For example, it is possible that **17** was a minor component of the Lycopodium alkaloid cocktail present in a shared common ancestor and that the AChE activity of **17** was selected for, thereby refining and enhancing the biosynthetic production of this molecule. Regardless of the specific mechanism, the Lycopodium alkaloids could prove to be a powerful system for understanding the evolution of specialized metabolism in early diverging plants.

In support of our proposed biosynthetic pathway, all main biosynthetic intermediates from **4** to **9** (Supplementary Fig. 6) and **10** to **17** (Supplementary Fig. 7) could be detected in extracts from tissues in *P. tetrastichus* in which **17** biosynthesis actively occurs. Transformation of the phlegmarane scaffold of **9** into the tetracyclic lycodane scaffold found in downstream alkaloids would putatively only require a double-bond isomerization and enamine–imine condensation (Fig. 5). Final N-acetylation of this scaffold on the A-ring would then yield **10**, thereby connecting upstream biosynthesis to the downstream transformations required to produce **17**. The identification of **10** as a precursor to **17** sheds critical light on the tentative chemical logical of this final tetracyclic scaffold formation. In particular, the addition of the N-acetyl group to the A-ring probably serves as a protecting group that 'locks' the tetracyclic lycodane scaffold in place, which would otherwise be in equilibrium with the enamine/imine (Fig. 5). In agreement with this premise, the N-acetyl group is only lost following formation of the A-ring lactam by *Pt*2OGD-4, which would also serve to deactivate

the basicity of the nitrogen and protect the stability of this tetracyclic ring structure. Although we do not know the nature of the enzyme(s) required to convert **9** into the theoretical enamine/imine intermediate, we can be confident that an acetyltransferase family enzyme is required for the final step to yield **10**.

Our efforts in identifying new enzymes in **17** biosynthesis (Fig. 5) provide fundamental insight into the previously cryptic reactions used to build and tailor the scaffold structures of neuroactive Lycopodium alkaloids and greatly expand our broader understanding of the enzymatic capabilities present in the plant kingdom. Most notably, our identification of several, neofunctionalized CAH family enzymes suggests that proteins from this family may have more widespread roles throughout plant metabolism than previously realized. Ultimately, our results place CAL proteins among a relatively short list of enzymes known in plants for the biosynthesis of specialized metabolite scaffolds[2,3].

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

## Methods

### Chemicals and reagents

All common chemicals and reagents were obtained from commercial vendors. A mixture of 1-(piperidin-2-yl)propan-2-ol stereoisomers (**5**) was obtained commercially (MilliporeSigma). Authentic standards of (2*S*,4*S*)-**5** (otherwise known as (+)-sedridine) and (2*R*,4*S*)-**5** (otherwise known as (−)-allosedridine) were provided by P. Evans (University College Dublin). An authentic standard of lycophlegmarinine D (**14**) isolated from *Phlegmariurus phlegmaria*[51] was provided by K. Pan (China Pharmaceutical University) and 8,15-dihydrohuperzine (**21**) was provided by R. Sarpong (University of California, Berkeley)[17]. The following Lycopodium alkaloids were previously isolated from *Lycopodium platyrhizoma*[50]: flabellidine (**10**), des-*N*-methyl-α-obscurine (**18**), des-*N*-methyl-β-obscurine (**19**), casuarinine H (**20**) and lycoplatyrine B (**24**). Confirmatory NMR spectra for **10** and **20** can be found in the Supplementary Information; those of **18** and **19** were previously reported[23]. The following Lycopodium alkaloids were purchased from commercial vendors: huperzine B (**15**, MilliporeSigma), huperzine C (**16**; two independent sources: Shanghai Tauto Biotech and Toronto Research Chemicals) and HupA (**17**, ApexBio Technology).

### Transcriptomic and co-expression analysis

Transcriptomic data of *P. tetrastichus* were previously generated using PacBio IsoSeq for establishing a high-quality reference transcriptome of full-length sequences and Illumina HiSeq 4000 for quantification of gene expression across many tissue types and biological samples[23]. Protein sequences encoded by each transcript were annotated with the best-hit Pfam term[54] using HMMER (http://hmmer.org/). We performed differential expression analysis between samples from new growth leaves (biosynthetically active for HupA production) and mature shoot tissue (inactive for HupA production) using edgeR (ref. 55). This analysis yielded 2,227 unique transcripts that had significantly higher expression in the new growth leaves. These transcripts were then included in hierarchical clustering analysis using Cluster 3.0 (ref. 56). For this, expression counts (trimmed mean of *M*-values (TMM)-normalized, c.p.m.) for each transcript were normalized to the median expression value for that transcript and these values were then $\log_2$-transformed. Transcripts were then hierarchically clustered using the Pearson correlation (centred) metric with average linkage and visualized in TreeView software (https://jtreeview.sourceforge.net/). Relevant clusters were identified on the basis of the presence of previously characterized genes from Lycopodium alkaloid biosynthesis (*Pt*LDC-1, *Pt*LDC-2, *Pt*CAO-1, *Pt*CAO-2, *Pt*PIKS-1, *Pt*PIKS-2, *Pt*2OGD-1, *Pt*2OGD-2 and *Pt*2OGD-3). This allowed for the identification of a minimally sized cluster of 131 transcripts that contained all previously characterized transcripts. Specific clusters of transcripts (cluster131 and cluster273) referenced are given in the Supplementary Information.

### *Agrobacterium*-mediated transient expression

Candidate genes were cloned using complementary DNA from *P. tetrastichus* new growth leaves, much as previously described[23]. Following PCR amplification with primers containing appropriate overhangs, PCR products were gel purified and inserted into previously digested (AgeI/XhoI) pEAQ-HT plasmid (Kan^R) using isothermal DNA assembly. Assembled plasmid reactions were transformed into *E. coli* NEB 10-beta cells (New England Biolabs) and plated on selective LB agar plates (50 µg ml⁻¹ of kanamycin) for overnight growth at 37 °C. Colonies were screened using PCR and the sequences of PCR products were confirmed using Sanger sequencing. Positive transformants were then used to inoculate 4 ml of liquid LB cultures, which were then shaken overnight at 37 °C. Plasmids were subsequently purified through miniprep and inserts were again sequence verified using Sanger sequencing. Plasmids containing genes of interest were transformed into *Agrobacterium tumefaciens* GV3101 (Gent^R) using the freeze-thaw method, plated onto selective LB agar plates (50 µg ml⁻¹ of kanamycin and 30 µg ml⁻¹ of gentamycin) and grown for 2 days at 30 °C. Positive transformants were verified through colony PCR and these were then inoculated into 2 ml of liquid LB cultures, which were shaken for 2 days at 30 °C. Colony PCR was again used to verify the presence of the plasmid construct in the liquid cultures, after which 25% glycerol stocks were prepared and stored at −80 °C for future use.

Screening of candidate genes through *Agrobacterium*-mediated transformation in *N. benthamiana* was performed much as previously described[23,57]. *Agrobacterium* strains harbouring plasmid constructs of interest were first thickly streaked from glycerol stocks onto LB agar plates (50 µg ml⁻¹ of kanamycin and 30 µg ml⁻¹ of gentamycin) and grown for 2 days at 30 °C. This lawn of cell growth was then removed using a sterile pipette tip, resuspended in 0.5 ml of LB and then pelleted through centrifugation at 8,000*g* for 5 min. Cells were then resuspended in 0.5 ml of *Agrobacterium* induction media (10 mM MES, 10 mM MgCl₂, 150 µM acetosyringone, pH 5.6) and allowed to incubate at room temperature for at least 1 h. The concentrations of cell resuspensions were measured by taking their optical density $OD_{600}$ and combinations of strains of interest were then combined at a final $OD_{600}$ of 0.2–0.3 for each strain. A needleless syringe was then used to infiltrate these strain mixtures into the abaxial side of *N. benthamiana* leaves from 4–5-week-old plants, which were germinated and grown exactly as previously described[23,57]. For a typical experiment, three leaves from three different plants were used for each strain mixture to minimize any batch effects or biological variation among plants. Following infiltration, plants were grown as usual for 3–5 days, after which leaves were excised for subsequent metabolite extraction. For substrate co-infiltration experiments, plants were grown for 3 days after *Agrobacterium* infiltration, after which 100 µl of substrate (25 µM in water) was infiltrated into the infected portion of the leaf using a needleless syringe. The area infiltrated with substrate was marked and after one more day of plant growth, this area was excised for subsequent metabolite analysis.

### Metabolite extraction

Following transient gene expression, *Agrobacterium*-infected leaf tissue was excised, placed in a preweighed 2 ml Safe-Lock tube (Eppendorf) and immediately snap frozen in liquid nitrogen. Typically, only one-quarter of a leaf was excised for analysis. When substrate was co-infiltrated, the entire marked area of substrate infiltration was excised and snap frozen. Snap-frozen samples were either stored at −80 °C or immediately lyophilized to dryness. Following lyophilizing, samples were kept on ice or at 4 °C during all stages of processing. After removal from the lyophilizer, samples were weighed to collect dry masses. A 5 mm diameter steel bead was then added to each sample tube and plant tissue was homogenized to a powder by shaking at 25 Hz for 2 min on a ball mill homogenizer (Retsch MM 400). Steel beads were removed with tweezers and homogenized tissue was extracted with an appropriate volume of solvent. For routine extraction, 80% methanol in water was added at an amount of 20 µl of solvent per milligram of dry leaf weight and, after mixing, samples were incubated on ice for at least 20 min. During the course of our experiments, we noted that certain intermediates (for example, **3**, **7** and **8**) would be depleted over time, either because of decomposition or reactivity with other metabolites from *N. benthamiana*. We found that extracting samples with ice-cold water + 0.1% (v/v) formic acid would improve the stability of these compounds without any major losses in alkaloid yield. As such, most of the LC–MS chromatograms that are shown for early pathway intermediates were derived from experiments in which water + 0.1% formic acid was used as the extraction solvent.

After incubation, samples were briefly vortexed and cell debris was pelleted through centrifugation at 10,000*g* and 4 °C for 5 min. After centrifugation, samples were prepared differently on the basis of the type of chromatographic analysis that was to be used (for example,

C18 versus hydrophilic interaction chromatography (HILIC)). Samples related to the analysis of the early biosynthetic pathway (that is, any of the products generated by *Pt*LDC-1/2, *Pt*CAO-1/2, *Pt*PIKS-1/2, *Pt*SDR-1/2, *Pt*ACT-1, *Pt*CYP782C1, *Pt*CAL-1/*Pt*CAL-2 and *Pt*CAL-3) were diluted tenfold in ice-cold acetonitrile (ACN) to better match the starting solvent conditions for HILIC analysis. Samples related to the analysis of downstream intermediates (that is, any intermediates downstream of **10**) were diluted 1:1 with water + 0.1% formic acid. All samples were then filtered through Multiscreen Solvinert filter plates (MilliporeSigma, Hydrophilic PTFE, 0.45 µm pore size) and subsequently transferred into LC–MS vials, which were stored at −20 or −80 °C until analysis.

### Preparation of metabolites for chiral analysis

Many of the early intermediates could only be observed by HILIC analysis, which made it difficult to resolve enantiomers with standard chiral chromatography. Protection of the secondary amines of **4** and its pathway derivatives through N-acetylation allowed us to readily separate enantiomers (Extended Data Fig. 1c). The N-acetylation of standards, plant extracts and enzyme reactions was performed as follows. A 10 µl aliquot of sample was diluted into 90 µl of ACN (for standards, 10 µl of a 10 mM stock solution in methanol was used) and 200 µl of acetic anhydride was then added. Samples were then heated at 60 °C for 30 min, although we noted that heating was not strictly necessary for N-acetylation to readily occur. After this incubation, samples were moved onto ice for at least 5 min, after which 300 µl of methanol was added to quench the reaction. Quenched samples were then filtered and transferred into LC–MS vials, as described above. Standards were subsequently diluted to a concentration of 10–20 µM in 80% methanol before analysis.

### LC–MS analysis

Samples were routinely analysed on two different LC–MS instrument setups: (1) an Agilent 1260 high-performance liquid chromatography (HPLC) instrument paired with an Agilent 6520 accurate-mass quadrupole time-of-flight (Q-TOF) mass spectrometer (6520 LC–MS) or (2) an Agilent 1290 Infinity II UHPLC paired with a coupled Agilent 6546 Q-TOF mass spectrometer (6546 LC–MS). For both instruments, all samples were analysed using electrospray ionization (ESI) in positive ionization mode. Each instrument also had an in-line diode array detector (DAD) for routine analysis of UV active compounds (Agilent 1100 DAD for 6520 LC–MS; Agilent 1290 Infinity II DAD for 6546 LC–MS). UV data were typically collected at wavelengths of 210, 230, 254 and 280 nm (4 nm bandwidth for each) with reference to 360 nm (100 nm bandwidth). Reversed-phase (C18) analysis was predominantly performed on the 6546 LC–MS using a ZORBAX RRHD Eclipse Plus C18 column (Agilent, 1.8 µm, 2.1 × 50 mm) with water + 0.1% formic acid and ACN + 0.1% formic acid as mobile phases. HILIC analysis was predominantly performed on the 6520 LC–MS using a Poroshell 120 HILIC-Z column (Agilent, 2.7 µm, 2.1 × 100 mm) with water and 9:1 ACN:water, each with 0.1% formic acid and 10 mM ammonium formate, as mobile phases. Chiral chromatography was performed on the 6520 LC–MS using a CHIRALPAK IC-3 column (Daicel, 3 µm, 4.6 × 100 mm) with water + 0.1% formic acid and ACN + 0.1% formic acid as mobile phases. Specific LC–MS method parameters can be found in the Supplementary Methods. In general, early pathways intermediates (compounds **3** through **9**) were observed with HILIC analysis, whereas downstream intermediates (compounds **10** to **25**) were observed with C18 analysis. We note that **6** in particular could be observed using either C18 or HILIC analysis. However, although diastereomers of **6** could be resolved with C18 analysis, these seemed to co-elute as a single peak in HILIC analysis. When applicable, mass ions pertaining to individual metabolites were fragmented using targeted MS². This was normally performed with several collision energies (10, 20 and 40 V) but most of the presented data were collected with a collision energy of 20 V.

LC–MS data were routinely visualized and analysed using MassHunter Qualitative Analysis software. Extracted ion chromatograms shown in each figure were typically generated by extracting for the exact *m/z* for the target ion of interest with a 20 ppm mass tolerance window. Quantification of relative ion abundance was performed using the automated 'Agile2' method in MassHunter Quantitative Analysis software. For untargeted analysis, data files were converted into mzML format and XCMS software[35] was used to identify any differentially produced mass ions between different gene expression conditions/reactions. This output was typically filtered to remove low-abundance ions (less than $1 \times 10^5$ ion abundance) and any ions that were not clearly differential between treatments ($P > 0.2$). XCMS analysis was typically followed with CAMERA software[36] analysis to identify potential in-source ion adducts of detected metabolites. UV spectra for **8** and **9** were produced by using the Extract Spectrum function on the corresponding compound peak in MassHunter Qualitative Analysis.

### Apoplast protein isolation

The three CAL proteins identified in this study have predicted N-terminal signal peptides, which were identified using the TargetP-2.0 server (https://services.healthtech.dtu.dk/service.php?TargetP-2.0)[58]. Preliminary confocal microscopy of C-terminal, GFP-tagged proteins did not support their main localization to be the endoplasmic reticulum or Golgi and initial analysis of images suggested that they may be localized to the apoplast. To assess this possibility, CAL genes with or without C-terminal 6xHis tags were transiently expressed in *N. benthamiana*, as described above. Each CAL gene was expressed individually; also, *Pt*CAL-1a and *Pt*CAL-2a were transiently co-expressed in the same leaf because we had found them to cofunction in Lycopodium alkaloid biosynthesis. At 4 days after *Agrobacterium* infiltration, apoplast protein extracts were isolated using the infiltration–centrifugation method, much as previously described[40]. Two leaves per reaction were excised from the plant and submerged in ice-cold apoplast extraction buffer (100 mM MES, 300 mM NaCl, pH 5.5) in an open-capped 50 ml Falcon tube and these tubes were placed in a plastic vacuum chamber attached to a Welch Model 2025 vacuum pump. The chamber was brought down to full vacuum and after 2 min at this pressure, the vacuum was slowly released to allow for buffer to infiltrate the leaf apoplastic space. Buffer-infiltrated leaves were carefully removed from the Falcon tubes, blotted dry with paper towels and were then rolled into Parafilm and placed in a plungerless 5 ml plastic syringe. The syringe was placed in a 15 ml Falcon tube and this was then centrifuged at 1,000*g* and 4 °C for 10 min to collect apoplast extract. The resulting extract was centrifuged at 10,000*g* and 4 °C for 15 min to pellet any larger cellular debris and the supernatant was concentrated using an Amicon Ultra-4 Centrifugal Filter Unit (10 kDA MWCO, MilliporeSigma UFC501024). Protein concentrations were measured using the BIO-RAD Protein Assay or Bradford assay (Abcam 119216) and adjusted with apoplast extraction buffer to a final concentration between 0.5 and 1.5 mg ml⁻¹. Aliquots of the extracts were snap frozen in liquid nitrogen and stored at −80 °C.

### Western blot analysis of plant extracts

To determine localization of CAL proteins in our *N. benthamiana* transient expression system, we performed western blot analysis of epitope tagged versions of each protein. Each CAL gene was PCR amplified from previously generated plasmid constructs using primers with overhangs for subsequent isothermal assembly into pEAQ-HT plasmid digested at the AgeI/XmaI restriction sites, which creates constructs with a C-terminal 6xHis tag. The reverse primer in this cloning strategy omitted the native stop codon of the CAL coding sequences to ensure that the final coding sequence included the C-terminal tag. These constructs were sequence verified, transformed into *A. tumefaciens* GV3101 and these strains were then used to transiently express these genes in *N. benthamiana*, as described above.

For the analysis of different protein fractions, apoplast extracts were prepared exactly as described above. Once apoplast extracts were obtained, the remaining leaf tissue was flash-frozen in liquid nitrogen and lyophilized to dryness. Lyophilized leaf tissue was pulverized to a powder with 5 mm stainless steel beads in a ball mill homogenizer (Retsch MM400) at 25 Hz for 2 min. Protein from homogenized samples was then extracted with ice-cold phosphate-buffered saline (PBS) supplemented with Halt protease and phosphatase inhibitor cocktail (Thermo Scientific PI78443) using 20 µl of buffer per mg dry leaf mass. This was incubated on ice for 20 min with periodic, gentle inversion, after which samples were centrifuged at 18,210$g$ for 10 min at 4 °C to remove insoluble plant material. The remaining supernatant was kept and represented the 'internal' cell fraction, which would presumably contain cytosolic and microsomal proteins. Protein concentration was determined by Bradford assay (Abcam 119216) and extracts were stored at −80 °C until future use.

Samples for immunoblots were prepared by adding 4× NuPAGE LDS sample buffer (Fisher Scientific AAJ61894AC) to a final concentration of 1× sample buffer with 2.5% β-mercaptoethanol and samples were then heated for 20 min at 70 °C. Total protein for apoplast (2.5 µg) and PBS extracts (5 µg) was separated on NuPAGE gels and then transferred onto a PVDF membrane (BIO-RAD 1704272) using a Trans-Blot semidry transfer system (BIO-RAD). Blots were blocked in EveryBlot blocking buffer (BIO-RAD 12010020) for more than 5 min at room temperature and incubated with mouse anti-His (Genscript A00186) at 0.1 µg ml⁻¹ in EveryBlot buffer for 1 h at room temperature or overnight at 4 °C. After washing three times with PBST (PBS + 0.1% Tween), blots were incubated with horse antimouse IgG, HRP-linked antibody (Cell Signaling Technology 7076) at 1:3,000 dilution. Blots were then washed five times with PBST and imaged with an iBright FL1500 Imaging System (Thermo Fisher Scientific).

## Heterologous expression of CYP782C1 in yeast, microsomal protein preparation and in vitro enzyme assays

Expression of *Pt*CYP782C1 in *Saccharomyces cerevisiae* (yeast) was performed as previously described[57,59]. Briefly, the coding sequence of *Pt*CYP782C1 was PCR amplified and annealed into the pYeDP60 plasmid. This plasmid construct was transformed into *S. cerevisiae* WAT11 (*ade2*) and positive transformants were selected on synthetic drop-out medium plates lacking adenine (6.7 g l⁻¹ of yeast nitrogen base without amino acids, 20 g l⁻¹ of glucose, 2 g l⁻¹ of drop-out mix minus adenine, 20 g l⁻¹ of agar) through growth at 30 °C for 2 days. Presence of the plasmid constructs was confirmed by colony PCR. A single, positive colony was used to inoculate a starter 4 ml of culture of liquid drop-out medium, which was grown at 28 °C and 250 r.p.m. Following 2 days of growth, 2 ml of the starter culture was used to inoculate 500 ml of YPGE medium (10 g l⁻¹ of Bacto yeast extract, 10 g l⁻¹ of Bacto peptone, 5 g l⁻¹ of glucose and 3% (v/v) ethanol). This culture was grown at 28 °C and 250 r.p.m until reaching a cell density of $5 \times 10^7$ cells ml⁻¹, which was estimated through OD$_{600}$ measurements. After reaching this density, expression was induced by adding 50 ml of a sterile galactose solution (200 g l⁻¹) to achieve a concentration of approximately 10% (v/v). The culture was then grown at 28 °C and 250 r.p.m. for another 16 h to achieve a cell density of approximately $5 \times 10^8$ cells ml⁻¹, after which this culture was immediately used for microsomal protein isolation, which was performed exactly as previously described[59]. Microsomal protein was stored in TEG buffer (50 mM Tris-HCl, 1 mM EDTA, 20% (v/v) glycerol, pH 7.4), aliquoted into 1.5 ml microfuge tubes, snap frozen in liquid nitrogen and stored at −80 °C.

Enzyme reactions with *Pt*CYP782C1-enriched microsomal protein were performed in potassium phosphate buffer (50 mM potassium phosphate, 100 mM sodium chloride, pH 7.8) and typically contained 4 µg of microsomal protein (final concentration of 0.02 µg µl⁻¹), 500 µM NADPH and 50 µM of **6** substrate in a total reaction volume of 200 µl. Control reactions omitted NADPH or used microsomal protein that

was heated at 95 °C for at least 10 min. Following addition of all components, reactions were incubated at room temperature for a minimum of 10 min. At specific time points, 20 µl aliquots of the reaction were added to 180 µl of ACN + 0.1% formic acid to quench the reaction. Quenched reactions were then filtered and transferred into LC–MS vials, as previously described. Products of *Pt*CYP782C1 activity on **6** were assessed through LC–MS using HILIC analysis.

## In vitro enzyme reactions with apoplastic CAL protein

Reactions with CAL-enriched apoplast were routinely performed in low-pH potassium phosphate buffer (50 mM potassium phosphate, 100 mM NaCl, pH 5.9) at a volume of 20 µl. For *Pt*CAL-1a/*Pt*CAL-2a, these reactions contained approximately 1.4 µg of apoplast protein for each CAL (final concentration of 0.07 µg µl⁻¹). Control reactions used apoplast from leaves expressing only one CAL or with apoplast generated from GFP-expression *N. benthamiana* leaves. The requisite substrates for this reaction were generated through the activities of in vitro *Pt*PIKS-1 and *Pt*CYP782C1 enzyme reactions. We found that the *Pt*CYP782C1 microsomal reaction did not work well at the lower pH (pH 5–6) at which the CAL enzymes seemed to be most active (Extended Data Figs. 2g and 5k). Therefore, before the CAL reactions, we ran a separate *Pt*CYP782C1 microsomal protein assay in high pH buffer (50 mM potassium phosphate, 100 mM sodium chloride, pH 7.8), much as described above, for a minimum of 2 h to generate sufficient **8** as a substrate. To maximize the amount of **8** produced, substrate-generating *Pt*CYP782C1 reactions (100 µl) contained 1.5 mM of substrate (**6**), 10 µg of *Pt*CYP782C1 microsomes (final concentration of 0.1 µg µl⁻¹) and 4 mM NADPH. After these incubations, a 2 µl aliquot of the *Pt*CYP782C1 reaction (now containing **8**) was added to the *Pt*CAL-1a/*Pt*CAL-2a apoplast enzyme assay setup (20 µl of total reaction volume). To generate **3** and **4** as potential substrates, 1 µg of previously purified *Pt*PIKS-1 enzyme[23] and 150 µM **1** and 300 µM malonyl-CoA were added directly to the CAL reaction mixtures. After thorough mixing, reactions were incubated at room temperature. An alternative route for producing **3** and **4** independently of thioester intermediates was achieved by mixing stocks of **1** (10 mM in water) and **2** (10 mM in water; always prepared fresh to minimize compound decomposition) in equal proportion, followed by incubation at room temp for 1–2 h, as these two substrates can non-enzymatically condense to yield **3** (which can spontaneously decarboxylate to produce **4**). A 2 µl aliquot of this mixture was then added as a component of the *Pt*CAL-1a/*Pt*CAL-2a enzyme reaction (20 µl total reaction volume) in addition to the *Pt*CYP782C1 microsomal reaction mixture. After predesignated incubation times, reactions were quenched by diluting tenfold into ACN with 0.1% formic acid.

For *Pt*CAL-3 activity assays, 5 µg of *Pt*CAL-3 apoplast (final concentration 0.1 µg µl⁻¹ of apoplast protein), was diluted into in low-pH potassium phosphate buffer (50 mM potassium phosphate, 100 mM NaCl, pH 5.9) at a volume of 50 µl just as with *Pt*CAL-1a/*Pt*CAL-2a. To generate **3** and **4** as potential substrates in vitro, 1 µg of previously purified *Pt*PIKS-1 enzyme[23] was added to this reaction (final concentration of 0.02 µg µl⁻¹) with 150 µM **1** and 300 µM malonyl-CoA added as substrates. In follow-up experiments, the PIKS reaction was omitted and **1** and **2** were added as direct substrates to a final concentration of 500 µM each. When **4** was tested as a substrate, it was added at a concentration of 150 µM. All reactions were incubated at room temperature for predesignated amounts of time, after which aliquots were quenched through fivefold dilution in ice-cold ACN. For all CAL apoplast enzyme reactions, product formation was predominantly assessed through LC–MS using HILIC analysis. To assess the formation of specific enantiomers or consumption of specific enantiomeric substrates, quenched reactions were N-acetylated and analysed by chiral LC–MS, as described above.

To evaluate the potential decarboxylation of β-keto acid substrates by *Pt*CAL-1/*Pt*CAL-2 and *Pt*CAL-3, only **3** or **2**, respectively, were added as substrate. *Pt*CAL-1/*Pt*CAL-2 reactions were quenched by diluting

aliquots tenfold into ACN with 0.1% formic acid and were subsequently analysed through HILIC LC–MS. *Pt*CAL-3 reactions were quenched by mixing aliquots with an equal volume of water with 0.2% formic acid and were then analysed through C18 LC–MS. Decarboxylation was assessed by comparing the relative ion abundances of each substrate to that of their decarboxylated product; for **3**, this pertained to **4** and for **2**, this pertained to acetoacetic acid (AcAc). For all reactions, GFP apoplast with relevant substrates was analysed as a negative control. This was critical for relative quantification of decarboxylation, as this can happen readily to both **3** and **2** at room temperature.

For evaluation of $Zn^{2+}$ as a cofactor for *Pt*CAL-1/*Pt*CAL-2 and *Pt*CAL-3 catalytic activity, 0.3 ml of apoplast extract containing the CAL proteins was incubated with 13 ml of physiological pH potassium phosphate buffer (50 mM potassium phosphate, 100 mM NaCl, pH 7.5) containing 10 mM of the $Zn^{2+}$ chelating reagent 2,6-pyridinedicarboxylic acid (PDCA)[44] at 4 °C for 4 h with gentle rocking. PDCA was then diluted out by a factor of $10^8$ through buffer exchange (50 mM potassium phosphate, 100 mM NaCl, pH 5.9) using Amicon Ultra-4 Centrifugal Filter Units (10 kDA MWCO, MilliporeSigma UFC501024). To control for possible loss of activity during this treatment and purification time, separate CAL apoplast extracts were treated and prepared as above but without PDCA. Protein concentrations were measured using the BIO-RAD Protein Assay or Bradford assay (Abcam 119216) and adjusted with potassium phophate buffer to a final concentration between 0.5 and 1.5 mg ml$^{-1}$. Aliquots of the extracts were then snap frozen in liquid nitrogen and stored at −80 °C. Standard in vitro reactions for *Pt*CAL-1/*Pt*CAL-2 and *Pt*CAL-3 were then run as described above to evaluate any effects on product formation. For $Zn^{2+}$ supplementation, a final concentration of 1 mM $ZnCl_2$ was added to reaction mixtures[44].

### Synthesis of 6 stereoisomers
To synthesize **6** stereoisomers, 150 mg of previously synthesized pelletierine (**4**, oil, 1 mmol)[23] was added to 1 ml of methanol in a glass vial with a magnetic stir bar. This mixture was stirred on ice and 0.095 g (2.5 equiv.) of $NaBH_4$ was added slowly. This reaction was allowed to incubate on ice for 2 h. The reaction was quenched through the addition of 2 ml of distilled water followed by 2 ml of 2 M HCl. The pH of the reaction was increased to pH 10 with 6 M NaOH (about 0.3 ml) and this was then extracted with diethyl ether (5 × 5 ml). The organic fractions were pooled, dried with anhydrous sodium sulfate, clarified using a filter and evaporated to dryness using a rotary evaporator system. A portion of this residue, which would be mainly composed of **5** stereoisomers, was then O-acetylated following an established protocol[60]. To accomplish this, 50 mg (0.35 mmol) of the synthesized **5** stereoisomers was dissolved in 100 μl of 6 N HCl in a glass vial. Next, 100 μl of acetic acid was added and this mixture was cooled to about 0 °C in an ice bath. Once this mixture was chilled, 1 ml of acetyl chloride was slowly added dropwise. This reaction was then incubated in the ice bath for 1 h, with periodic, gentle mixing. After this incubation, a 1 μl aliquot of this reaction was diluted in 1 ml of water + 0.1% (v/v) formic acid and this was analysed through C18 LC–MS to confirm the formation of the same acetylated compounds that were produced by *Pt*ACT-1. The full reaction was diluted in 25 ml of ice-cold distilled water, then clarified through filter paper.

The putative **6** stereoisomers were then purified by using a Sep-Pak C18 12 cc, 2 g Vac Cartridge (Waters). To do so, this cartridge was pre-equilibrated with 3 column volumes (CVs) of ACN + 0.1% (v/v) formic acid, followed by equilibration with 4 CVs of water + 0.1% (v/v) formic acid. The reaction mixture was then loaded onto the cartridge and the solvent was allowed to flow through. The loaded cartridge was then washed with 3 CVs of water + 0.1% (v/v) formic acid and the products (visibly yellow on the cartridge) were eluted with 30% ACN in water (with 0.1% v/v formic acid). Small (about 0.5 ml) fractions of the eluent were collected and 1 μl of each were diluted in water + 0.1% (v/v) formic acid and analysed through C18 LC–MS to confirm the

presence of putative **6** diastereomers. Relatively pure fractions were combined, diluted into 20 ml of water + 0.1% (v/v) formic acid and repurified over the same type of cartridge, much as described above. For this second round of purification, ACN in water (+0.1% v/v formic acid) was added as an eluent at incrementally increasing concentrations (1 CV each of 2%, 4%, 6%, 8%, 10%, 20% and 40% ACN). Collected fractions were screened through LC–MS and pure fractions were combined, frozen and lyophilized to dryness. The resulting purified compound (about 20 mg) consisted of a yellowish powder. For structural confirmation, this was dissolved in $CDCl_3$ and we then performed $^1H$ and $^{13}C$ NMR analysis using a Varian Inova 500 MHz NMR spectrometer (Supplementary Figs. 8 and 9).

### Synthesis of enantio-enriched (*R*)- and (*S*)-pelletierine (4)
Enantiomers of **4** were synthesized by following a previously established protocol[61]. To a 25 ml round bottom flask with a magnetic stir bar were added 1-piperideine (**1**, 81 mg, 0.97 mmol, 1 equiv.), acetone (3.26 ml, 44.46 mmol, 46 equiv.), DMSO (3.26 ml), water (0.41 ml) and either D- or L-proline (21.2 mg, 0.19 mmol, 0.2 equiv.). L-proline was used to achieve enantio-enriched (*S*)-**4**, whereas D-proline was used to produce enantio-enriched (*R*)-**4** (ref. 61). The reaction mixtures were stirred for 1 h at room temperature, after which 10 ml of saturated sodium bicarbonate in water was added. This was then extracted twice with 50 ml of dichloromethane. These organic fractions were combined and then extracted with 50 ml of brine. Residual water was removed from the remaining organic extract through the addition of anhydrous magnesium sulfate, after which this extract was clarified through filter paper and dried on a rotary evaporator system. The remaining yellow/brown oil represented the **4** product. Successful reactions were confirmed by N-acetylating a fraction of the product and analysing through chiral LC–MS, as described above. This method resulted in approximately 70% enantiomeric excess for each specified enantiomer.

### Scaled-up production of CAL-1a/CAL-2a enzymatic product
To achieve milligram quantities of the observed product of *Pt*CAL-1a/*Pt*CAL-2a (*m*/*z* 164, **9**), the leaves of 109 *N. benthamiana* plants (410 g fresh weight) were vacuum infiltrated[62] with a combination of *Agrobacterium* strains necessary for engineering the production of this compound (*Pt*LDC-2, *Pt*CAO-1, *Pt*PIKS-1, *Pt*SDR-2, *Pt*ACT-1, *Pt*CYP782C1, *Pt*CAL-1a, *Pt*CAL-2a and *Pt*CAL-3). To prepare sufficient quantities of *Agrobacterium* for this scale, *Agrobacterium* strains harbouring the necessary gene constructs were first streaked on selective LB agar plates (50 μg ml$^{-1}$ of kanamycin and 30 μg ml$^{-1}$ of gentamycin) and grown for 2 days at 30 °C to achieve colonies. Single colonies were then used to inoculate 1 l of liquid LB cultures (50 μg ml$^{-1}$ of kanamycin and 30 μg ml$^{-1}$ of gentamycin), which were shaken overnight at 30 °C and 250 r.p.m. Bacteria were then pelleted through centrifugat at 5,000*g* for 10 min, after which they were resuspended in a minimal volume of *Agrobacterium* induction buffer. Bacterial densities were measured through $OD_{600}$ and strains were mixed together into a 3 l volume of induction buffer such that each strain had a final calculated density of $OD_{600} = 0.2$. This solution was transferred into a plastic beaker and this was placed into a plastic, vacuum desiccator. Each *N. benthamiana* plant was placed upside-down into the *Agrobacterium* mixture, and the desiccator chamber was brought down to vacuum for 2 min using a Welch Model 2025 vacuum pump, which removed air from the leaves. Pressure was then slowly released, which results in *Agrobacterium* solution infiltrating the previous air space of the leaves. This process was repeated for all 109 *N. benthamiana* plants. Infiltrated plants were then grown as usual for 6 days, after which they were collected and stored at −80 °C until compound purification.

To extract metabolites, frozen plant samples were homogenized in a blender along with 1.5 l of 100% ethanol. This extract was incubated overnight at room temperature in a 4 l flask, after which plant material was removed through clarification over filter paper (this was repeated

twice to remove particulates). This ethanol extract was then dried on a rotary evaporator with gentle heating from a water bath (around 30 °C), after which about 50 ml of water still remained. This was resuspended in 400 ml of 3% tartaric acid in water (w/v) and then extracted with 3× 200 ml ethyl acetate to remove hydrophobic compounds. The pH of the aqueous extract was then increased to pH 8–9 using sodium bicarbonate and this was then extracted with 3× 200 ml ethyl acetate. LC–MS screening of extracts demonstrated that almost none of the Lycopodium alkaloid-related intermediates were extracted from the aqueous phase at pH 8–9; instead, this fraction largely contained nicotine-related alkaloids that are native to *N. benthamiana* metabolism. The aqueous phase was then basified to pH 10–11 using 6 M NaOH and this was extracted with 3× 400 ml ethyl acetate. LC–MS screening confirmed that nearly all of the Lycopodium alkaloid intermediates, including our desired compound (proposed **9**), could be found in this organic extract. These ethyl acetate fractions were combined, dried with anhydrous magnesium sulfate, clarified through filter paper and then evaporated to dryness using a rotary evaporator. The remaining residue (about 170 mg) consisted mainly of a yellow/brown oil. This was redissolved in 20 ml of ethyl acetate and this was filtered to remove any insoluble components and then evaporated to dryness. This residue was then resuspended in a minimal volume of 50:50 hexanes/ethyl acetate (about 5 ml) and was purified using a Biotage Selekt Flash Purification System with a Biotage Sfär KP-Amino D column (50 μm particle size, 5 g volume). Purification conditions consisted of an initial isocratic elution of 100% hexanes/0% ethyl acetate for 3 CVs, followed by a gradient from 100% hexanes/0% ethyl acetate to 0% hexanes/100% ethyl acetate over 10 CVs, with a final 5 CVs at 0% hexanes/100% ethyl acetate. All fractions were collected in 10 ml increments. Each fraction was then screened for **9** through LC–MS with HILIC conditions. This purification strategy allowed for partial purification of our compound. Fractions containing **9** were combined, evaporated to dryness and subjected to the same purification workflow several times (with smaller fraction sizes) to achieve pure **9**. All other fractions, which contained other Lycopodium alkaloid-related compounds, were dried and saved at 4 °C for future use.

We found that our isolated compound (predicted **9**) was relatively unstable; resuspension of this compound in either deuterated chloroform ($CDCl_3$) or deuterated methanol ($CD_3OD$) and analysis through $^1H$ NMR demonstrated loss of indicative chemical shifts over time, although this did allow us to obtain a crude $^1H$ NMR ($CDCl_3$, 500 MHz) for this compound (Supplementary Figs. 10 and 11). Also, on drying of our sample from $CDCl_3$, we noted a colour change from yellow/brown to red. Loss of our compound was confirmed through LC–MS analysis. However, we observed that during the course of purification, a compound pertaining to an oxidation of **9** (*m/z* 263.2118; equal to **9** + oxygen) accumulated to high concentrations. This compound (**9′**) showed a similar LC–MS retention time to **9** and had an $MS^2$ fragmentation pattern that seemed to indicate a phlegmarane-like scaffold structure, which suggested that it may be an oxidized byproduct of **9** (Extended Data Fig. 3o–q). As such, we purified this compound using the same strategy outlined above (yield of about 3 mg) and determined a putative structure (proposed **9′**) through NMR analysis. For **9′**, deuterated ACN ($CD_3CN$) was used as a solvent and spectra were collected on a Varian Inova 600 MHz NMR spectrometer at room temperature (Supplementary Figs. 12–18). Although we were not able to resolve the stereochemistry of the C16 methyl with our NMR analyses, nearly all isolated Lycopodium alkaloids, including those with the phlegmarane scaffold, exhibit *R* stereochemistry at this location[53] and thus, we tentatively predict this same *R* stereochemistry for the C16 methyl of **9′** and thus **9**.

## Sequence analysis and structural modelling of CAL genes and proteins

All analyses of CAL genes and proteins were performed in Geneious (v.2019.2). To generate protein alignments of CAH family proteins, an assortment of protein sequences containing the CAH domain were downloaded from UniProt (https://www.uniprot.org/). Most of these downloaded proteins were selected from plant species (these were selected pseudorandomly to capture a breadth of phylogenetic diversity) and we included all CAHs from plants that, to our knowledge, have been biochemically verified to have canonical CAH activity. We also included sequences from animals, fungi, algae and bacteria, including several proteins that have been biochemically verified to have canonical activity. The human CA2 protein (otherwise known as CAII, hCA II; UniProt ID: P00918) was used as a reference for amino acid numbering in alignments, as this is probably the most rigorously studied CAH protein[37]. The downloaded CAH proteins and the CAH family proteins identified in our transcriptomic dataset (for a set of 80 proteins total) were aligned using the MUSCLE algorithm and phylogenetic trees were constructed using the neighbour-joining method (100 bootstraps) with the Jukes–Cantor genetic distance model. The trees shown in Fig. 4b and Supplementary Fig. 2 have been transformed to align all sequence names. Shown adjacent to each tree in these figures are the amino acid sequences that align to well-defined active site residues in human CA2. Any changes to these residues are indicated in the figure and are colour-coded by amino acid.

The structures of *Pt*CAL-1a, *Pt*CAL-2a and *Pt*CAL-3 were modelled using AlphaFold2 through ColabFold (v.1.5.2)[48]. Each of these proteins is predicted to have an N-terminal signal peptide[58], which would be cleaved during processing and trafficking of these proteins, so structural models were generated with the predicted signal peptide removed (21 amino acid truncation for *Pt*CAL-1a, 23 amino acid truncation for *Pt*CAL-2a, 32 amino acid truncation for *Pt*CAL-3). The highest confidence models are shown in Extended Data Fig. 6. We also used AlphaFold-Multimer through ColabFold[48] to explore possible protein–protein interactions between *Pt*CAL-1a and *Pt*CAL-2a, given that these proteins must be co-expressed in *N. benthamiana* leaves to obtain biochemically active protein extracts. These data provide modest but not definitive support for the formation of a protein heterocomplex and the predicted aligned error plots for the top five ranked heterodimers, as well as the structural model for the top-ranked prediction, are shown in Supplementary Fig. 3.

## $IC_{50}$ values for AChE inhibition by lycodane-type Lycopodium alkaloids

Previous work has determined the ability of various Lycopodium alkaloids to inhibit AChE. A selection of these results are compiled and listed in Fig. 5. References for each of the $IC_{50}$ values for each of the compounds are cited as follows: lycophlegmarinine D (**14**)[51], huperzine B (**15**)[63], huperzine C (**16**)[63], HupA (**17**)[64], des-*N*-methyl-α-obscurine (**18**)[65], des-*N*-methyl-β-obscurine (**19**)[66], casuarinine H (**20**)[67], 8,15-dihydrohuperzine A (**21**)[68] and lycoplatyrine B (**24**)[50]. Compounds annotated as 'low/not detected' were not found to have AChE inhibition in the detectable range of each experiment in question (typically, $IC_{50}$ values in these experiments were not measurable or were greater than 30 μM).

## General statistical analysis

All statistical analyses in this manuscript represent measurements from distinct biological samples, not repeat measurements. No statistical methods were used to predetermine sample sizes, and in general, three replicates were used in each experiment, unless stated otherwise. For experiments involving transient gene expression in *N. benthamiana*, triplicates were spread across three different plants to minimize any biological batch effects inherent to individual plants. All bar graphs shown in the manuscript represent the mean and error bars represent standard deviation from the mean. Essentially all experimental results reported in this manuscript were confirmed through at least two independent experiments and, in most cases, in more than three

independent experiments. Blinding was not used during data collection and analysis, and randomization was not used in experimental design.

## General software use and graph generation

Routine data compilation was performed in Microsoft Excel 2016. General analysis of LC–MS data was performed with Agilent MassHunter Qualitative Analysis 10.0. Chromatograms and mass spectra were plotted using IGOR Pro 6.0. Bar graphs and line graphs were plotted using GraphPad Prism 9 and this software was also used for routine statistical analysis. We performed hierarchical clustering analysis using Cluster 3.0 (ref. 56). R (v.4.2.2) was used for bar graph generation, visualization of hierarchical clustering data and for performing XCMS analysis[69]. Geneious Prime (v.2019.2.3) was used for bioinformatic analyses of nucleic acid and protein sequences. This software was also used for several sequence alignments (MUSCLE algorithm) and phylogenetic tree generation (Jukes–Cantor genetic distance model, neighbour-joining tree build method). The TargetP-2.0 server (https://services.healthtech. dtu.dk/service.php?TargetP-2.0)[58] was used for predicting signal peptides and protein localization. MNova (v.1.6) was used for visualization and processing of NMR data. ChemDraw Professional (v.21.0.0.28) was used for chemical structure visualization and analysis. Structural modelling was performed using AlphaFold-Multimer through Colab-Fold (v.1.5.2) and protein models were visualized in PyMol (v.2.5.4).

## Reporting summary

Further information on research design is available in the Nature Portfolio Reporting Summary linked to this article.

## Data availability

All data in this manuscript are available on request. The raw RNA-seq data analysed in this manuscript have previously been deposited to the NCBI Sequence Read Archive (BioProject PRJNA731132)[23]. Gene sequences for enzymes characterized in this study are deposited in the National Center for Biotechnology (NCBI) GenBank under the following accessions: *Pt*2OGD-4 (OR538095), *Pt*2OGD-5 (OR538096), *Pt*ABH-1 (OR538097), *Pt*ACT-1 (OR538098), *Pt*CAL-1a (OR538099), *Pt*CAL-1b (OR538100), *Pt*CAL-2a (OR538101), *Pt*CAL-2b (OR538102), *Pt*CAL-3 (OR538103), *Pt*CYP782C1 (four homologues; OR538104, OR538105, OR538106, OR538107), *Pt*SDR-1 (OR538108) and *Pt*SDR-2 (OR538109). The UniProt database (https://www.uniprot.org/) was used for identifying and obtaining CAH family sequences that were used in phylogenetic analyses. The human CA2 protein structure (2vva) was acquired from PDB (https://www.rcsb.org/). Any materials generated in this manuscript will be made available, as possible. Source data are provided with this paper.

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

**Acknowledgements** We thank F. Schroeder (Cornell University) and J. Liu (Stanford University) for assistance and useful discussion related to NMR analysis. We also thank D. Nelson (University of Tennessee) for providing cytochrome P450 nomenclature. Thank you to K. Pan (China Pharmaceutical University), R. Sarpong (University of California, Berkeley) and P. Evans (University College Dublin) for providing us with authentic standards. We acknowledge G. Lomonossoff for providing us with the pEAQ-HT plasmid. The research in this manuscript was supported by NIH R01 GM121527 to E.S.S. and NIH R35 GM150908 to R.S.N. Also, R.S.N. was supported as a Howard Hughes Medical Institute Fellow of the Life Sciences Research Foundation. We dedicate this paper to the memory of our late mentor and friend C. T. Walsh.

**Author contributions** R.S.N. and E.S.S. led the project and conceived experimental procedures. R.S.N. performed transcriptomic analysis, cloned candidate genes, characterized enzyme function through transient expression in *N. benthamiana* and in vitro enzyme assays, synthesized chemical substrates, isolated chemical intermediates, carried out structural analysis of small molecules and performed bioinformatic analyses. Y.D. performed in vitro biochemical assays characterizing CAL proteins. C.T. analysed heterologous production of CAL proteins through western blots. D.P. assisted with in vitro enzyme characterization of CALs and analysis of their heterologous production. J.M.G. analysed heterologous production and biochemical properties of CAL proteins. Y.Y.L. isolated and structurally verified Lycopodium alkaloid standards. R.S.N. and E.S.S. wrote the manuscript. All authors contributed to data analysis and presentation.

**Competing interests** The authors declare no competing interests.

**Additional information**
**Correspondence and requests for materials** should be addressed to Ryan S. Nett or Elizabeth S. Sattely.

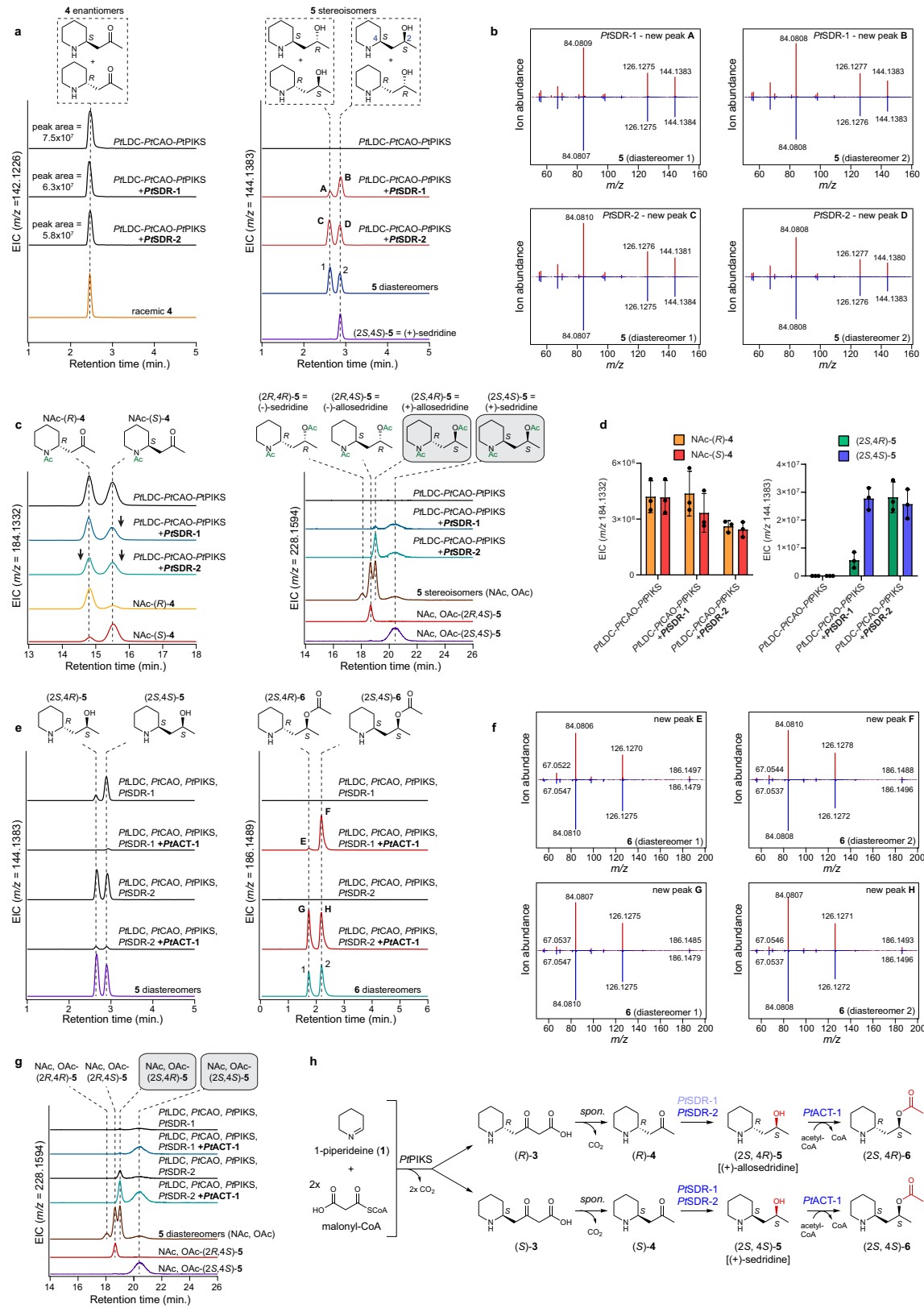

**Extended Data Fig. 1** | See next page for caption.

**Extended Data Fig. 1 | Functional characterization of *Pt*SDR-1, *Pt*SDR-2 and *Pt*ACT-1. a)** Transient expression of *Pt*SDR-1 and *Pt*SDR-2 together with a biosynthetic module for **4** production (*Pt*LDC, *Pt*CAO, and *Pt*PIKS) in *N. benthamiana*. Shown are LC–MS extracted ion chromatograms (EICs) for **4** ([M + H]$^+$ = $m/z$ 142.1226) and products of *Pt*SDR-1 and *Pt*SDR-2 (A, B, C and D) that each pertain to a single reduction ([M + H]$^+$ = $m/z$ 144.1383), which are shown to represent stereoisomers of **5** via comparison to authentic standards. **b)** MS$^2$ spectra ($m/z$ 144.1383, 20 V) for the new compounds produced by *Pt*SDR-1 (A and B) and *Pt*SDR-2 (C and D) in comparison to the co-eluting stereoisomers of **5** standard. **c)** Chiral LC–MS analysis of the biosynthetic products produced in *N. benthamiana*. Samples were *N*-acetylated to allow for retention and separation on a chiral column. Note that hydroxy groups were also acetylated under our derivatization conditions. The left panel shows biosynthetic *N*-acetyl (NAc)-**4** enantiomers ([M + H]$^+$ = $m/z$ 184.1332) in comparison to synthesized standards, while the right panel shows biosynthetic NAc, O-acetyl (OAc)-**5** stereoisomers ([M + H]$^+$ = $m/z$ 228.1594) in comparison to authentic standards. **d)** Quantification of **4** enantiomers (as *N*-acetylated derivatives) that are consumed by *Pt*SDR-1 and *Pt*SDR-2 and **5** diastereomers that are produced in the *N. benthamiana* transient expression system. n = 3 for

each gene combination. Each bar graph shows the mean +/− standard deviation. **e)** Transient expression of *Pt*ACT-1 with a biosynthetic module for production of **5** diastereomers (*Pt*LDC, *Pt*CAO, *Pt*PIKS and *Pt*SDR-1 or *Pt*SDR-2) in *N. benthamiana*. Shown are LC–MS EICs for **5** diastereomers ([M + H]$^+$ = $m/z$ 144.1383) and products of *Pt*ACT-1 (E, F, G and H) that each pertain to the addition of an acetyl group ([M + H]$^+$ = $m/z$ 186.1489), which are shown to represent diastereomers of **6** via comparison to a synthesized standard (which consists of multiple stereoisomers). **f)** MS$^2$ spectra ($m/z$ 186.1489, 20 V) for the new compounds produced by *Pt*ACT-1 (E and F for experiments with *Pt*SDR-1; G and H for experiments with *Pt*SDR-2) in comparison to the co-eluting stereoisomers of **6** standard. **g)** Chiral LC–MS analysis of the biosynthetic products produced by *Pt*ACT-1 in *N. benthamiana*. Samples were *N*-acetylated to allow for retention and separation on a chiral column. Note that hydroxy groups were also acetylated under our derivatization conditions, so products of *Pt*SDR-1/*Pt*SDR-2 would also gain an *O*-acetyl moiety. Shown are biosynthetic NAc-**6** diastereomers ([M + H]$^+$ = $m/z$ 228.1594) in comparison to authentic standards. **h)** Biosynthetic proposal for the activities of *Pt*SDR-1, *Pt*SDR-2 and *Pt*ACT-1 to yield **6** diastereomers. spon., spontaneous.

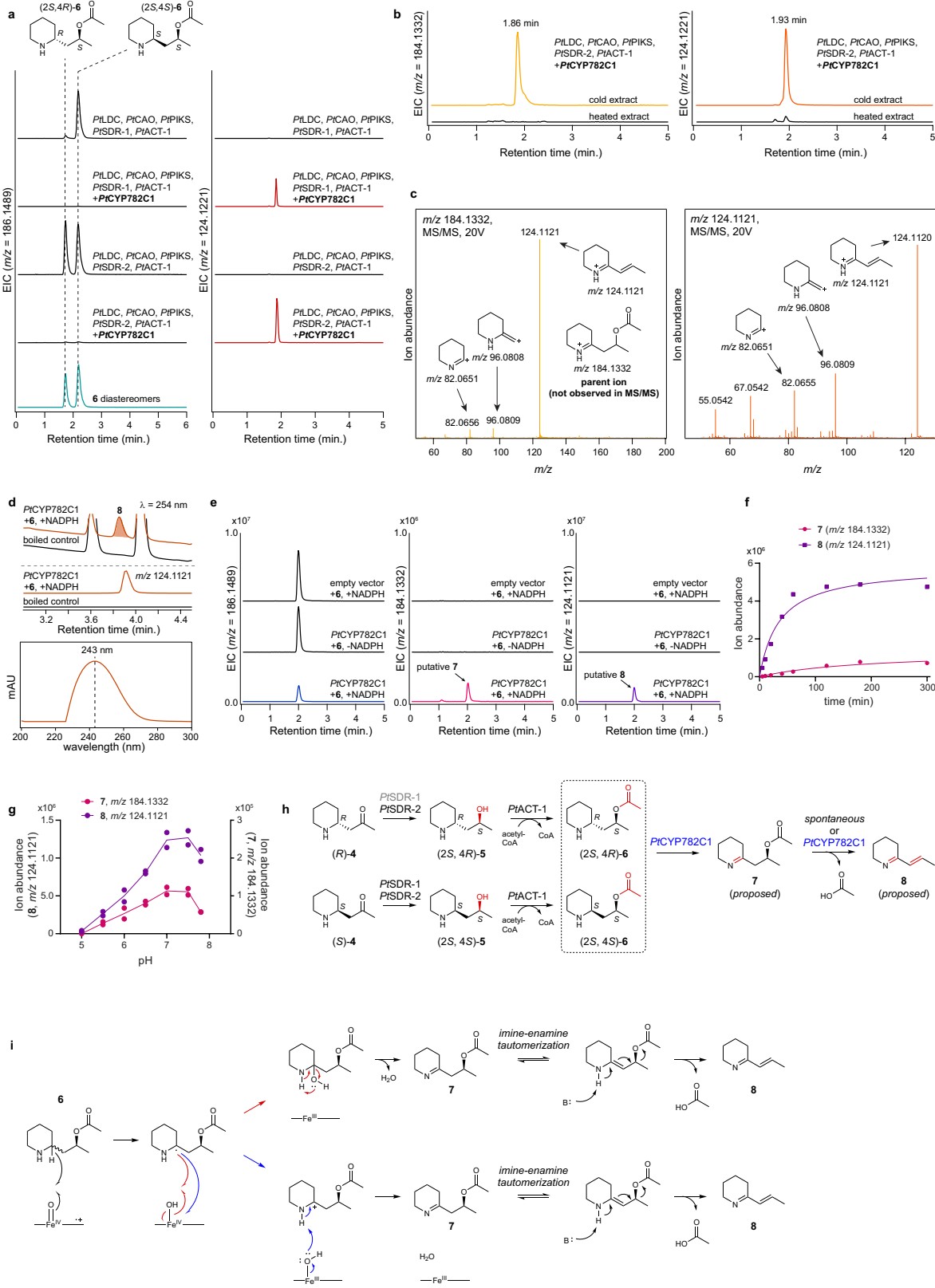

**Extended Data Fig. 2** | See next page for caption.

**Extended Data Fig. 2 | Functional characterization of *Pt*CYP782C1.**

**a)** Transient expression of *Pt*CYP782C1 with a biosynthetic module for the production of **6** diastereomers (*Pt*LDC, *Pt*CAO, *Pt*PIKS, *Pt*SDR-1 or *Pt*SDR-2 and *Pt*ACT-1) in *N. benthamiana*. Shown are LC–MS extracted ion chromatograms (EICs) for **6** diastereomers ([M + H]$^+$ = $m/z$ 186.1489) and a product from *Pt*CYP782C1 activity that represents both an oxidation and elimination of the *O*-acetyl group ([M + H]$^+$ = $m/z$ 124.1121). Note that **6** is detected here with C18 analysis, while $m/z$ 124.1121 is observed with HILIC analysis. **b)** The two new mass features (putative **7**, left panel, [M + H]$^+$ = $m/z$ 184.1332; putative **8**, right panel, [M + H]$^+$ = $m/z$ 124.1121) generated by *Pt*CYP782C1 activity were detected in leaf extract that was prepared under cold conditions, but were mostly lost upon incubation at room temperature. **c)** MS$^2$ spectra of the two new compounds produced by *Pt*CYP782C1 ($m/z$ 184.1332, 20 V and $m/z$ 124.1121, 20 V), along with predicted ion fragment structures. **d)** UV analysis of **8** produced via the activity of yeast microsomes enriched with *Pt*CYP782C1, with **6** and NADPH as substrates. Shown in the top panel are the DAD (λ = 254 nm) and extracted ion ($m/z$ 124.1121) chromatograms from LC-DAD-MS analysis. The bottom panel shows the background-extracted UV spectrum of **8** from LC-DAD analysis. Note that retention time differences between this panel and panel b are due to different columns and LC methods. **e)** In vitro assays with yeast microsomes containing *Pt*CYP782C1 protein. Shown are HILIC LC–MS chromatograms representing **6**, as well as the two mass features (**7**, $m/z$ 184.1332 and **8**, $m/z$ 124.1121) previously identified as putative products of *Pt*CYP782C1. We note that diastereomers of **6** are not resolved during HILIC analysis (as they are in C18 analysis) and thus only one co-eluting peak is observed here. **f)** Formation of **7** and **8** over the course of an in vitro reaction with *Pt*CYP782C1-enriched yeast microsomes. Product abundance is calculated as the integration of the peak generated in the EIC for each mass ion. **g)** Relative activity of *Pt*CYP782C1 microsomes at varying pH, as determined via production of both **7** and **8**. Note that the scales differ between the left and right axes. **h)** Biosynthetic proposal for the activity of *Pt*CYP782C1 on **6** diastereomers to produce **7** and **8**. **i)** Possible catalytic mechanisms for the conversion of **6** into **8**.

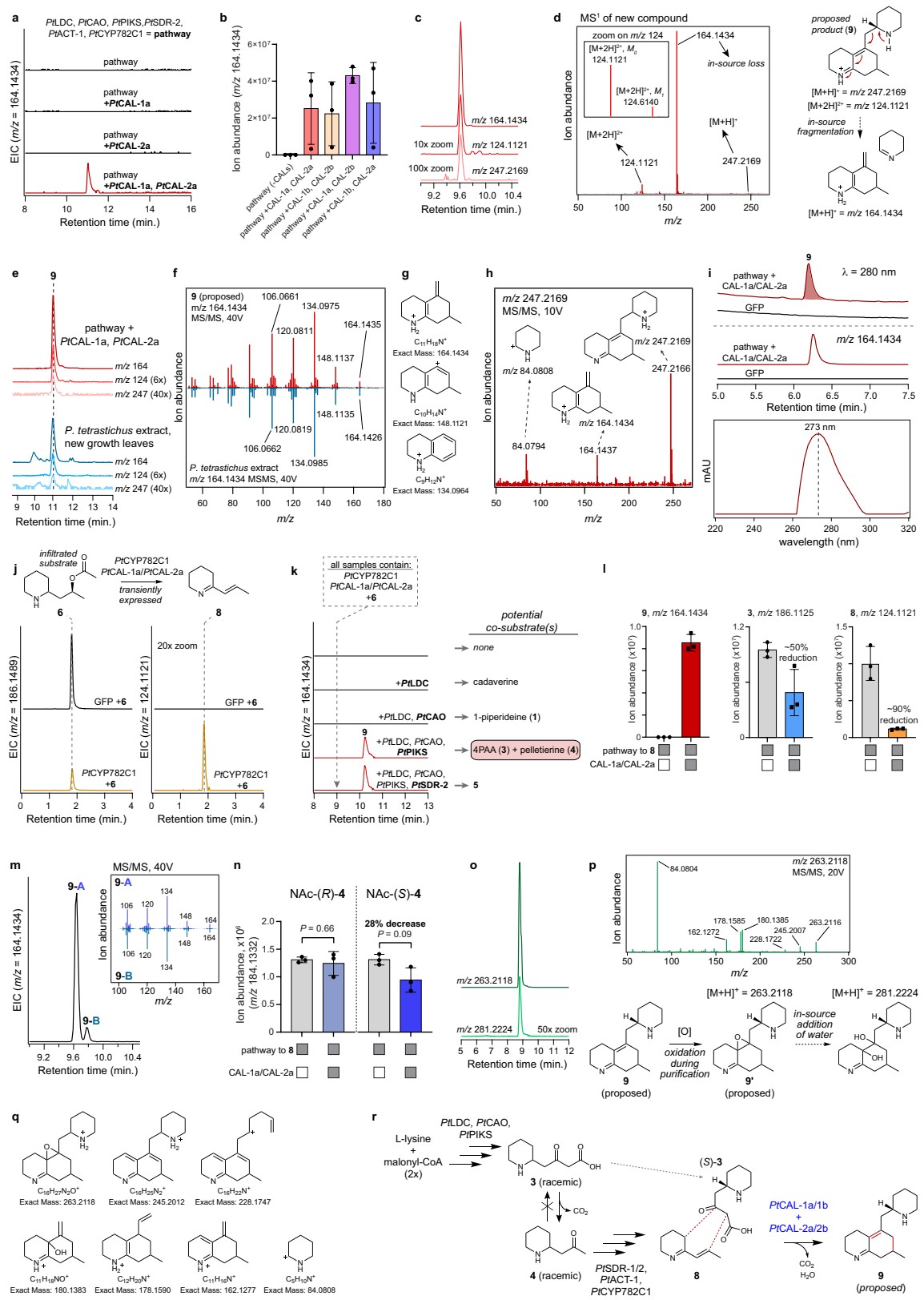

**Extended Data Fig. 3** | See next page for caption.

**Extended Data Fig. 3 | Functional characterization of *Pt*CAL-1 and *Pt*CAL-2 in *Nicotiana benthamiana*. a)** Transient expression of *Pt*CAL-1a and *Pt*CAL-2a with a biosynthetic module for producing **8** (*Pt*LDC, *Pt*CAO, *Pt*PIKS, *Pt*SDR-2, *Pt*ACT-1 and *Pt*CYP782C1). Shown is an LC–MS extracted ion chromatogram (EIC) for the major ion ([M + H]$^+$ = $m/z$ 164.1434) associated with the activity of *Pt*CAL-1a/*Pt*CAL-2a when they are both co-expressed in the transient expression system. **b)** Quantification of new product ($m/z$ 164.1434) abundance through the activity of *Pt*CAL-1 and *Pt*CAL-2 homologues in different combinations. Each bar graph shows the mean +/− standard deviation. n = 3 infiltrated leaves for each condition. **c)** Multiple mass ions were found to co-elute with $m/z$ 164.1434, suggesting that this ion could be an artefact of in-source fragmentation. **d)** MS$^1$ profile of the new compound generated by *Pt*CAL-1/*Pt*CAL-2. Note the presence of presumed parent mass ions ([M + H]$^+$ = $m/z$ 247.2169, [M + 2H]$^{2+}$ = $m/z$ 124.1121), which suggest that $m/z$ 164.1434 results from an in-source loss of 1-piperideine from the proposed product **9** during ionization in the mass spectrometer. **e)** LC–MS EICs ($m/z$ 164.1434, $m/z$ 124.1121 and $m/z$ 247.2169) comparing the biosynthetic product of *Pt*CAL-1/*Pt*CAL-2 (**9**, proposed) to a co-eluting compound in the new growth leaf tissue of *Phlegmariurus tetrastichus*. **f)** MS$^2$ spectra ($m/z$ 164.1434, 40 V) comparing the biosynthetic product (**9**) to the compound identified in *P. tetrastichus* extract. **g)** Proposed structures for major ion fragments shown in panel f. **h)** MS$^2$ spectrum ($m/z$ 247.2169, 10 V) of the parent ion for the new compound (**9**) with predicted structures of fragments. **i)** UV analysis of **9** produced via biosynthetic reconstitution in *N. benthamiana*. Shown in the top panel are the DAD (λ = 280 nm) and extracted ion ($m/z$ 164.1434) chromatograms from LC-DAD-MS analysis. The bottom panel shows the background-extracted UV spectrum of **9** from LC-DAD analysis. Note that retention time differences between this panel and panels a, c and e are due to different columns and LC methods. **j)** Co-infiltration of **6** ($m/z$ 186.1489, left panel) as a substrate for transiently expressed *Pt*CYP782C1, with *Pt*CAL-1 and *Pt*CAL-2a co-expressed, leads to production of **8** ($m/z$ 124.1121, middle panel).

However, as shown in panel k, this does not lead to production of **9**. **k)** Deconvolution of the substrates required for *Pt*CAL-1/*Pt*CAL-2 activity. For this, *Pt*CYP782C1, *Pt*CAL-1 and *Pt*CAL-2 were transiently expressed in *N. benthamiana* and **6** was co-infiltrated as substrate. With this established, different combinations of upstream genes were included in the transient co-expression system to provide putative cosubstrates necessary for the formation of **9** ($m/z$ 164.1434). **l)** Production of **9** ($m/z$ 164.1434) coincides with the depletion of **3** ($m/z$ 186.1125) and **8** ($m/z$ 124.1121). Relative product abundance was quantified via integration of peaks generated in EICs. Each bar graph shows the mean +/− standard deviation. n = 3 infiltrated leaves for each condition. **m)** Observations of major (**9**-A) and minor (**9**-B) diastereomers of **9** upon biosynthetic reconstitution. Shown here is an EIC LC–MS chromatogram of **9**, as well as MS$^2$ comparison between the two diastereomers. **n)** Chiral chromatography of *N*-acetylated precursors was performed to assess which enantiomer of **3** (measured here via consumption of **4**) serves as the substrate for production of **9**. Each bar graph shows the mean +/− standard deviation. n = 3 infiltrated leaves for each condition. Statistical comparisons were made using a two-tailed Welch's t-test assuming unequal variance. **o)** HILIC LC–MS analysis of a new major compound ([M + H]$^+$ = $m/z$ 263.2118) purified while trying to isolate **9**. This compound corresponds to the addition of an oxygen, suggesting this to be an oxidized product of **9**. We also observed an in-source ion fragment that pertains to the addition of a water ([M + H]$^+$ = $m/z$ 281.2224). **p)** MS$^2$ spectrum ($m/z$ 263.2118, 20 V) of putative **9'** and proposed oxidation of **9** to produce **9'**, which can undergo water addition during ionization. **q)** Predicted structures of major MS$^2$ ion fragments. The corresponding NMR data for **9'** can be found in Supplementary Figs. 12–18. **r)** Biosynthetic proposal for the condensation of (*S*)-**3** and **8** by *Pt*CAL-1 and *Pt*CAL-2 to produce the proposed phlegmarane scaffold of **9**. Partial NMR data for the structural characterization of **9** can be found in Supplementary Figs. 10 and 11.

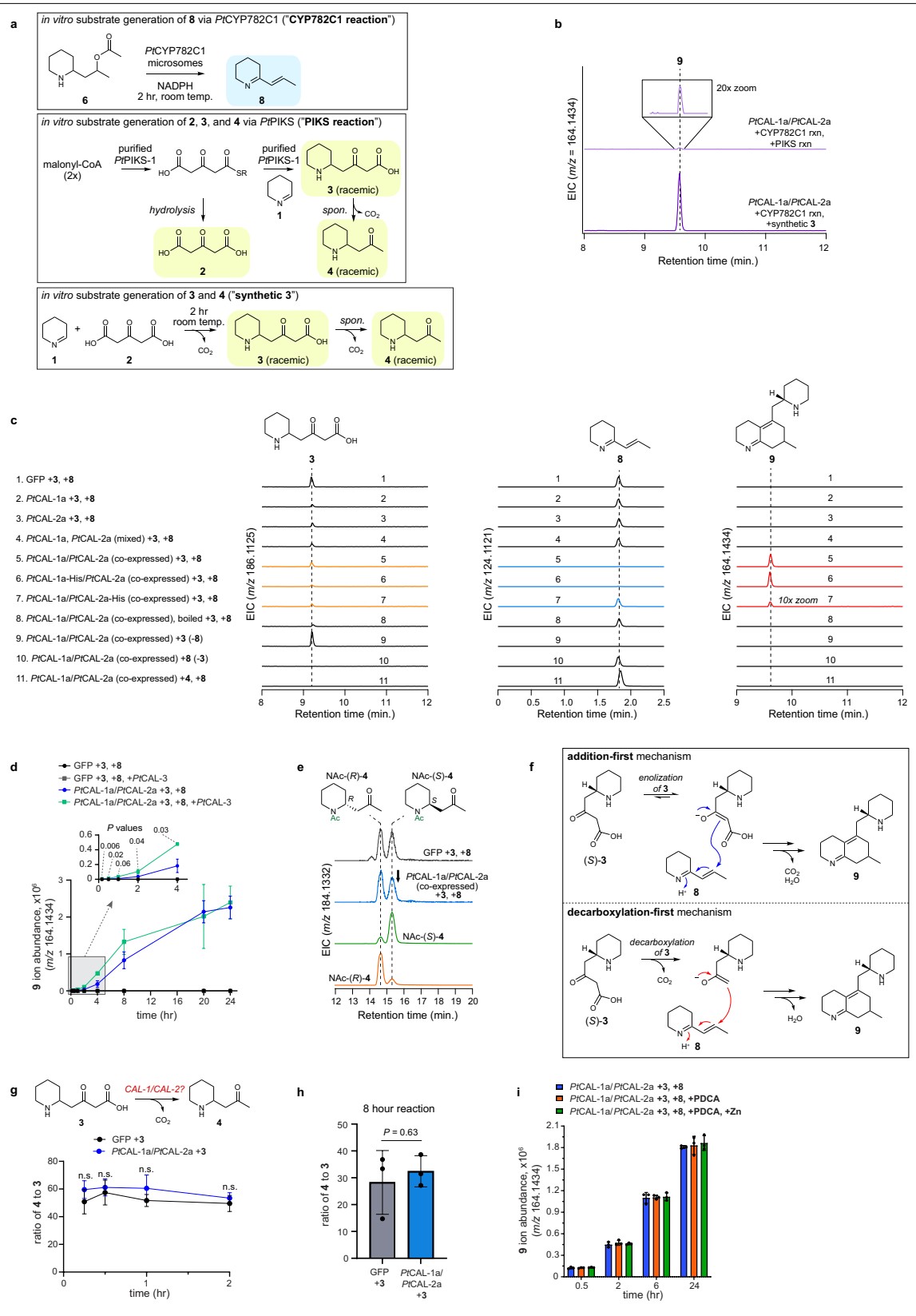

**Extended Data Fig. 4** | See next page for caption.

**Extended Data Fig. 4 | In vitro characterization of *Pt*CAL-1 and *Pt*CAL-2 from isolated apoplast extract. a)** Enzymatic and synthetic reactions used to produce substrates for assays with *Pt*CAL-1a and *Pt*CAL-2a apoplast extracts. **b)** Confirmation of *Pt*CAL-1a/*Pt*CAL-2a (co-expressed) apoplast activity when the PIKS reaction or **1** and **2** are provided as substrates along with the CYP782C1 reaction. Shown is a LC–MS extracted ion chromatogram (EIC) for **9** ($m/z$ 164.1434). Note that production of **9** in this system was dramatically higher when **1** and **2** were used as substrates (to generate **3**) with the CYP782C1 reaction. For all other panels in this figure, indication of +**3** indicates that **1** and **2** were used to produce this substrate spontaneously in vitro. **c)** In vitro apoplast extract reactions with different combinations of apoplast extracts and substrates. The different conditions are listed and numbered to the left of this panel. Shown are the EICs for the substrates (**3** and **8**) as well as the product (**9**). Note that in these experiments, **3** is generated by the spontaneous condensation of **1** and **2**. **d)** Time course of **9** production, as measured via ion abundance ($m/z$ 164.1434) Shown are GFP apoplast extracts (control) or *Pt*CAL-1a/*Pt*CAL-2a (co-expressed) extracts with **3** and **8** generated as in vitro substrates. Additionally, the presence of *Pt*CAL-3 in this reaction was assessed here. n = 3 individual reactions for each condition. Shown in the inset are $P$ values for the statistical comparison between *Pt*CAL-1a/*Pt*CAL-2a +/− *Pt*CAL-3. **e)** Chiral LC–MS EICs analysing the abundance of *N*-acetylated **4** enantiomers in the *Pt*CAL-1a/*Pt*CAL-2a +**3**, +**8** reaction. Note the decrease in the abundance of NAc-(*S*)-**4** in the presence of *Pt*CAL-1a/*Pt*CAL-2a (indicated with arrow). This is quantified in Fig 3e. **f)** Two possible mechanisms to initiate formation of the **9** scaffold. **g)** Analysis to determine if *Pt*CAL-1a/*Pt*CAL-2a accelerates decarboxylation of **3**. Shown are the ratios of **4** to **3** ion abundances over two hours when **3** is included as a substrate alone with either GFP or *Pt*CAL-1a/*Pt*CAL-2a. n = 3 individual reactions for each condition. n.s. = not significant, $P > 0.05$. **h)** Eight-hour time point for assessing the potential decarboxylation of **3** by *Pt*CAL-1a/*Pt*CAL-2a apoplast. n = 3 individual reactions for each condition. **i)** Effect of a zinc (Zn) chelator (2,6-pyridinedicarboxylic acid, PDCA) and zinc supplementation of the enzyme activity of *Pt*CAL-1a/*Pt*CAL-2a, as measured by **9** ion abundance. n = 3 individual reactions for each condition. For all statistical analyses in this figure, a two-tailed Welch's t-test assuming unequal variance was used. All bar graphs in this figure show the mean +/− standard deviation.

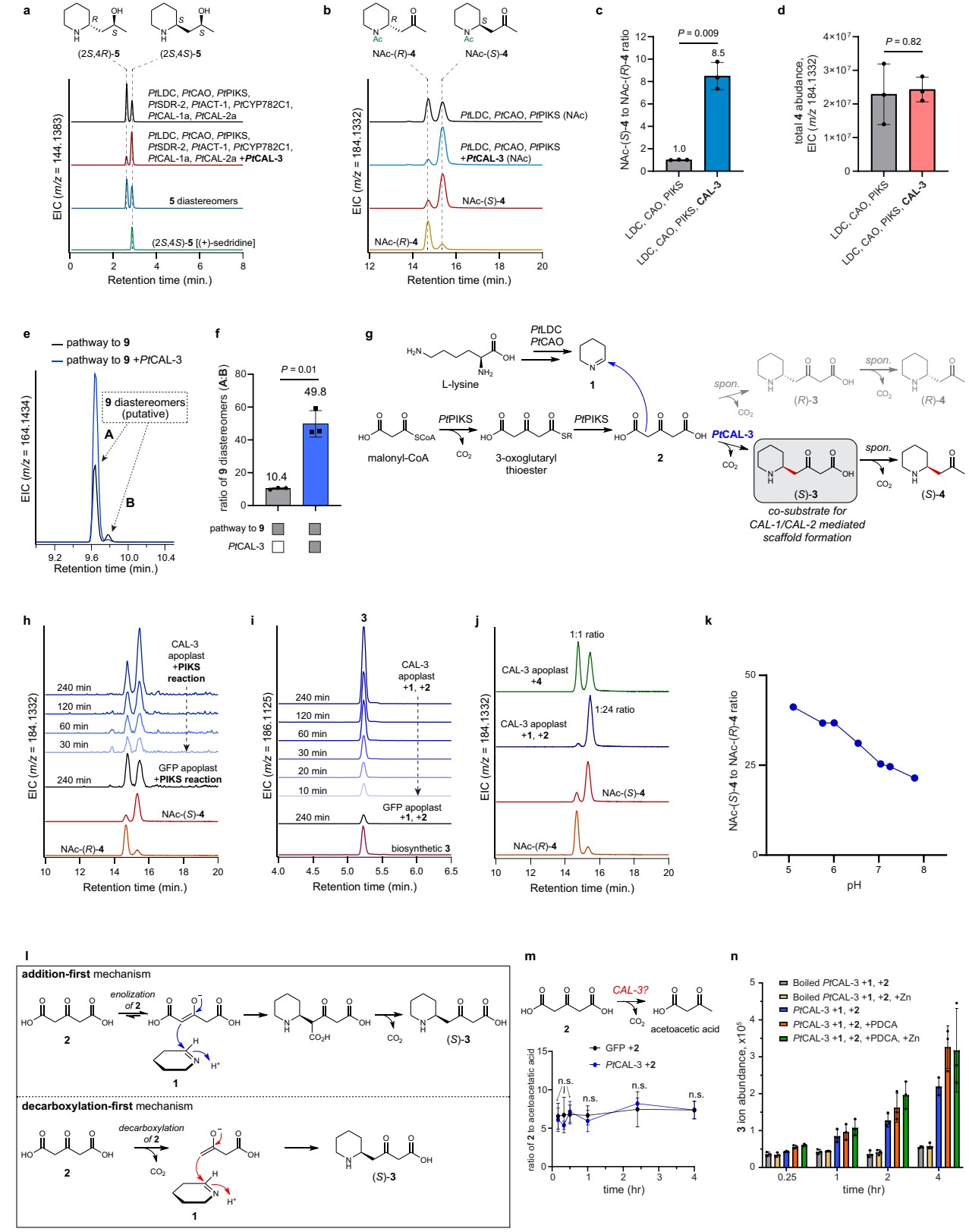

**Extended Data Fig. 5 |** See next page for caption.

**Extended Data Fig. 5 | Functional characterization of *Pt*CAL-3 in *Nicotiana benthamiana* and in vitro with isolated apoplast extract. a)** Transient expression of *Pt*CAL–3 with the pathway to produce **9** (*Pt*LDC, *Pt*CAO, *Pt*PIKS, *Pt*SDR-2, *Pt*ACT-1, *Pt*CYP782C1, *Pt*CAL-1a and *Pt*CAL-2a). Shown are LC–MS extracted ion chromatograms (EICs) for the **5** diastereomer (*m/z* 144.1383) intermediates that remain in this biosynthetic system. **b)** Chiral LC–MS analysis of *N*-acetylated products from a transient expression system that generates **4** (NAc-**4** = *m/z* 184.1332) with or without co-expression of *Pt*CAL-3. **c)** Effect on the ratio of (*S*)-**4** to (*R*)-**4** when *Pt*CAL-3 is included with *Pt*LDC, *Pt*CAO and *Pt*PIKS in *N. benthamiana*. n = 3 infiltrated leaves for each condition. **d)** Effect of *Pt*CAL-3 on the total accumulation of **4** in *N. benthamiana*. For both panels c and d, each bar graph shows the mean +/− standard deviation. n = 3 infiltrated leaves for reach condition. The statistical comparison was made using a two-tailed Welch's t-test assuming unequal variance. **e)** Effect of including *Pt*CAL-3 on the ratio of **9** diastereomers (*m/z* 164.1434), as observed via LC–MS. **f)** Quantification of the ratio of **9** diastereomers when *Pt*CAL-3 is absent or co-expressed with the rest of the pathway for **9** biosynthesis. Bar graphs show the mean +/− standard deviation, with the mean shown above each bar. n = 3 infiltrated leaves for each condition. The statistical comparison was made using a two-tailed Welch's t-test assuming unequal variance. **g)** Biosynthetic proposal for the function of *Pt*CAL-3 based upon its effect on pathway reconstitution in *N. benthamiana*. The specific production of (*S*)-**3** by *Pt*CAL-3 explains the enrichment of (*S*,*S*)-**5** shown in panel a, the enrichment of (*S*)-**4** shown in panels b and c, as well as the increase in the major **9** diastereomer (**9**-A) shown in panels e and f. We propose that the minor **9** diastereomer (**9**-B) is formed via the low incorporation of (*R*)-**3** as a cosubstrate with **8**. spon., spontaneous. **h)** In vitro assay with *Pt*CAL-3-enriched apoplast and purified *Pt*PIKS. Shown here are chiral LC–MS EICs for *N*-acetylated **4** enantiomers (*m/z* 184.1332). Apoplast from plants expressing GFP was used as a negative control. Reactions contained an enzymatic mixture for the production of **2**, **3** and **4** (purified *Pt*PIKS-1 +malonyl-CoA, +**1**), as defined in Extended Data Fig 4a. Note the enrichment of (*S*)-**4** over time in the reactions that contain *Pt*CAL-3. **i)** LC–MS analysis (HILIC) of in vitro *Pt*CAL-3 apoplast reactions where **1** and **2** are used as substrates. Shown are EICs for **3** (*m/z* 186.1125) over time. Biosynthetic **3** (shown as a positive control) was generated via transient expression of *Pt*LDC, *Pt*CAO and *Pt*PIKS in *N. benthamiana*, as usual. **j)** In vitro assay with *Pt*CAL-3 apoplast where either racemic **4** or **1** and **2** (which can spontaneously condense to produce **3** and subsequently, **4**) are included as substrates. Shown here are chiral LC–MS EICs for *N*-acetylated **4** enantiomers (*m/z* 184.1332). The ratio of enantiomers is listed next to the peaks for each reaction. **k)** Assessment of *Pt*CAL-3 apoplast activity at different pH conditions. This was measured by determining the ratio of (*S*)-**4** to (*R*)-**4** (*N*-acetylated derivatives) via chiral LC–MS at the end point of each reaction. **l)** Two possible mechanisms for the *Pt*CAL-3-catalysed condensation of **1** and **2** to produce (*S*)-**3**. **m)** Analysis to determine if *Pt*CAL-3 accelerates decarboxylation of **2**. Shown are the ion abundance ratios of **2** ([M+Na]$^+$ = *m/z* 169.0107) to acetoacetic acid ([M+Na]$^+$ = *m/z* 125.0209) over four hours when **2** is included as a substrate alone with either GFP or *Pt*CAL-3. n = 3 individual reactions for each condition. n.s. = not significant, *P* > 0.05. **n)** Effect of a zinc (Zn) chelator (2,6-pyridinedicarboxylic acid, PDCA) and zinc supplementation on the enzyme activity of *Pt*CAL-3, as measured by **3** ion abundance. n = 3 individual reactions for each condition. Boiled *Pt*CAL-3 was included as a negative control since spontaneous formation of **3** can occur when **1** and **2** are co-incubated. For all statistical analyses in this figure, a two-tailed Welch's t-test assuming unequal variance was used. All bar graphs in this figure show the mean +/− standard deviation.

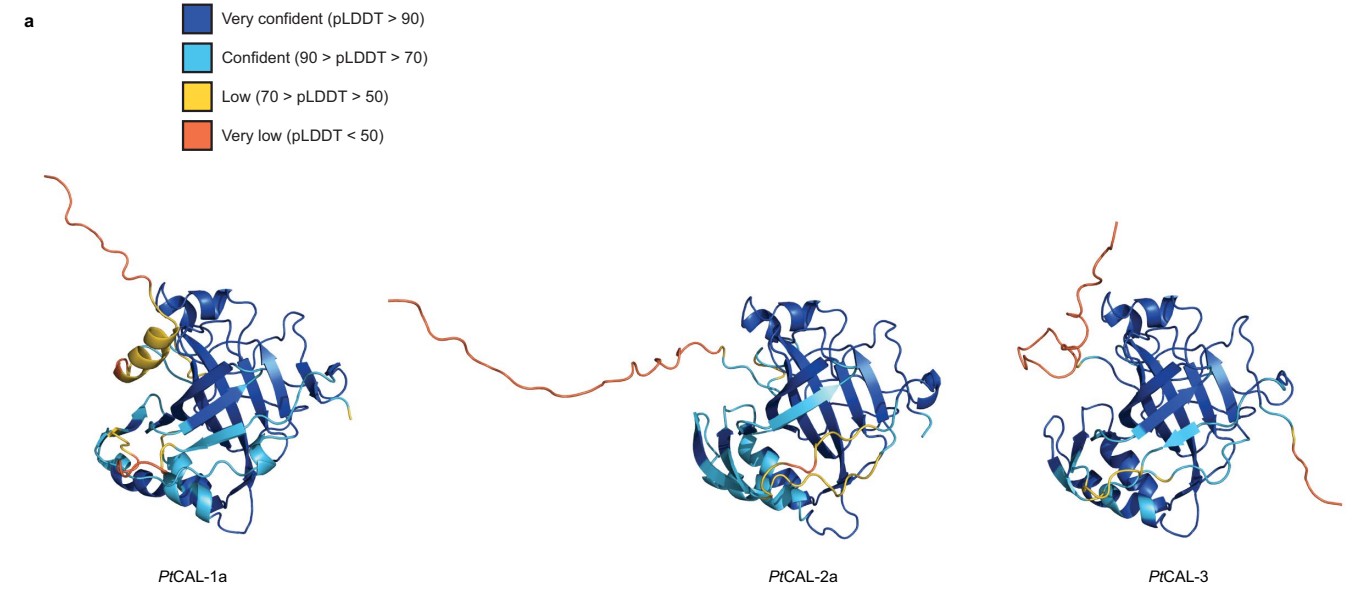

**a**

■ Very confident (pLDDT > 90)
■ Confident (90 > pLDDT > 70)
■ Low (70 > pLDDT > 50)
■ Very low (pLDDT < 50)

*Pt*CAL-1a            *Pt*CAL-2a            *Pt*CAL-3

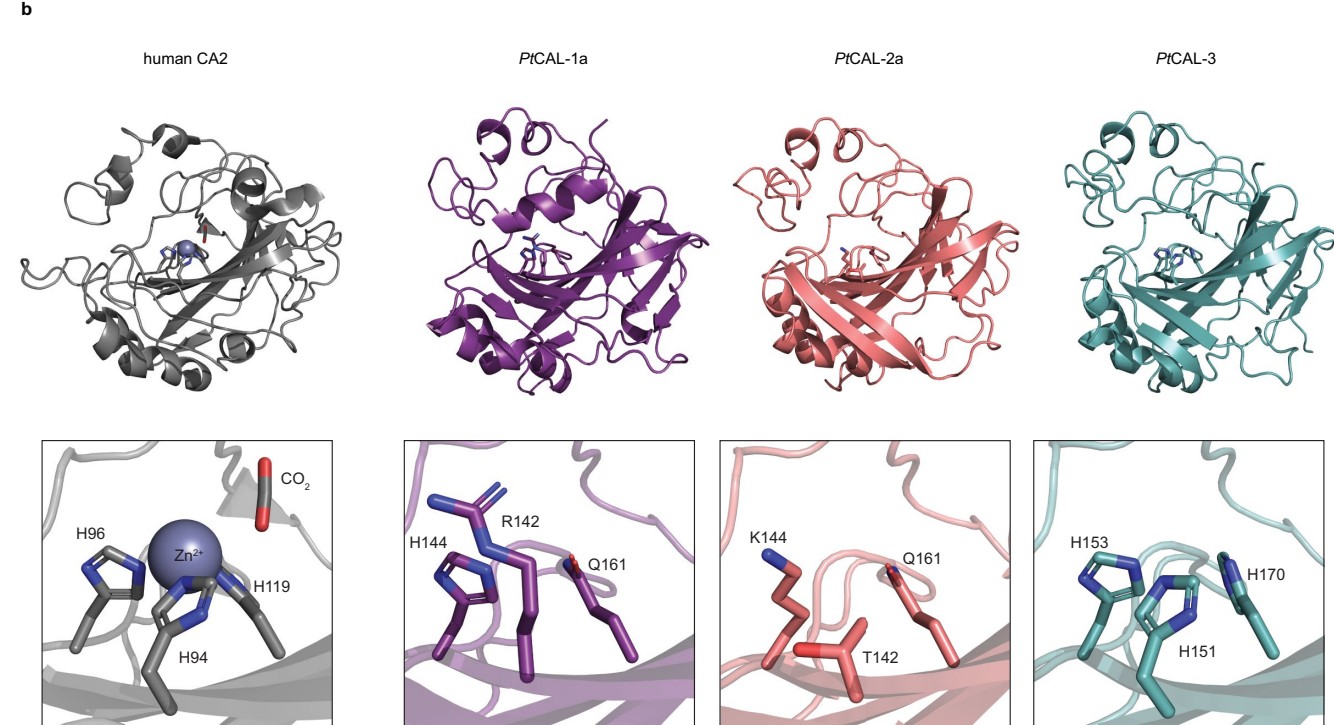

**b**

human CA2            *Pt*CAL-1a            *Pt*CAL-2a            *Pt*CAL-3

**Extended Data Fig. 6 | Structural modelling of CAL proteins. a)** Structures of *Pt*CAL-1a, *Pt*CAL-2a and *Pt*CAL-3 were modelled using AlphaFold2 via ColabFold (v1.5.2)[48]. The predicted N-terminal signal peptide for each CAL protein was removed prior to structural prediction. Shown here are the highest-ranked models for each structure, which are coloured according to the predicted local distance difference test (pLDDT) confidence score for each residue. Note that the top left, disordered region of each protein corresponds to the N-terminal sequence immediately downstream from the predicted signal peptide. **b)** Comparison of overall structure and active site architecture of modelled *P. tetrastichus* CALs compared to human carbonic anhydrase 2 (CA2, PDB structure 2VVA). For clarity, the disordered N-terminal regions were removed from the CAL proteins in this panel. Residues are numbered based upon the full-length version of each protein.

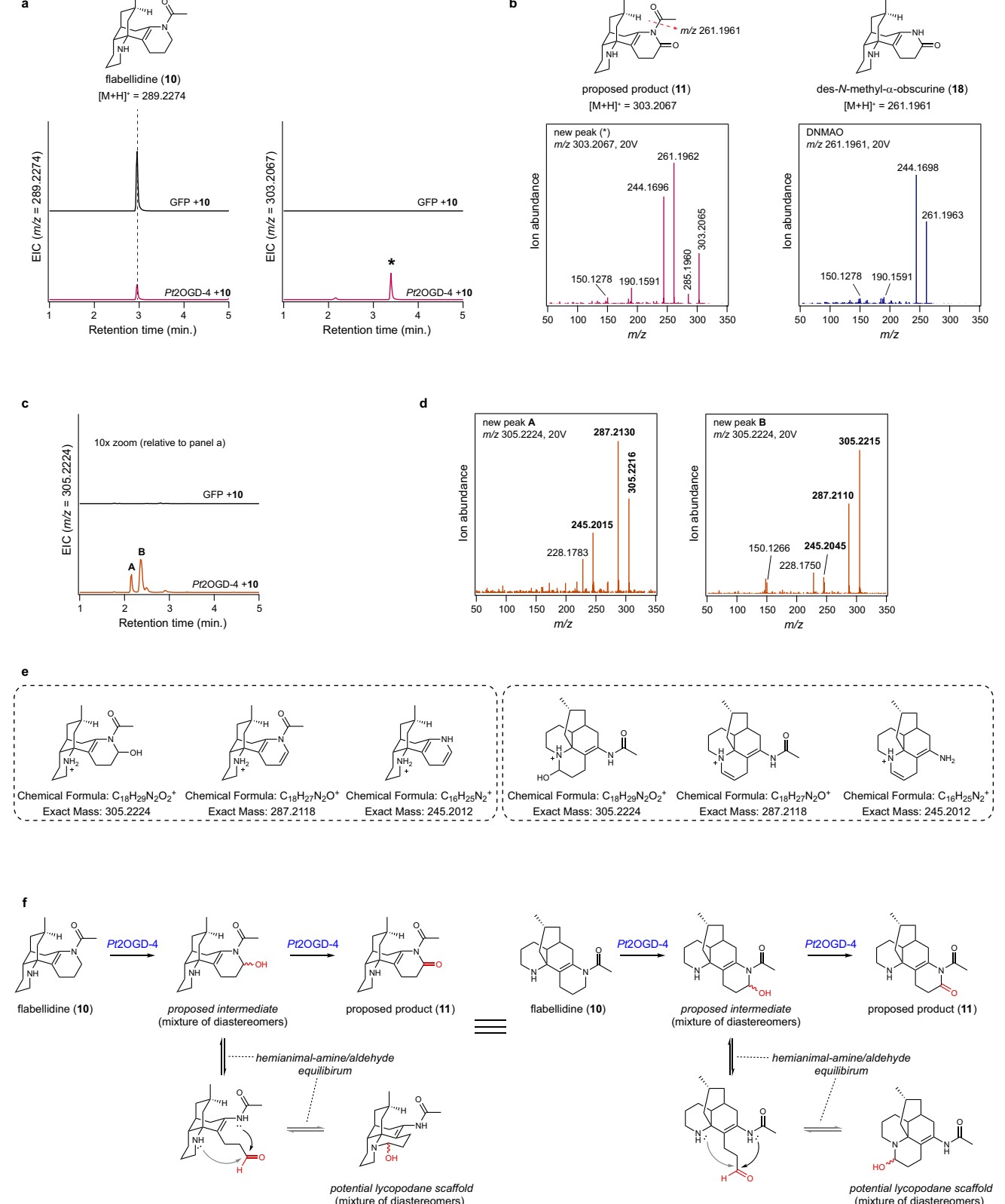

**Extended Data Fig. 7 | Functional characterization of *Pt*2OGD-4.**
**a)** Transient expression of *Pt*2OGD-4 in *N. benthamiana* with co-infiltration of
**10** as substrate. Shown are LC–MS extracted ion chromatograms (EICs) for the
**10** substrate ([M + H]⁺ = *m/z* 289.2274, left panel) and a product (*) of *Pt*2OGD-4
that corresponds to the addition of a carbonyl ([M + H]⁺ = *m/z* 303.2067, right
panel). **b)** MS² spectra of the new compound (*m/z* 303.2067, 20 V) in comparison
to that of **18** (*m/z* 261.1961). Note the similarity in major ion fragments, which
suggests that the new compound (proposed as **11**) bears structural similarity to

**18**. **c)** Minor products (peaks A and B) pertaining to the addition of a hydroxyl
([M + H]⁺ = *m/z* 305.2224) are also generated by *Pt*2OGD-4 activity. **d)** MS²
spectra (*m/z* 305.2224, 20 V) for compounds "A" and "B" generated by *Pt*2OGD-
4. **e)** Putative structures of the ion fragments shown in bold in panel d.
**f)** Biosynthetic proposal for the conversion of **10** into **11** by *Pt*2OGD-4. Note
that the right panel shows the same chemistry as the left panel, but in a
different 3D orientation.

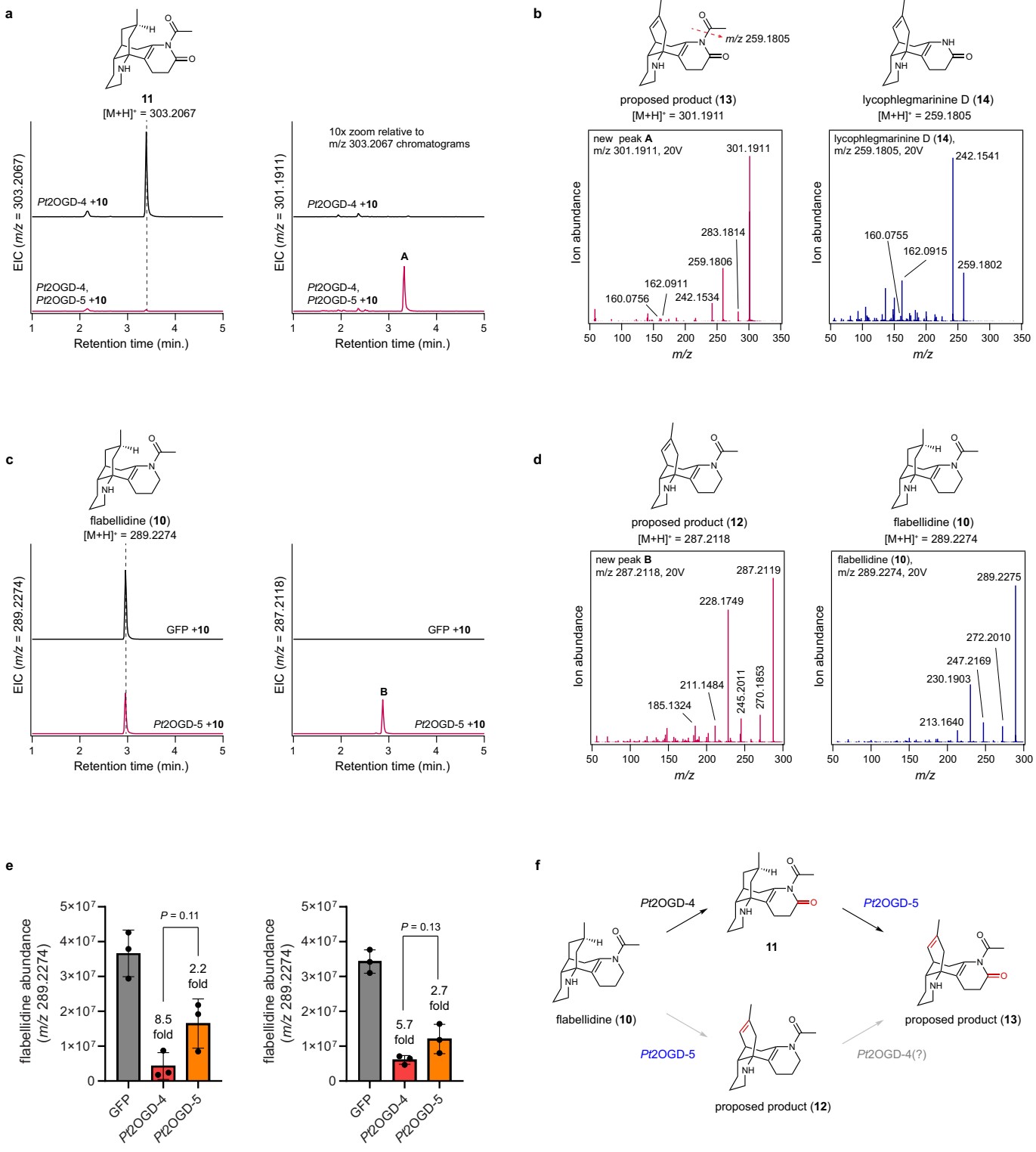

**Extended Data Fig. 8** | See next page for caption.

**Extended Data Fig. 8 | Functional characterization of *Pt*2OGD-5. a)** Transient expression of *Pt*2OGD-5 with *Pt*2OGD-4 in *N. benthamiana* with co-infiltration of **10** as substrate. Shown are LC–MS extracted ion chromatograms (EICs) for the product of *Pt*2OGD-4 (**11**, $m/z$ 303.2067, left panel) and a product (A) of *Pt*2OGD-5 that corresponds to a desaturation ($[M + H]^+ = m/z$ 301.1911, right panel). **b)** MS$^2$ spectra of the new compound "A" ($m/z$ 301.1911, 20 V) in comparison to that of **14** ($m/z$ 259.1805). Note the similarity in major ion fragments, which suggests that the new compound (proposed as **13**) bears structural similarity to **14**. **c)** Transient expression of *Pt*2OGD-5 alone in *N. benthamiana* with co-infiltration of **10** as substrate. Shown are LC–MS extracted ion chromatograms (EICs) for **10** as substrate ($m/z$ 289.2274, left panel) and a product (B) of *Pt*2OGD-5 that corresponds to a desaturation ($[M + H]^+ = m/z$ 287.2118, right panel). **d)** MS$^2$ spectra of the new compound "B" ($m/z$ 287.2118, 20 V) in comparison to that of **10** ($m/z$ 289.2274). Note that the major ion fragments in "B" are typically 2 $m/z$ units less than those of **10**, which supports that the new compound (proposed as **12**) bears the same scaffold as **10**, but with a desaturation. **e)** Comparison of **10** consumption by *Pt*2OGD-4 vs. *Pt*2OGD-5. Each of the bar graphs represents an independent experiment. Pairwise comparisons between *Pt*2OGD-4 and *Pt*2OGD-5 reactions were assessed using a two-tailed Welch's t-test, assuming unequal variance. n = 3 infiltrated leaves for each condition. Each bar graph shows the mean +/− standard deviation. **f)** Biosynthetic proposal for the activity of *Pt*2OGD-5. While *Pt*2OGD-5 can desaturate **10** to produce **12** (putative), *Pt*2OGD-4 appears to have higher activity on **10**, suggesting that *Pt*2OGD-4 activity prior to *Pt*2OGD-5 activity is the major metabolic route for producing **13** (putative).

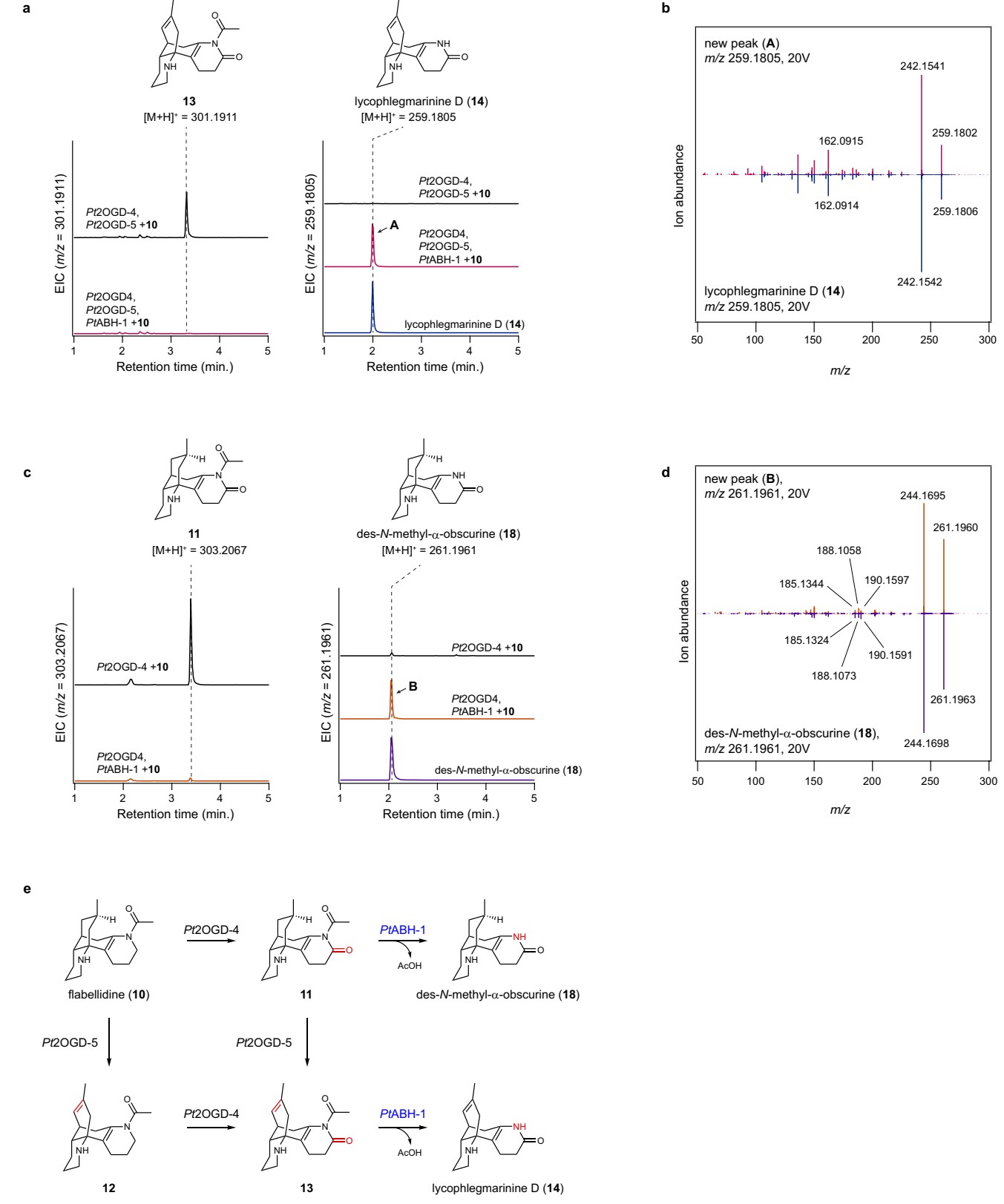

**Extended Data Fig. 9** | See next page for caption.

**Extended Data Fig. 9 | Functional characterization of *Pt*ABH-1. a)** Transient expression of *Pt*ABH-1 with *Pt*2OGD-5 and *Pt*2OGD−4 in *N. benthamiana* with co-infiltration of **10** as substrate. Shown are LC–MS extracted ion chromatograms (EICs) for the product of *Pt*2OGD−4 and *Pt*2OGD-5 (**13**, *m/z* 301.1911, left panel) and a product (A) of *Pt*ABH-1 that corresponds to a loss of an acetyl group ([M + H]⁺ = *m/z* 259.1805, right panel), which is confirmed to be **14** via comparison to an authentic standard. **b)** MS² spectra of the new compound "A" (*m/z* 259.1805, 20 V) in comparison to that of **14** (*m/z* 259.1805, 20 V). **c)** Transient expression of *Pt*ABH-1 with *Pt*2OGD-4 (*Pt*2OGD-5 omitted) in *N. benthamiana* with co-infiltration of **10** as substrate. Shown are LC–MS EICs for the product of *Pt*2OGD-4 (**11**, *m/z* 303.2067, left panel) and a new product (B) of *Pt*ABH-1 that corresponds to the loss of an acetyl group, ([M + H]⁺ = *m/z* 261.1961, right panel), which is confirmed to be **18** via comparison to an authentic standard. **d)** MS² spectra of the new compound "B" (*m/z* 261.1961, 20 V) in comparison to that of **18** (*m/z* 261.1961, 20V). **e)** Biosynthetic proposal for the activity of *Pt*ABH-1, which can deacetylate either **11** or **13** to produce **18** or **14**, respectively. Critically, the ability to access confirmed standards biosynthetically verifies the proposed location of the carbonyl installed by *Pt*2OGD-4 and the double bond installed by *Pt*2OGD-5.

**a**

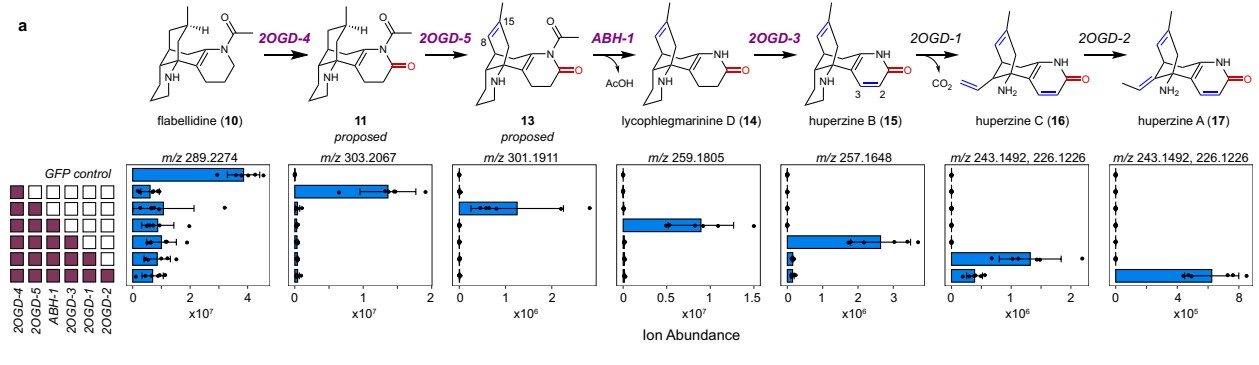

**b**

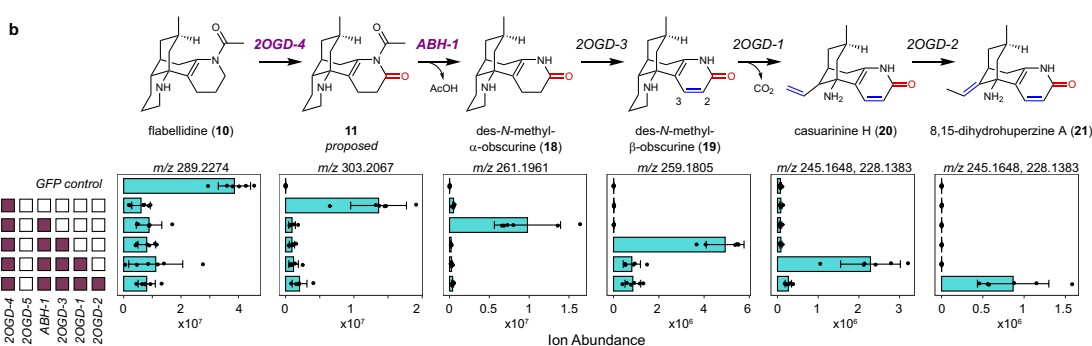

**c**

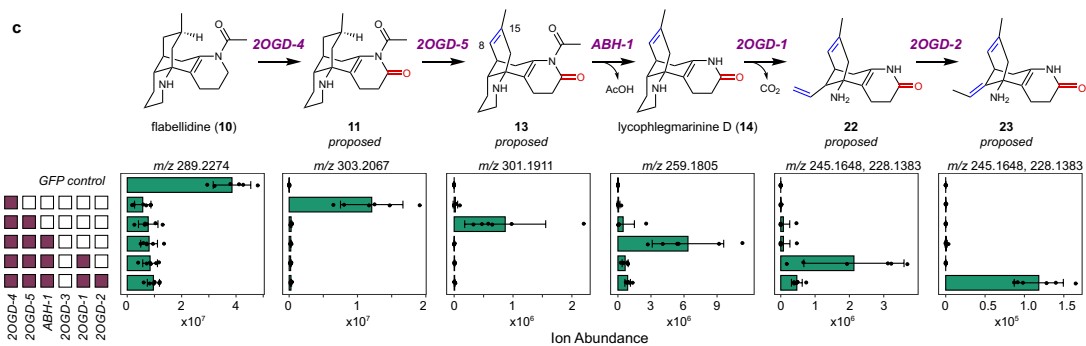

**d**

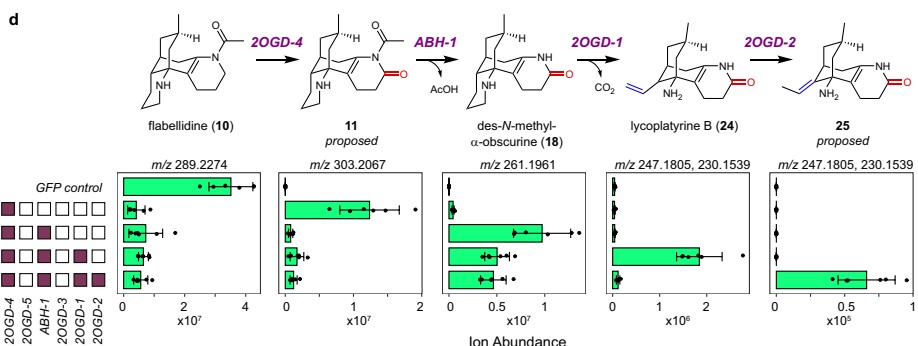

**Extended Data Fig. 10 | Step-by-step biosynthesis of downstream Lycopodium alkaloids. a)** Generation of HupA (**17**). **b)** Generation of 8,15-dihydro congeners. **c)** Generation of 2,3-dihydro congeners. **d)** Generation of 2,3,8,15-tetrahydro congeners. For all panels, filled in boxes to the left indicate presence of biosynthetic genes in our *N. benthamiana* transient expression system. Flabellidine (**10**) was co-infiltrated as a substrate in all experiments. Shown below each compound is the mean ion abundance for the indicated mass ions (*m/z*) for each compound. In all panels, n = 6 infiltrated leaves for each experimental condition. Error bars represent +/− standard deviation. New enzymes, or new reactions for previously described enzymes, are coloured purple. Lycopodium alkaloids with common names have been verified with authentic standards. All other structures are proposed based upon MS², biosynthetic logic and/or insight gained from downstream products or known enzyme activities. Additional details can be found in Extended Data Figs. 7–9, Supplemental Results and Supplementary Figs. 4 and 5.

# Reporting Summary

## Statistics

For all statistical analyses, confirm that the following items are present in the figure legend, table legend, main text, or Methods section.

| n/a | Confirmed | |
|---|---|---|
| ☐ | ☒ | The exact sample size (*n*) for each experimental group/condition, given as a discrete number and unit of measurement |
| ☐ | ☒ | A statement on whether measurements were taken from distinct samples or whether the same sample was measured repeatedly |
| ☐ | ☒ | The statistical test(s) used AND whether they are one- or two-sided<br>*Only common tests should be described solely by name; describe more complex techniques in the Methods section.* |
| ☒ | ☐ | A description of all covariates tested |
| ☐ | ☒ | A description of any assumptions or corrections, such as tests of normality and adjustment for multiple comparisons |
| ☐ | ☒ | A full description of the statistical parameters including central tendency (e.g. means) or other basic estimates (e.g. regression coefficient) AND variation (e.g. standard deviation) or associated estimates of uncertainty (e.g. confidence intervals) |
| ☐ | ☒ | For null hypothesis testing, the test statistic (e.g. $F$, $t$, $r$) with confidence intervals, effect sizes, degrees of freedom and $P$ value noted<br>*Give P values as exact values whenever suitable.* |
| ☒ | ☐ | For Bayesian analysis, information on the choice of priors and Markov chain Monte Carlo settings |
| ☒ | ☐ | For hierarchical and complex designs, identification of the appropriate level for tests and full reporting of outcomes |
| ☐ | ☒ | Estimates of effect sizes (e.g. Cohen's *d*, Pearson's *r*), indicating how they were calculated |

*Our web collection on statistics for biologists contains articles on many of the points above.*

## Software and code

Policy information about availability of computer code

| Data collection | No code was utilized to collect data in this study. |
|---|---|
| Data analysis | Routine data compilation was performed in Microsoft Excel 2016. General analysis of LC-MS data was performed with Agilent MassHunter Qualitative Analysis 10.0. Chromatograms and mass spectra were plotted using IGOR Pro 6.0. Bar graphs and line graphs were plotted using GraphPad Prism 9, and this software was also used for routine statistical analysis. Hierarchical clustering analysis was performed using Cluster 3.0.56  R (version 4.2.2) was used for bar graph generation, visualization of hierarchical clustering data, and for performing XCMS analysis.69 Geneious Prime (version 2019.2.3) was used for bioinformatic analyses of nucleic acid and protein sequences. This software was also used for multiple sequence alignments (MUSCLE algorithm) and phylogenetic tree generation (Jukes-Cantor genetic distance model, Neighbor-Joining tree build method). The TargetP-2.0 server (https://services.healthtech.dtu.dk/service.php?TargetP-2.0)58 was used for predicting signal peptides and protein localization. MNova (v1.6) was used for visualization and processing of NMR data. ChemDraw Professional (version 21.0.0.28) was used for chemical structure visualization and analysis. Structural modeling was performed using AlphaFold-Multimer via ColabFold (v1.5.2), and protein models were visualized in PyMol (version 2.5.4). |

For manuscripts utilizing custom algorithms or software that are central to the research but not yet described in published literature, software must be made available to editors and reviewers. We strongly encourage code deposition in a community repository (e.g. GitHub). See the Nature Portfolio guidelines for submitting code & software for further information.

## Data

Policy information about availability of data

All manuscripts must include a data availability statement. This statement should provide the following information, where applicable:

- Accession codes, unique identifiers, or web links for publicly available datasets
- A description of any restrictions on data availability
- For clinical datasets or third party data, please ensure that the statement adheres to our policy

All data in this manuscript are available upon request. The raw RNA-seq data analyzed in this manuscript have previously been deposited to the NCBI Sequence Read Archive (BioProject PRJNA731132).23 Gene sequences for enzymes characterized in this study are deposited in the National Center for Biotechnology (NCBI) GenBank under the following accessions: Pt2OGD-4 (OR538095), Pt2OGD-5 (OR538096), PtABH-1 (OR538097), PtACT-1 (OR538098), PtCAL-1a (OR538099), PtCAL-1b (OR538100), PtCAL-2a (OR538101), PtCAL-2b (OR538102), PtCAL-3 (OR538103), PtCYP782C1 (four homologs; OR538104, OR538105, OR538106, OR538107), PtSDR-1 (OR538108), and PtSDR-2 (OR538109). The UniProt database (https://www.uniprot.org/) was used for identifying and obtaining CAH family sequences that were used in phylogenetic analyses. The human CA2 protein structure (2vva) was acquired from PDB (https://www.rcsb.org/). Any materials generated within this manuscript will be made available, as possible.

## Research involving human participants, their data, or biological material

Policy information about studies with human participants or human data. See also policy information about sex, gender (identity/presentation), and sexual orientation and race, ethnicity and racism.

| Reporting on sex and gender | N/A |
|---|---|
| Reporting on race, ethnicity, or other socially relevant groupings | N/A |
| Population characteristics | N/A |
| Recruitment | N/A |
| Ethics oversight | N/A |

Note that full information on the approval of the study protocol must also be provided in the manuscript.

# Field-specific reporting

Please select the one below that is the best fit for your research. If you are not sure, read the appropriate sections before making your selection.

☒ Life sciences ☐ Behavioural & social sciences ☐ Ecological, evolutionary & environmental sciences

For a reference copy of the document with all sections, see nature.com/documents/nr-reporting-summary-flat.pdf

# Life sciences study design

All studies must disclose on these points even when the disclosure is negative.

| Sample size | In the experiments within this manuscript, we typically used a sample size of three replicates in order to have the power for statistical comparisons. |
|---|---|
| Data exclusions | No data has been excluded from any of the statistical analyses. |
| Replication | All experiments were replicated at least once, and in most circumstances, in greater than three independent experiments. |
| Randomization | Randomization is not relevant to the experiments of this manuscript. The various experimental conditions were specifically defined to probe for the function of distinct enzymes, and there was no random assigning of samples to experiment groups. |
| Blinding | We were not blinded to our analysis; experiments were designed to probe for the function of individual enzymes, and this required the researchers to know which samples contained which combination of enzymes. Thus, in general, blinding was not relevant to our studies. |

# Reporting for specific materials, systems and methods

We require information from authors about some types of materials, experimental systems and methods used in many studies. Here, indicate whether each material, system or method listed is relevant to your study. If you are not sure if a list item applies to your research, read the appropriate section before selecting a response.

## Materials & experimental systems

| n/a | Involved in the study |
|-----|-----------------------|
| ☐ | ☒ Antibodies |
| ☒ | ☐ Eukaryotic cell lines |
| ☒ | ☐ Palaeontology and archaeology |
| ☒ | ☐ Animals and other organisms |
| ☒ | ☐ Clinical data |
| ☒ | ☐ Dual use research of concern |
| ☐ | ☒ Plants |

## Methods

| n/a | Involved in the study |
|-----|-----------------------|
| ☒ | ☐ ChIP-seq |
| ☒ | ☐ Flow cytometry |
| ☒ | ☐ MRI-based neuroimaging |

# Antibodies

| | |
|---|---|
| Antibodies used | 1. Mouse anti-His antibody antibody (Genscript A00186) for detecting the presence of His-tagged proteins in our Nicotiana benthamiana gene expression system.<br>2. Horse anti-mouse IgG, HRP-linked antibody (Cell signaling Technology 7076) for Western blots. |
| Validation | 1. The mouse anti-His antibody antibody (Genscript A00186) is validated by the manufacturer to be specific to 6x His-tagged proteins via ELISA. This antibody has been shown to also be effective against 4x and 5x His tags and to work with both native and denatured synthetic proteins produced in diverse heterologous systems. Further information, including certificates of analysis and additional information on validation are available on the manufacturer website. (https://www.genscript.com/antibody/A00186-THE_His_Tag_Antibody_mAb_Mouse.html).<br>2. The horse anti-mouse IgG, HRP-linked antibody (Cell signaling Technology 7076) is thoroughly validated by the manufacturer with Cell Signaling Technology primary antibodies. Additional information, including certificates of analysis and antibody validation are available on the manufacturer website. (https://www.cellsignal.com/products/secondary-antibodies/anti-mouse-igg-hrp-linked-antibody/7076?_requestid=19655) |

# Dual use research of concern

Policy information about dual use research of concern

## Hazards

Could the accidental, deliberate or reckless misuse of agents or technologies generated in the work, or the application of information presented in the manuscript, pose a threat to:

| No | Yes | |
|----|-----|---|
| ☒ | ☐ | Public health |
| ☒ | ☐ | National security |
| ☒ | ☐ | Crops and/or livestock |
| ☒ | ☐ | Ecosystems |
| ☒ | ☐ | Any other significant area |

## Experiments of concern

Does the work involve any of these experiments of concern:

| No | Yes | |
|----|-----|---|
| ☒ | ☐ | Demonstrate how to render a vaccine ineffective |
| ☒ | ☐ | Confer resistance to therapeutically useful antibiotics or antiviral agents |
| ☒ | ☐ | Enhance the virulence of a pathogen or render a nonpathogen virulent |
| ☒ | ☐ | Increase transmissibility of a pathogen |
| ☒ | ☐ | Alter the host range of a pathogen |
| ☒ | ☐ | Enable evasion of diagnostic/detection modalities |
| ☒ | ☐ | Enable the weaponization of a biological agent or toxin |
| ☒ | ☐ | Any other potentially harmful combination of experiments and agents |

