## [Peer Review File · Nature]

Manuscript Title: Plant carbonic anhydrase-like enzymes in neuroactive alkaloid biosynthesis

Reviewer Comments & Author Rebuttals

Reviewer Reports on the Initial Version:

Referees' comments:

Referee #1 (Remarks to the Author):

Nett et al. describe the biosynthesis of Lycopodium alkaloids. Using transcriptomic and gene screening, the authors identified enzymes that catalyze the Lycopodium scaffold formation and modification. Surprisingly, the key annulation reaction between two units of C8 components is catalyzed by the combination of two carbonic anhydrase-like proteins. A substantial amount of data is provided in this manuscript to support the proposed biosynthetic pathway and enzyme functions. This is a topnotch study and should have a broad impact in the scientific community. I believe this work would be suitable for acceptance in this journal after addressing some concerns listed below.

1. For the stereochemical notion, it would be nice to number the carbon atoms. For example, (2S,4R).
2. For characterization of **8**, UV spectrum should be reported. I wonder if this species exists as a hydrate form in solution and undergoes dehydration during the mass spec detection.
3. I feel that the description of the mass detection of m/z 247 may be too long and can be simplified. Details can be described in the supporting document.
4. Page 5, line 9: The authors propose **3** is the nucleophile and **8** is the electrophilic for the annulation reaction. However, it is possible that **8** serves as a nucleophile to initiate the annulation.
5. Page 6 "This result, and the fact that **4**, which would be in equilibrium with an equivalent enol tautomer, does not serve as a co-substrate with **8** (Fig 3c, Fig S6), suggests that condensation of (S)-**3** with **8** likely precedes decarboxylation." I think this is not very strong evidence. For example, it is still possible that the binding of **8** may be required for binding and decarboxylation of **3**.
6. Another possibility is that the alpha-carbon of the imine **8** is first deprotonated and then attacks the ketone of **3** as an initial step of the annulation. The author may want to discuss this possibility.
7. While the purified PtCAL-1a/PtCAL-2a were inactive. The authors need to experimentally investigate the possibility of complexation of these two proteins.

8. Figure S4: The arrow pushing for the second step is not exactly correct. The organic radical intermediate should donate one electron to the ferryl species to generate carbocation. The current drawing implies a OH rebound.

9. Page 6: Figure S6H: The first mechanism is labeled as “condensation-first” mechanism. The first step should be an addition reaction, but not a condensation reaction, at least by the organic chemistry definition. Same for Figure S8F.

10. Page 8, line 35: For better characterization of 13 and 14, UV spectra should be reported. I think the desaturation may first occur either at C8-C15 or C2-C3, which could not be distinguished by mass, but UV spectra will be helpful.

Referee #2 (Remarks to the Author):

This manuscript by Nett et al. reports on the identification and characterization of most of the remaining unknown enzymes involved in the biosynthesis of specialized alkaloids unique to Lycopodeacea plants that harbor a phlegmarane scaffold. The paper is extensive and represents a major advance in plant natural product biochemistry. The key highlight of this paper is the discovery of three carbonic anhydrase (CAH) homologs that are responsible for catalyzing key condensation reactions in the pathway. This is a novel discovery and will have implications for future biosynthetic pathway discovery work in other plant natural product systems. The authors also performed a series of elegant and meticulous experiments to delineate the roles of pathway enzymes in handling alternative intermediates with complex stereochemistry. I also find the portion addressing the evolutionary development of HupA biosynthesis towards higher inhibition potency against the animal acetylcholinesterase interesting.

Major comments:

- Page 5, line 36-40, “To assess this, we produced His-tagged versions of these proteins in *N. benthamiana*, and used Western blotting of different protein fractions (apoplast and cellular) to evaluate PtCAL-1a and PtCAL-2a localization. This confirmed that both enzymes are mainly present within the apoplast (Fig 3b, Fig S6),” This is inconsistent with the western blot data, seems not to be present in the apoplast fraction when it’s expressed alone, but is more present when an untagged PtCAL-1a is co-expressed. This suggests that CAL-1a and CAL-2a form a complex in vivo.

- The authors may try AlphaFold to predict a heterodimer of PtCAL-1a/PtCAL-2a to gain some additional insight into the dimer hypothesis. The sequence/residue analysis shown in Fig. 4B can also be mapped to the structural model.

- The authors hypothesized PtCAL-1a/PtCAL-2a could be posttranslationally modified or require additional co-factors or protein partners. Could these hypotheses be probed by protein mass spec

and/or pulldown experiment since the His-tagged CAL transient expression system has been established in tobacco.

- The title of the paper should specifically refer to huperzine A or alkaloids with phlegmarane scaffold. The term “Neuroactive plant alkaloids” is too broad.

Minor comments:

- Page 1, line 29, “Many of these neuroactive compounds are alkaloids, nitrogen-containing compounds derived predominantly from amino acids, which act by mimicking animal neurotransmitters.” It’s more accurate to say “by targeting the nervous systems of animals”.

- Page 2, line 16, “Perhaps the most well-known member of this alkaloid class is huperzine A (HupA, 17),¹² an acetylcholine mimic that reversibly inhibits acetylcholinesterase (AChE) and which has been explored as a potential dementia treatment.¹³” Check grammar for the last clause.

Referee #3 (Remarks to the Author):

Summary of the key results

This article by Nett, Sattely et al represents a great step forward in the understanding of Lycopodium alkaloid biosynthesis, notably huperzine A, both early and late steps. In their previous work (PNAS 2021), they identified three enzymes from early pathway and three enzymes from the late pathway. In this work they identify a further seven enzymes from the early pathway and three more from the late pathway. Of particular importance are the identification of “neofunctionalised” carbonic anhydrases (CALs) that represent the first enzymes of this class involved in specialised metabolism. Completion of the full huperzine A pathway appears to be one or two enzymes away.

Originality and significance

The work is highly original, reporting previously undescribed enzymes from this important alkaloid pathway. The identification of CALs especially provides wider interest to the work as these activities and roles are unprecedented in (specialised) metabolism, making the work significant. The group have set high standards for themselves with a track record of solving whole pathways, so it is noticeable that the central steps connecting the early and late parts of the pathway remain missing. But this doesn’t detract from the high quality and importance of the rest of the work.

Data & methodology: validity of approach, quality of data, quality of presentation

Overall, the data is highly appropriate and robust. The presentation is generally exceptional and the supplementary figures provide plenty of important additional information. Some important clarifying information is absent from main figures and only present in the text or supplementary. Such information would help readers understand the methods more readily, and be able to judge the results. For example:

- Addition to figure 2 about how compound 9 (and oxidised 9') has been identified. Including clarity, on all figures, about the use of m/z 164 instead of 247 for 9.
- Presentation of full western blots on figure 3
- Clarity in figure 5 as to which peaks have been verified by standard, derived standard (NACs) or through putative m/z
- Treatment of supplementary figures – refer to specific panels of SFs in text, refer to relevant SFs in main Figure legends, order SFs chronologically (with NMRs at end if preferred).

Appropriate use of statistics and treatment of uncertainties

This is not typically a major concern in this field, so statistical analysis on all data is not required. Some t-test p-values have been calculated. Overall, interpretation of these are suitable, but could be refined a little. There are a couple of examples of p-values greater than 0.05 on figures but the authors go ahead and describe the differences without allusion to the $p\text{-value} > 0.05$. Examples are on Figure S5K, S11E. Here clarity in the text about the p-value would be best –do describe the differences but state that p-value (based on triplicates) was > 0.05 . As described above, being very clear about compound identity on main figures is important too.

Conclusions: robustness, validity, reliability

The core of the work – gene discovery, Nicotiana expression and LC-MS analysis via EICs and MS/MS – is highly robust. The main conclusions – newly described activities of ten enzymes involved in alkaloid biosynthesis are well supported. Details on stereoselectivity of key steps are well dealt with, with good chiral analysis. The identification of intermediate compounds in the native plant was impressive. Identification of 8 and 9 are challenging but sufficient, provided the information is clearly presented to the reader.

Details about CAL remain slightly unclear, but they seem to be highly challenging enzymes to work with so this is understandable. It is a shame that in vitro they did not work, and the apoplast isolation was very clever. However, the relationship between CAL1 and CAL2 remains unclear with respect to oligomerisation, activity or localisation. I do not question overall CAL activity observation, but the details on their cooperativity are ambiguous. The phylogenetic/residue analysis was very good, as was the zinc cofactor experiments. There is plenty to investigate about these enzymes in the future.

Suggested improvements: experiments, data for possible revision

Please see below for in depth discussion of the manuscript.

Clarity and context: lucidity of abstract/summary, appropriateness of abstract, introduction and conclusions

Overall very well written. I recommend some adjustments to the discussion, to reduce speculation, particularly with respect to the PIKS mechanism and evolutionary statements.

Comments

Abstract – page 1 line 21. “Cycloaddition” – as presented there is no evidence of cycloaddition in this pathway. Please remove reference to this reaction.

Page 2, line 5 – “...derived from unique chemical transformations that have not previously been observed in plants”. This reads strangely - how is the chemical transformation known if it has never been observed? Surely, this refers to “not previously been characterised” or something around the knowledge of the enzyme that catalyses it.

Page 2, line 9 – “unique, undefined chemical transformations”. It is unclear to me what this means – could it be reworded to clarify. “Unique” in what way? “Undefined” meaning uncharacterised?

Page 2, line 10 – “inability” is too strong – perhaps “challenge”. I’d also add a point that this relates to the fact that these scaffold forming steps appear to be redox-neutral and require just acid/base catalysis so no co-factors are clearly associated. This is another factor that makes it difficult to predict enzyme class.

Page 2, line 22 – what does “novel” enzymes mean? They are not novel to nature, just unknown to us. Can this be rephrased to improve accuracy of language – “previously uncharacterised enzyme classes”.

Figure S1 – C – improve arrows. Ideally, we would see decarboxylation arrows, and we should see e-coming from enamine-N and going up into carboxyl on phlegmarine-type to lycodane type step.

Figure 1B – “3 not known to act in metabolism” – I assume these include the carbonic anhydrases. Does it depend how we define metabolism - is photosynthesis metabolism? Yes. The authors mean “specialised” or “secondary” metabolism or biosynthesis so should be rephrased to reflect this.

Figure 2 – could you add OR between SDR-1 and SDR-2 to emphasise these were described separately

Page 3 line 34 and Fig S2. The results are very nice. The alternative possibility here is different stereoselectivity, which although ruled out by results is not addressed. Can you note somewhere that the absolute stereoselectivity of the carbonyl reduction is conserved across both SDRs and substrates but the enantioselectivity of the substrate is different.

CYP78. Page 4 para1. Could 184.1332 and 124.1221 represent in-source fragmentation of 184.1332? Can you clarify why you think there are 2 peaks? Clarity about RT would help. Figure S4 – B - can we see the 2 peaks overlaid or arrayed vertically to judge the RT difference? D – which 6 diastereomers are being fed here? Why do we not see 2 peaks? F – there is the possibility that the ion abundance of 124 also represents a spontaneous deacylation – what do the other ion abundances look like over the pH range (substrate and 184)? G – what is the evidence that 7 to 8 is catalysed by CYP and not spontaneous/buffer catalysed? H –this mechanism doesn't work for me – the simplest option is typical hydroxylation of that position with P450 O-rebound mechanism and elimination/dehydration to the imine. Here it appears you are suggesting a SET on the second step to directly form the positive charge? If so, where is the second proton coming from on the P450? Please revise.

Page 5 para 2. Fig 2 and Fig S5. Why are you generally not showing 247 rather than 164? This would be more obvious. Is it because the peak is too small for good signal? In the main Fig 2 247 would be the ideal mass to show. Otherwise, you could depict on the structure of 9 what fragment you are observing. Figure S5. This is sufficient but you could have done MS³ from M+H 247 to 164. Similarly, do you see 247 in the clubmoss extract?

Page 5, line 3 – where are we supposed to find the second 247 peak? And matching MS² pattern? Is this Fig S5 L? If so, this is EIC 164 and not 247. In addition, the decrease in S-4 is not significant – this needs to be referred to in the text as a caveat if the statement is to remain.

Page 5, Line 15 – this information on the oxidised compound is buried too deep the SI. It should be referred to in Figure 2 or 3 or in Figure S6, not in Fig S21-S24. In addition, the figure must be properly referred to in the text. Also if the NMR spectra is confident on the structure of 9' then this is important evidence for your compound 9 identity so should be more prominently presented.

Page 5 – line 38 – remove “mainly” present as no comparative assessment was made. They are present in the apoplast. But, CAL2 expressed alone is not in the apoplast hence the whole cell extract which has strong signal for CAL2 but not in the apoplast. It seems maybe that CAL2 is expressed OK but not trafficked to correct compartment?

Figure 3 A – be clear about what steps you think are enzyme catalysed – either by drawing a box around the steps or labelling each straight-reaction arrow clearly. I.e. as it is drawn only the Nu-attack is CAL catalysed – what about decarboxylation, cyclisation and dehydration etc.

Similarly Figure 3G – which steps are you proposing CAL3 catalyses?

Fig 3B – highly selective blot – can you include all blots including whole cell extract – I am not sure that CAL2 is poorly expressed but it appear to not be trafficked correctly.

Page 6 line 22 = “... could also leverage CO₂ release to drive the reaction equilibrium.” I do not understand what this means – perhaps “drive to completion” could be a better way of phrasing. However, CO₂ release occurs in the decarboxylation-first mechanism too, so I am not sure how the later CO₂ release is energetically preferable. Please clarify

Figure S6 G – this is repeated in Figure 3E.

Page 7, line 21 – What on Fig 4 is this referring to? Unclear. This should not be Fig S33 but earlier in the SI. Also, I do not understand why you need to evoke mechanism 3 instead of mechanism 2 for PIKS. Enzymes can act non-stereoselectively so you can surely just keep mechanism 2 if you need to evoke PIKS catalysis. Mechanism 3 is unlikely as the open ring form is very unflavoured in water. There is no evidence for this at all, in fact I find the PIKS mechanism discussion a little speculative and does not add much to the work.

Page 8, line 25 - 8,15-double bond can apparently be found numbered in Fig 1A. Does this mean atom numbers? If so, they are not visible on the figure.

Page 8 line 31 – Which Supplementary information files?

Page line 38 . “If these were the oxidations...” this reads a bit strangely, please rephrase.

Figure S15 – should this be positioned earlier as it is referred to earlier in the paper?

Figure 5 – impressive bank of reactions. Please clarify on each plot whether the compound ID was validated by (i) comparison to standard, (ii) comparison to related standard (i.e. NAc derivatives) or (iii) putative ID.

Page 9, line 20-23. This is an interesting observation but the interpretation is speculative. Whilst yes I do think it shows a “metabolic SAR among the alkaloids” the evolutionary origin is unclear and I don’t think this should be stated as the only interpretation. You are taking the “forward” model of step-by-step evolution. There is an alternative “retrograde” model where the bioactive product huperzine was present as one product of a complex network catalysed by promiscuous enzyme ancestors. Only after it became selected for did the more specific / selective enzyme steps emerge. This fits with your network concept too. So practically, yes, the IC50 improves along the pathway but the evolutionary origin is unknown. Interestingly your CAL enzyme identification matches the “patchwork” ideas of metabolic evolution too.

Page 9 line 42 – What is the evidence, aside from your own identification of CALs, that they have wider roles? You have not identified many outside clubmoss with different Zn-motifs, for example.

Author Rebuttals to Initial Comments:

Response to referee comments:

We thank the referees for their thorough reading and assessment of our manuscript, and we appreciate the constructive critique and feedback. Below, we respond to each item raised, point-by-point. When necessary, we include major text changes below our response, with major/critical edits highlighted. We note that all major text changes are also highlighted in the resubmitted manuscript document.

Referee #1 (Remarks to the Author):

Nett et al. describe the biosynthesis of Lycopodium alkaloids. Using transcriptomic and gene screening, the authors identified enzymes that catalyze the Lycopodium scaffold formation and modification. Surprisingly, the key annulation reaction between two units of C8 components is catalyzed by the combination of two carbonic anhydrase-like proteins. A substantial amount of data is provided in this manuscript to support the proposed biosynthetic pathway and enzyme functions. This is a topnotch study and should have a broad impact in the scientific community. I believe this work would be suitable for acceptance in this journal after addressing some concerns listed below.

1. For the stereochemical notion, it would be nice to number the carbon atoms. For example, (2S,4R).

We have added the suggested carbon numbering to compound descriptions in both the manuscript and within each relevant figure, as suggested. Additionally, we have added select carbon numbers to main Figure 1a for clarity, and full numbering of the 8-carbon scaffold in Figure S1.

2. For characterization of **8**, UV spectrum should be reported. I wonder if this species exists as a hydrate from in solution and undergoes dehydration during the mass spec detection.

We now provide a UV spectrum for **8** (Fig S4d). We also provide one for **9** (Fig S5i), as we are proposing this compound to have an alpha/beta unsaturated imine as well. Each of these was measured using the DAD that is in-line with our LC-MS, and the provided spectra were generated by extracting from background signal in Agilent Qualitative analysis software.

This analysis demonstrates that **8** has an absorbance maximum at ~243 nm, which is in the expected range for this alpha/beta unsaturated imine under the acidic conditions of our LC-MS solvent system (Kosower & Sorensen, 1963).

We further also show that **9** exhibits an absorbance maximum at ~273 nm, which is also in the expected range for the proposed alpha/beta unsaturated imine (again under acidic conditions) we propose to be in **9**. In addition to the data in Figs S4 and S5, we have also added minor text changes to the manuscript indicating that UV analysis was also included for structural assessments of these molecules, as well as a description of this analysis in the Methods.

3. I feel that the description of the mass detection of m/z 247 may be too long and can be simplified. Details can be described in the supporting document.

We understand this concern, but we feel that the structural confirmation of m/z 247 (**9**) is imperative to our story - in line with this, another referee has asked for this description and corresponding data to be featured more prominently. To help streamline this section, and to limit confusion, we now use **9** instead of "mz247" throughout the results (shown below, highlighted), and we make sure to convey that we initially did not know the structure of **9** until subsequent analyses. Additionally, in response to another referee, we have added some clarifying text on the observed ion adducts associated with **9** (shown below, highlighted). Finally, we have gone through this section carefully to try and clarify our language as much as possible, though most of these changes are minor.

"We note that of the three MS adducts observed for this molecule, the m/z 164 ion was the most abundant, and therefore this was used as a diagnostic ion for all following analyses. Subsequent MS/MS fragmentation of both the parent ion (m/z 247) and the in-source fragment (m/z 164) suggested that this new compound (designated as **9**, though the structure was not immediately evident) possesses a phlegmarane-type scaffold (Fig S5f-h), and UV analysis supported the presence of an α/β -unsaturated imine (Fig S5i)."

4. Page 5, line 9: The authors propose **3** is the nucleophile and **8** is the electrophilic for the annulation reaction. However, it is possible that **8** serves as a nucleophile to initiate the annulation.

We have added two sentences explaining that this alternative order of events is plausible, and that our current work is not definitive in its support of the mechanism for phlegmarane ring formation.

"Though we favor the role of (S)-**3** as the initial nucleophile attacking the **8** electrophile in this reaction, we note that an alternative sequence of bond formation is also plausible. For example, formation of the enamine tautomer of **8** could allow for this molecule to serve as the initial nucleophile, wherein the enamine would attack the carbonyl of (S)-**3** first, followed by decarboxylative condensation to generate the final phlegmarane scaffold."

5. Page 6 "This result, and the fact that **4**, which would be in equilibrium with an equivalent enol tautomer, does not serve as a co-substrate with **8** (Fig 3c, Fig S6), suggests that condensation of (S)-**3** with **8** likely precedes decarboxylation." I think this is not very strong evidence. For example, it is still possible that the binding of **8** may be required for binding and decarboxylation of **3**.

We have added a sentence to clarify that our results do not rule out this possibility.

"This mechanism implies that the CAL enzymes may enhance formation of the enolate tautomer of **3**, which could serve as the requisite nucleophile (Fig S7g); however, our results do not rule out the possibility that binding of **8** is required for the decarboxylation of (S)-**3** to occur."

6. Another possibility is that the alpha-carbon of the imine **8** is first deprotonated and then attacks the ketone of **3** as an initial step of the annulation. The author may want to discuss this possibility.

We have addressed this within our response to point #4, as described above.

7. While the purified PtCAL-1a/PtCAL-2a were inactive. The authors need to experimentally investigate the possibility of complexation of these two proteins.

Understanding why CAL-1 and CAL-2 co-expression is necessary to observe enzyme activity is a notable, unanswered question regarding these neofunctionalized proteins, and we agree that it warrants further exploration. As suggested, one possible explanation is the formation of a heterodimer between CAL-1 and CAL-2; to explore this hypothesis, we have performed the following experiments/analyses:

1. At the request of referee #2, we have used AlphaFold-Multimer to investigate potential heterodimer formation between CAL-1a and CAL-2a. This resulted in a predicted interaction between these proteins with moderate to low confidence. We note that the AlphaFold2 model has very low confidence in the structure/positioning of the N-terminal regions of these proteins, and relatively low confidence in the residues at the predicted interface between CAL-1a and CAL-2a, so we interpret these data cautiously. The results of this analysis are now shown in Figures S11 and S12, and are mentioned briefly in the results, but we do not make strong conclusions from these data, other than that they provide modest support for a potential interaction.
2. We attempted 'pull-down' assays to probe for interactions between CAL-1 and CAL-2. To do this, we co-expressed a 6xHis-tagged version of CAL-1 with a non-tagged version of CAL-2 in *N. benthamiana*. As negative controls, we tested **a)** CAL-1-6xHis alone, **b)** CAL-2-6xHis alone, and **c)** CAL-1 and CAL-2 expressed together (both untagged) to control for non-specific binding during purification. We performed Ni-NTA purification with apoplast extracts from the leaves expressing these proteins, then subsequent *in vitro* enzyme assays with the purified fraction, much as previously described in the manuscript. Enzyme activity was observed in the Ni-NTA purified fraction from CAL-1-6xHis+CAL-2 (untagged), but not in those of CAL-1-6xHis or CAL-2-6xHis when expressed alone, which is suggestive of CAL-2 co-purifying with CAL-1. Further protein mass spectrometry of this purified fraction confirmed that CAL-2 could be identified along with CAL-1-6xHis in this sample. Although this was highly suggestive of an interaction between CAL-1 and CAL-2, the control with co-expressed, untagged CAL-1+CAL-2 (treated with the same Ni-NTA purification conditions) also retained nearly equivalent enzyme activity, indicating non-specific binding of these proteins to the Ni-NTA resin. Thus, we cannot conclude from our pull-down assays that the co-purification of CAL-2 is due to a specific interaction with CAL-1. Given the outcome of this control experiment, these data have not been added to the manuscript.
3. We attempted a co-immunoprecipitation assay in which CAL-1 was His-tagged and CAL-2 was tagged with a 1D4 tag (each C-terminal). These were co-expressed in *N. benthamiana* as usual, and isolated protein was affinity purified with appropriate beads for each affinity tag, then probed via Western blot. Ultimately, we were unable to detect the corresponding partner protein in these assays, and it seemed that the addition of the 1D4 tag was deleterious to CAL-2 expression/production.

Although these two initial experimental efforts did not provide evidence for an interaction of CAL-1 and CAL-2 that is required for enzymatic activity, we are excited to continue investigating this question in the context of future work. Given that a possible interaction between CAL proteins may be transient and could be required at different stages or for different functions (enzyme activity, protein processing, and/or protein transport to the apoplast), we anticipate this study will require thorough investigation that will go beyond the scope of the current manuscript. Furthermore, evidence for such an interaction will not change the key conclusions presented in this work (mainly the discovery of a novel enzyme class for the biosynthesis of plant alkaloids).

With that said, we have added additional text to the results describing the results from AlphaFold-Multimer (shown below), though we refrain from making anything more than speculative comments on the possible heterodimerization of CAL-1 and CAL-2. Additionally, we have added text to the results that summarizes the current unknowns and future directions for understanding these proteins (also below).

"Beyond understanding the detailed catalytic mechanisms of the CALs, additional work will be necessary to establish the reasons(s) for *PtCAL-1/PtCAL-2* co-dependence. Although computational modeling⁴⁷ predicts *PtCAL-1a* and *PtCAL-2a* to interact with a moderate level of confidence (**Fig S12**), *de novo* prediction of protein heterodimers remains challenging without experimental validation. Thus, it will be necessary in future work to rigorously assess potential interaction between these two proteins, as well as how this interaction may impact function. For example, although we have shown that the co-expression of *PtCAL-1* critically impacts the localization and post-translational modification of *PtCAL-2*, it is not yet clear how *PtCAL-1* may effect this change, and additional questions remain as to how these proteins may be cooperating to carry out phlegmarane scaffold formation. Thus, these CALs will provide an exciting model not only for investigating the catalytic mechanisms for a novel sub-class of enzymes, but also for understanding the nuanced roles for transport and protein cooperativity within specialized metabolism."

8. Figure S4: The arrow pushing for the second step is not exactly correct. The organic radical intermediate should donate one electron to the ferryl species to generate carbocation. The current drawing implies a OH rebound.

We have amended this scheme to more accurately reflect possible mechanisms for this reaction. Based upon your comment, as well as that of referee #3, we have provided two possibilities for how the initial imine formation occurs.

9. Page 6: Figure S6H: The first mechanism is labeled as "condensation-first" mechanism. The first step should be an addition reaction, but not a condensation reaction, at least by the organic chemistry definition. Same for Figure S8F.

We have made the change from 'condensation-first' to 'addition-first' within these figures, as well as within the text description of these mechanistic proposals.

10. Page 8, line 35: For better characterization of **13** and **14**, UV spectra should be reported. I think the

desaturation may first occur either at C8-C15 or C2-C3, which could not be distinguished by mass, but UV spectra will be helpful.

While UV spectra would provide additional evidence for the structural features of **13** and **14**, we note that **a)** the structure of **14** is confirmed via LC-MS comparison to an authentic standard and **b)** the specific desaturation at C2-C3 has previously been established (in our prior publication, ref. 22) via reactions that demonstrate conversion of **18** into **19** via the activity of 2OGD-3 (also with authentic standards for verification).

The confirmed structure of **14** verifies that 2OGD-5 is installing the double bond in the C8-C15 location. Moreover, we show that if 2OGD-5 is omitted from the reactions (Figure 5b) we generate the full series of 8,15-dihydro congeners (**11**, **18**, **19**, **20**, **21**), which were all confirmed with authentic standards (with the exception of **11**, but this structure can be inferred based on the structure of downstream products). Thus, our current data provide good confidence in the regioselectivity of the desaturations catalyzed by 2OGD-3 and 2OGD-5.

Referee #2 (Remarks to the Author):

This manuscript by Nett et al. reports on the identification and characterization of most of the remaining unknown enzymes involved in the biosynthesis of specialized alkaloids unique to Lycopodeacea plants that harbor a phlegmarane scaffold. The paper is extensive and represents a major advance in plant natural product biochemistry. The key highlight of this paper is the discovery of three carbonic anhydrase (CAH) homologs that are responsible for catalyzing key condensation reactions in the pathway. This is a novel discovery and will have implications for future biosynthetic pathway discovery work in other plant natural product systems. The authors also performed a series of elegant and meticulous experiments to delineate the roles of pathway enzymes in handling alternative intermediates with complex stereochemistry. I also find the portion addressing the evolutionary development of HupA biosynthesis towards higher inhibition potency against the animal acetylcholinesterase interesting.

Major comments:

- Page 5, line 36-40, "To assess this, we produced His-tagged versions of these proteins in *N. benthamiana*, and used Western blotting of different protein fractions (apoplast and cellular) to evaluate PtCAL-1a and PtCAL-2a localization. This confirmed that both enzymes are mainly present within the apoplast (Fig 3b, Fig S6)," This is inconsistent with the western blot data, seems not to be present in the apoplast fraction when it's expressed alone, but is more present when an untagged PtCAL-1a is co-expressed. This suggests that CAL-1a and CAL-2a form a complex in vivo.

We have altered the text in this section to more accurately convey that the localization of CAL-2 appears to depend on CAL-1 co-expression (as shown in the excerpt of the main text below, with changes highlighted). The possibility of a CAL-1/CAL-2 complex is discussed further with your next point, but in addition to this, we also speculate that CAL-1 may somehow be involved in the proper processing of CAL-2 such that it is properly secreted/trafficked to the apoplast (text for this also included below). Finally, in line with the request of referee #3, we have moved the full Western blots to Figure 3b.

"To assess this, we produced His-tagged versions of these proteins in *N. benthamiana*, and used Western blotting of different protein fractions (apoplast and cellular) to evaluate *PtCAL-1a* and *PtCAL-2a* localization. This demonstrated that both proteins can be found in the apoplast within this heterologous system, but also that this localization is impacted by their co-expression (**Fig 3b**). In particular, while *PtCAL-1a* exhibited apoplastic localization independently of co-expression with *PtCAL-2a*, very little *PtCAL-2a* protein could be detected in the apoplast when it was expressed alone, and it instead appeared to be mainly within the intracellular fraction, which contains both cytosolic and organellar proteins (**Fig 3b**). However, upon co-expression of *PtCAL-1a*, apoplastic *PtCAL-2a* was readily detected, and appeared to exhibit post-translational modifications of an unknown nature (**Fig 3b**). These results suggest that *PtCAL-1a* may have a critical role in the proper post-translational modification and/or trafficking of *PtCAL-2a*, though additional work will be necessary to determine the mechanism by which this occurs. Beyond providing details on localization, this information was critical for enzyme assay development since the pH of the apoplast is typically relatively low (~pH 5),³⁹ which suggested that these proteins may only exhibit function at a lower pH. Additionally, apoplast extracts can be readily isolated from *N. benthamiana* leaves expressing these proteins,³⁹ thereby providing a potential means to evaluate CAL protein outside of the living plant system."

- The authors may try AlphaFold to predict a heterodimer of *PtCAL-1a*/*PtCAL-2a* to gain some additional insight into the dimer hypothesis. The sequence/residue analysis shown in Fig. 4B can also be mapped to the structural model.

As suggested (and summarized above for referee #1), we have used AlphaFold-Multimer to investigate the potential for heterodimer formation between *CAL-1a* and *CAL-2a*. This resulted in a predicted interaction between these proteins with moderate to low confidence; however, we note that the AlphaFold2 model has very low confidence in the structure/positioning of the N-terminal regions of these proteins, and relatively low confidence in the residues at the predicted interface between *CAL-1a* and *CAL-2a*. These results are now shown in Figures S11 (structural models of individual CALs) and S12 (heterodimer analysis).

As 'negative controls' for this analysis, we also predicted heterodimers (via AlphaFold-Multimer) for both *CAL-1a* and *CAL-2a* with human *CA2*, *PtCAL-3*, and another CAH family protein from *P. tetrastrichus* that appears to be a canonical CAH (based upon its conserved active site, see CAH-VII in Fig S10). As would be expected, no interaction was predicted for either of the CALs with human *CA2* or *PtCAH-VII*, though each exhibited a reasonably high confidence for interacting with *CAL-3*. It is possible that the CALs can all interact with one another, since we have shown them to be involved in the same pathway, and to exhibit similar apoplast localization. However, given the collection of data, we do not feel we can make strong conclusions on the heterodimerization of *CAL-1*/*CAL-2* beyond that it is possible and that AlphaFold-Multimer predictions are somewhat suggestive of this. As such, we have added additional text to the results describing the results from AlphaFold-Multimer (shown below), though we refrain from making anything more than speculative comments on the possible heterodimerization of *CAL-1* and *CAL-2*. Additionally, we have added text to the results that summarizes the current unknowns and future directions for understanding these proteins (also shown below).

"Beyond understanding the detailed catalytic mechanisms of the CALs, additional work will be necessary to establish the reasons(s) for *Pt*CAL-1/*Pt*CAL-2 co-dependence. Although computational modeling⁴⁷ predicts *Pt*CAL-1a and *Pt*CAL-2a to interact with a moderate level of confidence (**Fig S12**), *de novo* prediction of protein heterodimers remains challenging without experimental validation. Thus, it will be necessary in future work to rigorously assess potential interaction between these two proteins, as well as how this interaction may impact function. For example, although we have shown that the co-expression of *Pt*CAL-1 critically impacts the localization and post-translational modification of *Pt*CAL-2, it is not yet clear how *Pt*CAL-1 may effect this change, and additional questions remain as to how these proteins may be cooperating to carry out phlegmarane scaffold formation. Thus, these CALs will provide an exciting model not only for investigating the catalytic mechanisms for a novel sub-class of enzymes, but also for understanding the nuanced roles for transport and protein cooperativity within specialized metabolism."

In addition to the computational modeling, we also performed multiple assays that sought to experimentally validate a possible interaction between these proteins, as detailed in our response to referee #1 and included here again for convenience:

1. We attempted 'pull-down' assays to probe for interactions between CAL-1 and CAL-2. To do this, we co-expressed a 6xHis-tagged version of CAL-1 with a non-tagged version of CAL-2 in *N. benthamiana*. As negative controls, we tested **a)** CAL-1-6xHis alone, **b)** CAL-2-6xHis alone, and **c)** CAL-1 and CAL-2 expressed together (both untagged) to control for non-specific binding during purification. We performed Ni-NTA purification with apoplast extracts from the leaves expressing these proteins, then subsequent *in vitro* enzyme assays with the purified fraction, much as previously described in the manuscript. Enzyme activity was observed in the Ni-NTA purified fraction from CAL-1-6xHis+CAL-2 (untagged), but not in those of CAL-1-6xHis or CAL-2-6xHis when expressed alone, which is suggestive of CAL-2 co-purifying with CAL-1. Further protein mass spectrometry of this purified fraction confirmed that CAL-2 could be identified along with CAL-1-6xHis in this sample. Although this was highly suggestive of an interaction between CAL-1 and CAL-2, the control with co-expressed, untagged CAL-1+CAL-2 (treated with the same Ni-NTA purification conditions) also retained nearly equivalent enzyme activity, indicating non-specific binding of these proteins to the Ni-NTA resin. Thus, we cannot conclude from our pull-down assays that the co-purification of CAL-2 is due to a specific interaction with CAL-1. Given the outcome of this control experiment, these data have not been added to the manuscript.
2. We attempted a co-immunoprecipitation assay in which CAL-1 was His-tagged and CAL-2 was tagged with a 1D4 tag (each C-terminal). These were co-expressed in *N. benthamiana* as usual, and isolated protein was affinity purified with appropriate beads for each affinity tag, then probed via Western blot. Ultimately, we were unable to detect the corresponding partner protein in these assays, and it seemed that the addition of the 1D4 tag was deleterious to CAL-2 expression/production.

Because the results of these experiments were inconclusive, we have opted to present only the AlphaFold-Multimer results as we have observed them, but with the caveats that any protein-protein interaction needs to be rigorously shown experimentally in future work. As detailed in our response to referee #1, we agree that the mechanism for the co-functionality of CAL-1 and CAL-2 is a major outstanding question. However,

we expect this to require a significant amount of additional, rigorous experimentation, including multiple different assays and methods of verification, as we would not feel comfortable claiming a protein-protein interaction (or other fundamental mechanism) without several, solid lines of evidence.

- The authors hypothesized PtCAL-1a/PtCAL-2a could be posttranslationally modified or require additional co-factors or protein partners. Could these hypotheses be probed by protein mass spec and/or pulldown experiment since the His-tagged CAL transient expression system has been established in tobacco.

Please refer to our response to the previous question for our attempts to address this comment using a pulldown/co-purification approach. We are quite interested in eventually understanding the specific, mechanistic nature of the requisite CAL-1/CAL-2 co-expression, as well as characterizing any potential post-translational modifications that occur on CAL-2 upon co-expression with CAL-1. Unfortunately, the two sets of new experiments we have described above indicate that conclusive evidence for a possible protein-protein interaction will require extensive in-depth investigation. Given these new data, we feel that proper and robust biochemical analysis of the CALs is beyond the scope of this manuscript. As mentioned earlier, we have added a paragraph to the end of our description of the CALs that highlights the unknowns that still exist in our understanding of these proteins.

- The title of the paper should specifically refer to huperzine A or alkaloids with phlegmarane scaffold. The term "Neuroactive plant alkaloids" is too broad.

We have altered the title to include 'Lycopodium alkaloids' instead of 'plant alkaloids'.

Minor comments:

- Page 1, line 29, "Many of these neuroactive compounds are alkaloids, nitrogen-containing compounds derived predominantly from amino acids, which act by mimicking animal neurotransmitters." It's more accurate to say "by targeting the nervous systems of animals".

This sentence has been altered in line with this suggestion.

"Many of these neuroactive compounds are alkaloids - nitrogen-containing compounds derived predominantly from amino acids – that act like neurotransmitter mimics to affect animal nervous systems.²"

- Page 2, line 16, "Perhaps the most well-known member of this alkaloid class is huperzine A (HupA, 17),¹² an acetylcholine mimic that reversibly inhibits acetylcholinesterase (AChE) and which has been explored as a potential dementia treatment.¹³" Check grammar for the last clause.

We have restructured/expanded this section for clarity.

"Perhaps the most well-known member of this alkaloid class is huperzine A (HupA, **17**),¹² an acetylcholine mimic that reversibly inhibits acetylcholinesterase (AChE), an important enzyme at the neural synapse. This

pharmaceutical activity has led to interest in the use of **17** as a potential treatment for the symptoms of dementia,¹³ and elucidating its biosynthesis offers the possibility for the engineered production of this molecule, which has historically been non-sustainably sourced from wild *Huperzia* plants.¹⁴

Referee #3 (Remarks to the Author):

Summary of the key results

This article by Nett, Sattely et al represents a great step forward in the understanding of Lycopodium alkaloid biosynthesis, notably huperzine A, both early and late steps. In their previous work (PNAS 2021), they identified three enzymes from early pathway and three enzymes from the late pathway. In this work they identify a further seven enzymes from the early pathway and three more from the late pathway. Of particular importance are the identification of "neofunctionalised" carbonic anhydrases (CALs) that represent the first enzymes of this class involved in specialised metabolism. Completion of the full huperzine A pathway appears to be one or two enzymes away.

Originality and significance

The work is highly original, reporting previously undescribed enzymes from this important alkaloid pathway. The identification of CALs especially provides wider interest to the work as these activities and roles are unprecedented in (specialised) metabolism, making the work significant. The group have set high standards for themselves with a track record of solving whole pathways, so it is noticeable that the central steps connecting the early and late parts of the pathway remain missing. But this doesn't detract from the high quality and importance of the rest of the work.

Data & methodology: validity of approach, quality of data, quality of presentation

Overall, the data is highly appropriate and robust. The presentation is generally exceptional and the supplementary figures provide plenty of important additional information. Some important clarifying information is absent from main figures and only present in the text or supplementary. Such information would help readers understand the methods more readily, and be able to judge the results. For example:

- Addition to figure 2 about how compound **9** (and oxidised **9'**) has been identified. Including clarity, on all figures, about the use of m/z 164 instead of 247 for **9**.

We have added additional text to the caption of Figure 2 that highlights the data used to infer the structure of **9**, as well as **8**. We have also added clarification to the Results (see text excerpt below) and the caption of Figure 2 that describe the use of m/z 164 as a diagnostic ion for routine detection of **9**. Finally, in response to the comments of referee #1, we now include a UV spectrum of **9** in the SI, which further supports the alpha/beta unsaturated imine that we propose to be in this structure.

"We note that of the three MS adducts observed for this molecule, the m/z 164 ion was the most abundant, and therefore this was used as a diagnostic ion for all following analyses."

- Presentation of full western blots on figure 3

We have updated our language within the Results to more accurately describe the data of our Western blots, and we now include the full Western blots in Figure 3.

- Clarity in figure 5 as to which peaks have been verified by standard, derived standard (NAcs) or through putative m/z

We have added a sentence in the figure caption indicating that all molecules with common names in this figure have been verified with an authentic standard. We have also added an additional sentence that clarifies the proposed structures for 'un-named' intermediates and directs the reader to the SI for the supporting structural evidence. Finally, we have added 'proposed' under each structure in Figure 5 that was not verified with an authentic standard.

- Treatment of supplementary figures – refer to specific panels of SFs in text, refer to relevant SFs in main Figure legends, order SFs chronologically (with NMRs at end if preferred).

We have made the suggested changes for all figure references.

Appropriate use of statistics and treatment of uncertainties

This is not typically a major concern in this field, so statistical analysis on all data is not required. Some t-test p-values have been calculated. Overall, interpretation of these are suitable, but could be refined a little. There are a couple of examples of p-values greater than 0.05 on figures but the authors go ahead and describe the differences without allusion to the $p > 0.05$. Examples are on Figure S5K, S11E. Here clarity in the text about the p-value would be best –do describe the differences but state that p-value (based on triplicates) was > 0.05 . As described above, being very clear about compound identity on main figures is important too.

We have added p-values within the text of the Results for any quantitative comparisons that were made in the manuscript. Also, as discussed with other points, we have adjusted the main figures to indicate the degree of confidence for structural confirmation with each compound.

Conclusions: robustness, validity, reliability

The core of the work – gene discovery, Nicotiana expression and LC-MS analysis via EICs and MS/MS – is highly robust. The main conclusions – newly described activities of ten enzymes involved in alkaloid biosynthesis are well supported. Details on stereoselectivity of key steps are well dealt with, with good chiral analysis. The identification of intermediate compounds in the native plant was impressive. Identification of **8** and **9** are challenging but sufficient, provided the information is clearly presented to the reader.

Details about CAL remain slightly unclear, but they seem to be highly challenging enzymes to work with so this is understandable. It is a shame that in vitro they did not work, and the apoplast isolation was very clever. However, the relationship between CAL1 and CAL2 remains unclear with respect to oligomerisation, activity or localisation. I do not question overall CAL activity observation, but the details on their cooperativity are ambiguous. The phylogenetic/residue analysis was very good, as was the zinc cofactor experiments. There is plenty to investigate about these enzymes in the future.

Although ultimately inconclusive, we point this referee to our response to referees #1 and #2, wherein we describe computational modeling and attempted experiments that sought to provide preliminary info on the nature of the co-functionality of CAL-1 and CAL-2. Also, as described later, we have adjusted our language of CAL localization to provide a more thorough and nuanced presentation of these data.

Suggested improvements: experiments, data for possible revision
Please see below for in depth discussion of the manuscript.

Clarity and context: lucidity of abstract/summary, appropriateness of abstract, introduction and conclusions

Overall very well written. I recommend some adjustments to the discussion, to reduce speculation, particularly with respect to the PIKS mechanism and evolutionary statements.

Comments

Abstract – page 1 line 21. "Cycloaddition" – as presented there is no evidence of cycloaddition in this pathway. Please remove reference to this reaction.

We have removed this from the abstract and instead refer to this reaction as 'bicyclic scaffold generation'.

Page 2, line 5 – "...derived from unique chemical transformations that have not previously been observed in plants". This reads strangely - how is the chemical transformation known if it has never been observed? Surely, this refers to "not previously been characterised" or something around the knowledge of the enzyme that catalyses it.

As suggested, this was our original intent. We have re-structured this section in line with this comment, as well as your two subsequent comments (see text below, with changes highlighted).

"Furthermore, many classes of plant alkaloids are derived through chemical transformations for which there is no known biosynthetic precedent. This is exemplified within the lysine-derived quinolizidine and Lycopodium alkaloids, which serve as the precursors for hundreds of bioactive compounds,⁷ and whose scaffolds are thought to be constructed via reactions for which no enzyme catalyst has yet been observed in nature.^{8,9} This challenge to readily predict enzymes that build alkaloid scaffolds confounds the rapid

elucidation of biosynthetic pathways, and suggests that there are novel enzyme classes within plant metabolism yet to be identified.”

Page 2, line 9 – “unique, undefined chemical transformations”. It is unclear to me what this means – could it be reworded to clarify. “Unique” in what way? “Undefined” meaning uncharacterised?

We have altered this sentence to more clearly convey our intent. Please see the previous point for the corresponding adjustments.

Page 2, line 10 – “inability” is too strong – perhaps “challenge”. I’d also add a point that this relates to the fact that these scaffold forming steps appear to be redox-neutral and require just acid/base catalysis so no co-factors are clearly associated. This is another factor that makes it difficult to predict enzyme class.

We have changed ‘inability’ to ‘challenge’ as suggested. We completely agree that redox neutral reactions are particularly challenging in this case, but due to space constraints, we have decided not to include this point. See earlier point for text change highlights.

Page 2, line 22 – what does “novel” enzymes mean? They are not novel to nature, just unknown to us. Can this be rephrased to improve accuracy of language – “previously uncharacterised enzyme classes”.

We have altered ‘novel enzymes’ to ‘previously undescribed enzyme classes’.

Figure S1 – C – improve arrows. Ideally, we would see decarboxylation arrows, and we should see e-coming from enamine-N and going up into carboxyl on phlegmarine-type to lycodane type step.

We have included the suggested changes.

Figure 1B – “3 not known to act in metabolism” – I assume these include the carbonic anhydrases. Does it depend how we define metabolism - is photosynthesis metabolism? Yes. The authors mean “specialised” or “secondary” metabolism or biosynthesis so should be rephrased to reflect this.

This has been altered to ‘specialized metabolism’.

Figure 2 – could you add OR between SDR-1 and SDR-2 to emphasise these were described separately Page 3 line 34 and Fig S2. The results are very nice. The alternative possibility here is different stereoselectivity, which although ruled out by results is not addressed. Can you note somewhere that the absolute stereoselectivity of the carbonyl reduction is conserved across both SDRs and substrates but the enantioselectivity of the substrate is different.

We have made the requested change to Fig 2. Also, as suggested, we have added a summary sentence within our description of these enzymes to compare/contrast the activity of these two homologs.

"Taken together, these results demonstrate that these SDR enzyme homologs each catalyze the ketone reduction of **4** with conserved stereoselectivity to yield an alcohol in the (*S*) orientation, but also that they have different enantioselectivity, with *Pt*SDR-1 preferably reducing (*S*)-**4**, while *Pt*SDR-2 seems to act equally well on both enantiomers of **4** (Fig S2e)."

CYP78. Page 4 para1. Could 184.1332 and 124.1221 represent in-source fragmentation of 184.1332? Can you clarify why you think there are 2 peaks? Clarity about RT would help. Figure S4 – B - can we see the 2 peaks overlaid or arrayed vertically to judge the RT difference?

To clarify this point, we have added the peak retention time over each EIC trace in Fig S4b, and we have also added the retention times to the text to clarify that these *m/z* features do belong to distinctly eluting molecules. We note that these mass ions are within the same sample, and since they do not co-elute, we can rule out *m/z* 124 as an in-source fragment of *m/z* 184.

D – which **6** diastereomers are being fed here? Why do we not see 2 peaks?

The synthesized substrate **6**, which is the substrate used in this in vitro reaction, is a mixture of all diastereomers (no stereoselectivity during its synthesis). We can observe this molecule with either C18 or HILIC methods on our LC-MS; however, we only separate the diastereomers using C18, and these co-elute as a single peak during HILIC analysis. Many of the early, proposed Lycopodium alkaloid precursors (e.g. **7**, **8**, and **9**) can only be resolved via HILIC LC-MS, so that is why **6** is shown here with HILIC. We have added text to this figure caption, as well as the Methods (shown below), to clarify this point.

"In general, early pathways intermediates (compounds **3** through **9**) were observed with HILIC analysis, while downstream intermediates (compounds **10** through **25**) were observed with C18 analysis. We note that **6** in particular could be observed using either C18 or HILIC analysis. However, while diastereomers of **6** could be resolved with C18 analysis, these appeared to co-elute as a single peak within HILIC analysis."

F – there is the possibility that the ion abundance of 124 also represents a spontaneous deacylation – what do the other ion abundances look like over the pH range (substrate and 184)?

The abundance of **7** (*m/z* 184.1332), while much lower than that of **8**, follows the same pattern over the tested pH range of this enzyme assay. The consumption of **6** follows the same trend, with the biggest decrease observed where the abundances of **7** and **8** are greatest. We have added the data for **7** abundance to this figure panel to help clarify this.

G – what is the evidence that 7 to 8 is catalysed by CYP and not spontaneous/buffer catalysed?

We do not have direct evidence for the conversion of **7** to **8** being enzyme catalyzed, though we suspect this to be the case. We have made adjustments to the Results (shown below), figure, and figure caption to express this uncertainty.

"These results suggest a series of transformations in which the diastereomers of **6** are oxidized to produce **7**, which then undergoes an allylic elimination to yield **8** (Fig S4h, i), though it is uncertain whether the elimination is spontaneous or enzyme-catalyzed by *PtCYP782C1*."

H –this mechanism doesn't work for me – the simplest option is typical hydroxylation of that position with P450 O-rebound mechanism and elimination/dehydration to the imine. Here it appears you are suggesting a SET on the second step to directly form the positive charge? If so, where is the second proton coming from on the P450? Please revise.

We initially intended for the SET mechanism; thank you for catching this error. We have amended this scheme to more accurately reflect possible mechanisms for this reaction. Based upon your comment, as well as that of referee #1, we have provided two possibilities for how the initial imine formation occurs.

Page 5 para 2. Fig 2 and Fig S5. Why are you generally not showing 247 rather than 164? This would be more obvious. Is it because the peak is too small for good signal? In the main Fig 2 247 would be the ideal mass to show. Otherwise, you could depict on the structure of **9** what fragment you are observing. Figure S5. This is sufficient but you could have done MS³ from M+H 247 to 164. Similarly, do you see 247 in the clubmoss extract?

Your assumption is correct - we show *m/z* 164 rather than *m/z* 247 because the latter is >100x less abundant, and thus more difficult to routinely detect (especially at low abundance, where the detection of *m/z* 247 can be at or near the MS limit of detection). We do indeed see *m/z* 247 in the native extract of *P. tetrastrichus*, as well as the *m/z* 124 adduct ([M+2H]²⁺), and these data have been added to Fig S5e.

We agree that it should be abundantly clear why *m/z* 164 is the mass ion being depicted throughout the manuscript. As described in response to one of your earlier comments, we have added additional text to the figure caption of Figure 2, and to the Results section (shown below), that clarifies the use of *m/z* 164 as a diagnostic ion for routine detection of **9**.

"We note that of the three MS adducts observed for this molecule, the *m/z* 164 ion was the most abundant, and therefore this was used as a diagnostic ion for all following analyses."

Page 5, line 3 – where are we supposed to find the second 247 peak? And matching MS2 pattern? Is this Fig S5 L? If so, this is EIC 164 and not 247. In addition, the decrease in S-**4** is not significant – this needs to be referred to in the text as a caveat if the statement is to remain.

We were indeed referring to Fig S5l (now Fig S5m), and we now reference this figure panel specifically. Also, as mentioned above, we have added text to the Results clarifying that *m/z* 164 was the representative ion measured for detecting **9**. Finally, in line with comments from referee #1, we have decided to refer to 'mz247' as **9** throughout the Results to limit confusion (and we make a point to convey that we did not know the structure of **9** initially).

Regarding the decrease in (S)-**4**; we have added the statistical values shown in Fig S5I to our text in the Results. We also note that *in vitro* assays with apoplast extract further verify that (S)-**4** is the major enantiomer that is depleted by the activity of CAL1/CAL2 (current Fig 3e).

Page 5, Line 15 – this information on the oxidised compound is buried too deep the SI. It should be referred to in Figure 2 or 3 or in Figure S6, not in Fig S21-S24. In addition, the figure must be properly referred to in the text. Also if the NMR spectra is confident on the structure of **9'** then this is important evidence for your compound **9** identity so should be more prominently presented.

We have added text to the figure caption of Figure 2 pointing to the supporting figures that provide evidence for the structure of **9** (including the analysis of the oxidized by-product, **9'**). Also, as suggested we have moved Fig S21 to follow immediately after Figure S5 (so it becomes Figure S6); as per your earlier comment, we prefer to keep the NMR data together at the end of the SI document, though we make sure to specifically reference these figures in the captions of Figure 2, Figure S6, and in the Methods.

Page 5 – line 38 – remove “mainly” present as no comparative assessment was made. They are present in the apoplast. But, CAL2 expressed alone is not in the apoplast hence the whole cell extract which has strong signal for CAL2 but not in the apoplast. It seems maybe that CAL2 is expressed OK but not trafficked to correct compartment?

We have made this suggested edit to more accurately reflect the data, and we have reworked the section describing the localization data overall to provide additional nuance and clarity on our interpretation (shown below with changes highlighted). Also, we agree that proper protein trafficking/localization may be a key component to the observed co-dependence of CAL-1 and CAL-2, and some of our new text in the Results discusses this possibility.

“To assess this, we produced His-tagged versions of these proteins in *N. benthamiana*, and used Western blotting of different protein fractions (apoplast and cellular) to evaluate *PtCAL-1a* and *PtCAL-2a* localization. This demonstrated that both proteins can be found in the apoplast within this heterologous system, but also that this localization is impacted by their co-expression (**Fig 3b**). In particular, while *PtCAL-1a* exhibited apoplastic localization independently of co-expression with *PtCAL-2a*, very little *PtCAL-2a* protein could be detected in the apoplast when it was expressed alone, and it instead appeared to be mainly within the intracellular fraction, which contains both cytosolic and organellar proteins (**Fig 3b**). However, upon co-expression of *PtCAL-1a*, apoplastic *PtCAL-2a* was readily detected, and appeared to exhibit post-translational modifications of an unknown nature (**Fig 3b**). These results suggest that *PtCAL-1a* may have a critical role in the proper post-translational modification and/or trafficking of *PtCAL-2a*, though additional work will be necessary to determine the mechanism by which this occurs. Beyond providing details on localization, this information was critical for enzyme assay development since the pH of the apoplast is typically relatively low (~pH 5),³⁹ which suggested that these proteins may only exhibit function at a lower pH. Additionally, apoplast extracts can be readily isolated from *N. benthamiana* leaves expressing these proteins,³⁹ thereby providing a potential means to evaluate CAL protein outside of the living plant system.”

Figure 3 A – be clear about what steps you think are enzyme catalysed – either by drawing a box around the steps or labelling each straight-reaction arrow clearly. I.e. as it is drawn only the Nu-attack is CAL catalysed – what about decarboxylation, cyclisation and dehydration etc.

Similarly Figure 3G – which steps are you proposing CAL3 catalyses?

We have altered the mechanistic schemes in Fig 3 (for both CAL-1/CAL-2 and CAL-3) to indicate the general chemical transformations that we propose to be catalyzed in each CAL-mediated reaction, and indicate where we think a spontaneous (non-enzymatic) reaction is occurring. We are hesitant to speculate further on each individual step of the proposed mechanism without further experimental support, and we note that while these mechanisms are plausible and consistent with our data, the specific reactions shown in this figure are hypothetical.

Fig 3B – highly selective blot – can you include all blots including whole cell extract – I am not sure that CAL2 is poorly expressed but it appear to not be trafficked correctly.

We now include the full Western blots in Figure 3b, and as described above, we have updated our language within the Results to describe these data more accurately. In order to make room for the full Western blot in Figure 3, we have omitted the comparison of heterologous **9** (produced in *N. benthamiana*) to the extract of *P. tetrastichus* (formerly Fig 3f); these data are now presented in Fig S5e/f.

Page 6 line 22 = "... could also leverage CO₂ release to drive the reaction equilibrium." I do not understand what this means – perhaps "drive to completion" could be a better way of phrasing. However, CO₂ release occurs in the decarboxylation-first mechanism too, so I am not sure how the later CO₂ release is energetically preferable. Please clarify

We have made the suggested change (see below). We note that this is intended as a comparison to the 'decarboxylation-first' mechanism, which would similarly be relying on CO₂ release to drive the equilibrium towards product. Our intent is to describe that either mechanism would leverage this.

"While a "decarboxylation-first" mechanism is reminiscent of several canonical strategies for C-C bond formation (e.g. in fatty acid biosynthesis), an "addition-first" mechanism could also leverage CO₂ release to drive the reaction equilibrium to completion, and thus seemed to be a plausible alternative."

Figure S6 G – this is repeated in Figure 3E.

Thank you for catching this. We have removed Fig S6g.

Page 7, line 21 – What on Fig 4 is this referring to? Unclear. This should not be Fig S33 but earlier in the SI. Also, I do not understand why you need to evoke mechanism 3 instead of mechanism 2 for PIKS. Enzymes can act non-stereoselectively so you can surely just keep mechanism 2 if you need to evoke PIKS catalysis.

Mechanism 3 is unlikely as the open ring form is very unflavoured in water. There is no evidence for this at all, in fact I find the PIKS mechanism discussion a little speculative and does not add much to the work.

Our intent was to provide an alternative explanation for how a racemic mixture of **4** is produced biosynthetically by PIKS, especially since this lack of enantioselectivity seems to be a relatively uncommon occurrence. Because we have previously provided evidence for the condensation reaction that forms **3** to be enzyme-catalyzed by PIKS, we hoped to provide a mechanism that could satisfy both the enzyme-catalyzed condensation, as well as the racemic product outcome.

With that said, our proposal is not directly supported by any experimentation in this manuscript, so we have removed it from the results (both the text description and the SI figure).

Page 8, line 25 - 8,15-double bond can apparently be found numbered in Fig 1A. Does this mean atom numbers? If so, they are not visible on the figure.

We have added select carbon numbering in Fig 1a to help clarify this point, and we have added a reference to Fig S1 for full carbon numbering.

Page 8 line 31 – Which Supplementary information files?

We have edited this to describe that in-depth text descriptions of these results can be found in the SI Results.

Page line 38 . "If these were the oxidations..." this reads a bit strangely, please rephrase.

We have rephrased this sentence for clarity.

"If our predictions for the oxidations catalyzed by *Pt2OGD-4* and *Pt2OGD-5* were correct, then the only remaining oxidation would be A-ring desaturation, which we have shown to be catalyzed by *Pt2OGD-3*.²²"

Figure S15 – should this be positioned earlier as it is referred to earlier in the paper?

We do not discuss the presence of each biosynthetic intermediate within the native plant (with the exception of **9**) until this section. Also, for the sake of continuity, we prefer to keep this description and figure (now Fig S18) immediately before that of the downstream pathway (now Fig S19).

Figure 5 – impressive bank of reactions. Please clarify on each plot whether the compound ID was validated by (i) comparison to standard, (ii) comparison to related standard (i.e. NAc derivatives) or (iii) putative iD.

As described above with your earlier comment, we have a sentence in the figure caption indicating that all 'named' molecules in this figure have been verified with an authentic standard, and we have added 'proposed' under all compounds in Figure 5 that do not have an authentic standard. We have also added

an additional sentence in the figure caption that clarifies the proposed structures for 'un-named' intermediates and points the reader to the SI for the supporting structural evidence.

Page 9, line 20-23. This is an interesting observation but the interpretation is speculative. Whilst yes I do think it shows a "metabolic SAR among the alkaloids" the evolutionary origin is unclear and I don't think this should be stated as the only interpretation. You are taking the "forward" model of step-by-step evolution. There is an alternative "retrograde" model where the bioactive product huperzine was present as one product of a complex network catalysed by promiscuous enzyme ancestors. Only after it became selected for did the more specific / selective enzyme steps emerge. This fits with your network concept too. So practically, yes, the IC50 improves along the pathway but the evolutionary origin is unknown. Interestingly your CAL enzyme identification matches the "patchwork" ideas of metabolic evolution too.

We have added additional text to this paragraph that indicates the hypothetical nature of this proposal, as well as the possibility for alternative explanations, much as you have described.

"While we cannot be sure that the biological function of **17** is to inhibit animal AChE enzymes, the relationship between Lycopodium alkaloid biosynthesis and AChE inhibitory activity suggests that this metabolic pathway has evolved successive biosynthetic steps that increase the potency of these alkaloids step-by-step to achieve the production of an "optimized" AChE inhibitor. However, we note that alternative explanations for the evolution of **17** biosynthesis are plausible, particularly given the complex, metabolic network of Lycopodium alkaloids that exists in extant plants. For example, it is possible that **17** was a minor component of the Lycopodium alkaloid cocktail present in a shared common ancestor, and that the AChE activity of **17** was selected for, thereby refining and enhancing the biosynthetic production of this molecule. Regardless of the specific mechanism, the Lycopodium alkaloids could prove to be a powerful system for understanding the evolution of specialized metabolism in early diverging plants."

Page 9 line 42 – What is the evidence, aside from your own identification of CALs, that they have wider roles? You have not identified many outside clubmoss with different Zn-motifs, for example.

In short, we do not yet have evidence for this beyond identifying a relatively small number of CAHs with mutations to either the His triad or other conserved active site residues (Figure S10); this is simply a working hypothesis. Since our work has indicated that CAH enzymes can neofunctionalize within specialized metabolism, we think that it is worthwhile to discuss the possibility for this to be a broader trend in plants.

While the evidence shown in Figure S10 only highlights a few examples of other CAHs with mutations to the His triad, we note that this phylogenetic tree is a relatively small sampling of the CAHs within the plant kingdom. Given the fact that we found potential examples of neofunctionalized CAHs (via His triad mutation) in this small sample size, we are optimistic that future, comprehensive analyses of large databases (e.g. 1KP, GenBank, or UniProt) will lead us to other CAHs with analogous mutations. Finally, we point out that mutations to other residues other than the His triad could lead to neofunctionalization (as is the case for CAL-3), and if we take these into account, the number of potential neofunctionalized candidates will likely

be even higher. Regardless, this will all require future investigation and is an open area of research for our groups.

Note for all referees:

- During our revision, we noticed an inconsistency in the numbering of two active site residues within the protein alignments/trees shown in Figure 4b and Figure S10. One of these, W208 (previously annotated as W209) was simply a typo, and none of the residues in the tree change. The other, T198 (previously T199) was misannotated due to an apparent discrepancy between the literature and the sequence of human CA2 that was retrieved from UniProt. T198 is immediately adjacent to a second highly conserved threonine (T199), and we mistakenly used the aligned residues for this threonine in our initial submission. We confirmed that T198 is actually the gate-keeping residue, so we have updated our alignments/trees to include this residue. We note that the overall theme does not change - T198 is conserved in nearly all CAH family proteins, but is mutated in one of the three CALs identified herein.

Reviewer Reports on the First Revision:

Referees' comments:

Referee #1 (Remarks to the Author):

The authors have responded to the reviewers' concerns. The manuscript is now acceptable for publication in the journal.

Referee #2 (Remarks to the Author):

The authors have thoughtfully addressed my previous comments. I do not have additional comments.

Referee #3 (Remarks to the Author):

The authors have thoroughly and thoughtfully responded to all my comments. I recommend the manuscript for publication and congratulate the authors on an excellent contribution to the field.